# HUMAN SIMULACRA: BENCHMARKING THE PERSONIFICATION OF LARGE LANGUAGE MODELS

**Qiujie Xie**[1] **Qiming Feng**[1] **Tianqi Zhang**[2] **Qingqiu Li**[1] **Linyi Yang**[3,4]
**Yuejie Zhang**†[1] **Rui Feng**†[1] **Liang He**[5] **Shang Gao**[6] **Yue Zhang**[7,8]
[1] School of Computer Science, Shanghai Key Lab of Intelligent Information Processing,
Shanghai Collaborative Innovation Center of Intelligent Visual Computing, Fudan University
[2] Tongji University [3] University College London [4] Huawei Noah's Ark Lab
[5] Shanghai Key Laboratory of Mental Health and Psychological Crisis Intervention
[6] Deakin University [7] Westlake University [8] Westlake Institute for Advanced Study

## ABSTRACT

Large Language Models (LLMs) are recognized as systems that closely mimic aspects of human intelligence. This capability has attracted the attention of the social science community, who see the potential in leveraging LLMs to replace human participants in experiments, thereby reducing research costs and complexity. In this paper, we introduce a benchmark for LLMs personification, including a strategy for constructing virtual characters' life stories from the ground up, a Multi-Agent Cognitive Mechanism capable of simulating human cognitive processes, and a psychology-guided evaluation method to assess human simulations from both self and observational perspectives. Experimental results demonstrate that our constructed simulacra can produce personified responses that align with their target characters. We hope this work will serve as a benchmark in the field of human simulation, paving the way for future research.

## 1 INTRODUCTION

Researchers in psychology and sociology have long relied on human participants to conduct experiments that explore patterns of human behaviors and mental states (Camerer et al., 2018; Folke et al., 2016; Qiu et al., 2017). However, this method often faces numerous challenges, including the difficulty of recruiting participants (Radford et al., 2016; Belson, 1960), high uncertainty (Haslam & McGarty, 2001), and potential ethical considerations (El-Hay, 2019). In this context, the potential of large language models (LLMs) to mimic human behaviors has garnered increasing attention (Ziems et al., 2024; Zhang et al., 2023a; Coda-Forno et al., 2024). Psychologists and sociologists are exploring the use of LLMs to replace human participants, aiming to reduce costs and complexity while avoiding potential ethical considerations (Demszky et al., 2023; Dillion et al., 2023; Hutson, 2023; Grossmann et al., 2023; Li et al., 2023b; Kjell et al., 2023).

Despite these advancements, current LLM-based human simulations are still limited in only simulating group studies (Li et al., 2023b; Zhao et al., 2024), giving rather inconsistent performance across different tasks (Dillion et al., 2023; Hutson, 2023), and lack the depth in capturing complex characteristics of human behaviors (Grossmann et al., 2023; Kjell et al., 2023; Hagendorff et al., 2023; Yin et al., 2024; Jones & Bergen, 2024). To address these issues, we consider a different perspective, proposing a psychology-driven simulacrum that aims to produce consistent behaviors indistinguishable from humans. To this end, we introduce a high-quality dataset, a comprehensive evaluation pipeline, and a unified benchmark, as shown in Figure 1. Using the proposed benchmark, we empirically discuss the research question: **How far are LLMs from replacing human subjects in psychological and sociological experiments?**

Rigorous personality modeling is crucial for human simulation as it ensures more realistic representations of human behavior and interactions. However, personality is a complex concept that is difficult to model. Prior studies in personality modeling (Pan & Zeng, 2023; Tu et al., 2023; Song et al., 2024; Wang et al., 2023b) use well-established frameworks like the Myers-Briggs Type Indicator (MBTI) (Myers, 1962; tse Huang et al., 2024; Huang et al., 2023). Despite the popularity,

Figure 1: Overview of the proposed benchmark.

these personality models face critical limitations. For example, LLMs' internal representation of psychological types may be inherently flawed or hallucinatory, given that they operate as "black boxes." Besides, personality is a multifaceted and dynamic construct that cannot be accurately reduced to a single type by these models. Inspired by Jung's psychology theory (Corr & Matthews, 2020; Mussel et al., 2016; Hogan et al., 1997; Jung, 1923), we employ an eight-dimensional strategy to address the LLM-based personality modeling issues. By dividing personality into eight complementary tendencies, we provide a more comprehensive framework with 640 detailed trait descriptions (Table 7). This approach allows for a more nuanced depiction of personality (§3.2) when constructing virtual characters, enhancing the variety of characters in "personified machines".

Identifying suitable targets for human simulation also poses significant challenges. One approach involves using role-playing datasets composed of fragmented information about **genuine characters** (Wang et al., 2023c; Zhou et al., 2023; Shao et al., 2023) (e.g., Albert Einstein, Beethoven). However, the simulations of existing characters are prone to be disrupted by hallucination (Mallen et al., 2022; Wang et al., 2023a) produced by LLMs. The fragmented data also fail to provide a comprehensive depiction of a character, especially for psychological experiments. Consequently, we build a **virtual character** dataset, named **Human Simulacra**, and use the characters' **detailed life stories** as the basis for simulations, which also avoids the potential *ethical and legal risks* of using historical figures. To this end, we decompose the task of crafting a detailed life story into interrelated subtasks and further propose a human-in-the-loop strategy that tackles each subtask with human feedback (§3). Our dataset contains 129k texts across 11 virtual characters, with each character having unique attributes, biographies, and stories (Figure 9).

Given the complexity of the human simulation, we propose a novel evaluation framework for measuring the "personified machines". We expand the traditional self-report method (Park et al., 2023; Gupta et al., 2024; Li et al., 2024) to a two-phase evaluation method, combining self reports (§4.1) and **observer reports**(§4.2), based on established personality measurement theories (Corr & Matthews, 2020; Mussel et al., 2016; Hogan et al., 1997; Jung, 1923). Our evaluation provides a suitable and robust testbed for exploring the opportunity of replacing human participants with LLM agents. Furthermore, to mimic the complex nature of human beings, we introduce a novel **Multi-Agent Cognitive Mechanism (MACM)** that simulates the human brain's information processing systems (§4.3). As an external module, this mechanism enables the LLMs to remember background stories, understand target personalities, and express accurate emotions in complex situations.

Based on our Human Simulacra dataset, we conduct an empirical study involving 14 widely-used LLMs with 4 different auxiliary methods (*None*, *Prompt*, *Retrieval Augmented Generation (RAG)*, and *MACM*) using 3 experimental settings (*self reports*, *observer reports*, and *psychology experiment on conformity*). Extensive results reveal that although the top-performing model approaches human performance levels (88.00% on GPT-4-Turbo) in self-report evaluations, it struggles in observer reports, achieving only 77.75% even with MACM support. In our conformity test, LLM agents exhibited submissive responses similar to humans, albeit with a more robotic and rigid demeanor.

To our knowledge, we are the first to build human simulation data based on Jung's psychology theory (Jung, 1923) and conduct standard human simulation experiments. We offer high-quality data, rigorous and innovative evaluation methods, and comprehensive benchmark tests. Our findings suggest the potential use of LLM agents as substitutes for humans in psychological experiments, shedding light on future applications of human simulacra.

## 2 RELATED WORK

**Memory Systems in Cognitive Psychology.** In cognitive psychology, information processing approaches assert that cognition encompasses the entire process through which sensory inputs are transformed, reduced, elaborated, stored, retrieved, and used (Neisser, 1976; Newell et al., 1972;

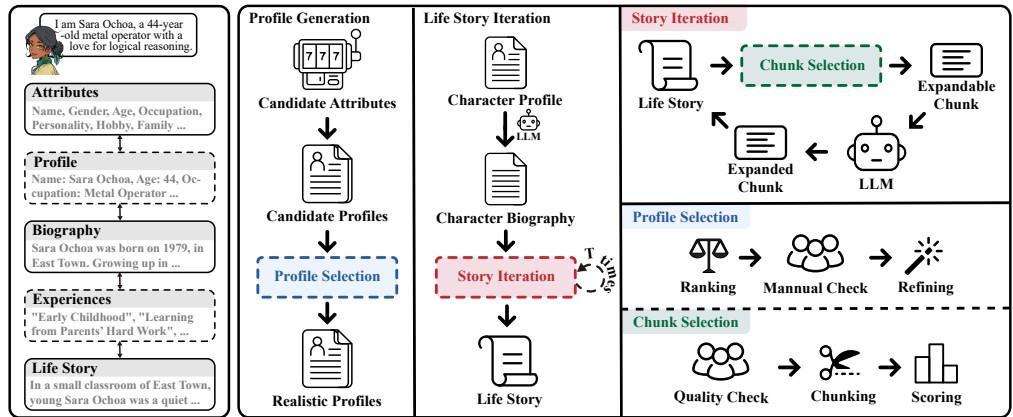

Figure 2: Process of constructing life stories for characters. At each step, humans are involved in thoroughly reviewing the generated content, ensuring it is free from biases and harmful information.

Dawes et al., 2020; Paas & van Merriënboer, 2020). Atkinson & Shiffrin (1968) proposed a Multi Store Model of memory that divides memory into sensory memory, short-term memory, and long-term memory. Baddeley & Hitch (1974) distinguished the concept of working memory from short-term memory, emphasizing that working memory is born for storing, invoking, and analyzing information. In this paper, based on the memory theories (Atkinson & Shiffrin, 1968; Baddeley & Hitch, 1974; Baddeley et al., 1984; Izquierdo et al., 1999; Norris, 2017) discussed above and the capabilities of LLMs (Zhao et al., 2023), we propose a Multi-Agent Cognitive Mechanism. It is designed to enhance the ability of LLMs to impersonate humans by transforming a narrative life story into long-term memories and engaging with the external world in a human-cognitive manner.

**Role-playing.** Role-playing tasks (Chen et al., 2024) focus on simulating characters with distinctive personalities. This line of work includes replicating the professional skills (Salewski et al., 2024; Hong et al., 2023; Binz & Schulz, 2023) and portraying the outward characteristics (Shao et al., 2023; Tu et al., 2023; Wang et al., 2023c; Li et al., 2023a; Yu et al., 2024; Zhou et al., 2023) of target personas. Our work differs

Table 1: Differences between Human Simulacra and Role-playing datasets.

| Features | Ours | Character-LLM[1] | Role-LLM |
|---|---|---|---|
| Virtual Characters | ✔ | ✗ | ✗ |
| Life Story | ✔ | ✗ | ✗ |
| Psychology Support | ✔ | ✗ | ✗ |
| Human Feedback | ✔ | ✗ | ✔ |

from existing role-playing studies in two key aspects: 1) Our work is grounded in psychological theories to ensure rigor in deep simulation of human personalities. Role-playing works do not need to follow psychological principles like our method does, and they are not intended for uses that require a deep imitation of human patterns (e.g., instinct (Tinbergen, 2020; Marler, 2014) , conditioning (Clark et al., 2002)). 2) While role-playing works typically portray characters from the past whose characteristics are generally fixed and immutable, our method constructs and simulates virtual human beings from scratch. This allows us to explore various personality combinations without being limited to existing roles and mitigates the factual errors or hallucinations that LLMs may generate when simulating real people. The difference between our dataset and existing role-playing datasets is shown in Table 1. We propose the first simulacra dataset that combines features of "virtual characters", "life story", "psychology support", and "human feedback".

## 3 HUMAN SIMULACRA DATASET

We break down the generation of character data into solvable sub-tasks (e.g., profiles and short biographies) by introducing a structured information model as shown in the left part of Figure 2. This model organizes the character's information into five inter-connected layers (e.g., character attributes and character biography). From bottom to top, the information becomes more concise, focusing on the character's most essential facts. Based on this information model, we decompose the task of generating a character's life story into interconnected subtasks and design a semi-automated strategy to iteratively build a detailed life story for the target character. The entire process is depicted in the

---

[1]Character-LLM (Shao et al., 2023), Role-LLM (Wang et al., 2023c)

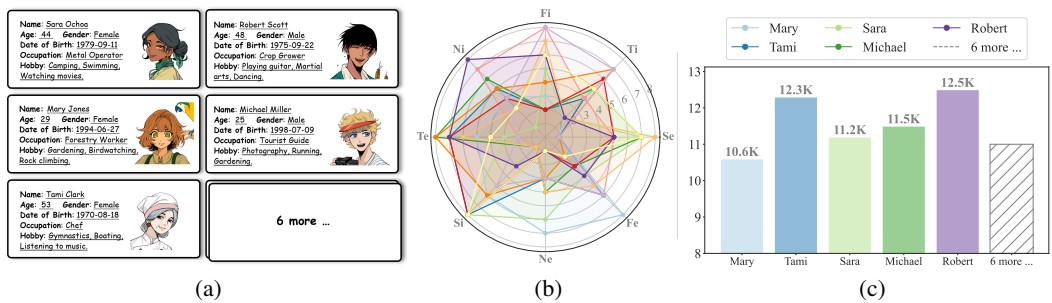

Figure 3: Human Simulacra dataset. (a) Profiles of virtual characters. (b) Personalities of characters, displayed in radar chart based on Jung's eight-dimensional theory. Line: character; Te / Si: abbrevs for personality dimensions. (c) Word count of life stories for each virtual character.

right part of Figure 2. In particular, we first generate 100 candidate profiles (varying in quality) and select 11 virtual characters as the protagonists based on their backgrounds. The selection details are provided in Appendix A.3. We then employ the GPT-3.5-Turbo model (Brown, 2020) as the data generator with a frequency_penalty of 1.0 and top_p of 0.95. Each life story is expanded through at least 50 rounds of iteration. At the end of each story iteration, multiple human reviewers, including graduate students in computer science and psychology, thoroughly review the content to ensure it is free from biases, discrimination, or harmful information.

## 3.1 CHARACTER ATTRIBUTES

Character attributes encapsulate the core facts of a virtual character, serving as anchor points for the life story of the character. While designing attributes, it is necessary to ensure that the attributes are diverse, have reasonable connections, and conform to natural laws. Following previous studies (Sloan, 2015; Park et al., 2023), we design a comprehensive attribute set for virtual characters, encompassing {*name*, *age*, *gender*, *date of birth*, *occupation*, ***personality traits***, *hobbies*, *family background*, *educational background*, *short-term goals*, and *long-term goals*} (Figure 3a). Each attribute has a candidate pool, covering diverse values applicable to most people. For instance, based on the International Standard Classification of Occupations (ISCO-08), we select 76 common occupations as the occupation candidate pool. More details about the attribute systems are provided in Appendix A.2.

## 3.2 PERSONALITY MODELING

Considering that personality encompasses an entity's characteristic patterns of thought, feeling, and behavior (Hogan et al., 1997), how to accurately model the personality traits of the target character becomes a core challenge in attribute design. We adopt the eight-dimensional theory derived from Jung's study (Jung, 1923) to accurately model the personality traits of the target character. This theory divides personality into eight tendencies such as extraverted thinking (Te) and introverted sensing (Si), with each tendency serving as a complementary facet.

Contrary to directly assigning numerical values to these tendencies, we employ a relative ranking strategy to indirectly assess the strength of each personality tendency within the character. Specifically, we rank the eight tendencies and establish a guideline that the tendencies at the top and bottom of the order are more pronounced in the character's personality, while those in the middle are less pronounced, manifesting a blend of traits that vary in direction. Under the guidance of psychology professionals, we prepare 10 suitable descriptions for each possible ranking, with each description corresponding to an aspect of the tendency in daily life. Ultimately, we form a personality candidate pool containing 640 trait descriptions (Figure 3b). Our personality modeling method grounded in authoritative, field-recognized theories (Jung, 1923), aims to depict the character's personality more comprehensively and specifically. Example descriptions for the extraverted intuition tendency are deferred to Appendix Table 7.

## 3.3 CHARACTER PROFILE AND LIFE STORY GENERATION

To assemble the character's profile, we first generate draft profiles by randomly selecting attribute values from their corresponding pools. Then, we add a Profile Selection module responsible for

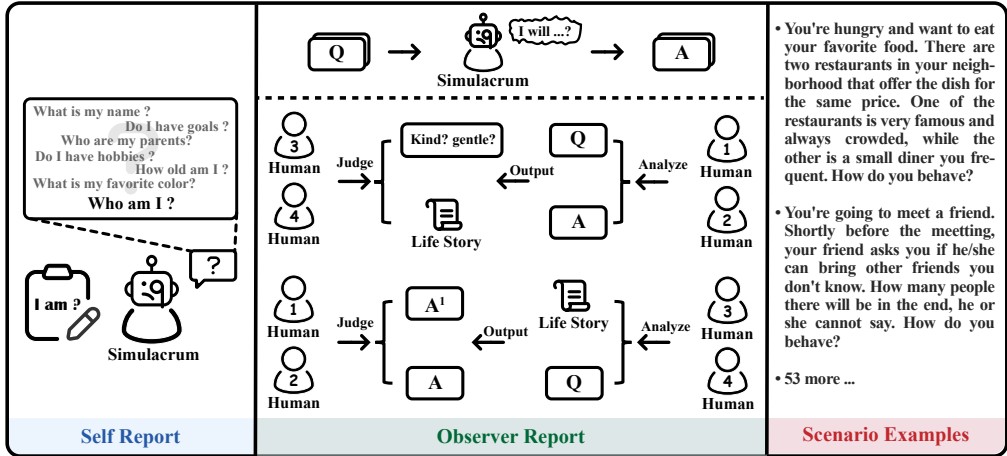

Figure 4: Psychology-guided evaluation. Self reports assess simulacra's self-awareness through character-specific questions based on their life stories. Observer report evaluates simulacra's realism by creating scenario-based assessments analyzed by human judges.

quality check and profile refinement in the generation process, as shown in the right part of Figure 2. In this way, high-quality profiles are manually filtered out and fed into the LLM to generate a short biography summarizing the character's life experience.

After obtaining the brief biography for the character, we use an iterative generation method to progressively enrich the biography with human feedback, transforming it into a detailed life story after $T$ iterations. Specifically, in each iteration, we perform: 1) Quality check: manually inspect the generated content for its rationality, and ensure it is free from biases, discrimination, or harmful information; 2) Chunking: divide the story into separate chunks; 3) Scoring: for each chunk, calculate its **Importance**, **Elaborateness**, and **Redundancy**, then select chunks with high importance, low elaborateness and redundancy for expansion; and 4) Expanding: prompt the LLM to expand the selected chunks and add reasonable life experiences to the story. Finally, we create the virtual character dataset Human Simulacra, comprising about 129k texts across 11 virtual characters (Figure 3c). See Algorithm 1, Appendix A for construction details, and Appendix F for relevant prompts.

## 4 PSYCHOLOGY-GUIDED EVALUATION

We propose a psychology-guided evaluation framework as shown in Figure 4. This framework draws on psychological assessment techniques (Hogan et al., 1997; Mussel et al., 2016), including self reports, observer reports, and the Multi-Agent Cognitive Mechanism to generate responses.

### 4.1 SELF REPORT

Self-reporting is a common personality measurement technique that requires individuals to answer questions about themselves (Hogan et al., 1997; Corr & Matthews, 2020). It refers to the degree to which an individual is aware of their own identities, thoughts, and values. We employ self-report assessments to evaluate the simulacra's ability to establish self-awareness, testing their memory and analytical capabilities regarding their character information. To this end, we manually craft a set of questionnaires for each virtual character, featuring fill-in-the-blank and single/multiple-choice questions. Each question is carefully reviewed to ensure they reflect the character's unique nature and the scores are evaluated based on exact matches. The test content covers key attributes, social relationships, and life experiences of the target characters. For example, *"What is your name?"*, *"What do you think of your father?"*, and *"What were the reasons behind not going through formal schooling for you?"*. See Appendix D.1 for additional example questionnaires.

### 4.2 OBSERVER REPORT

Self-assessment tests are insufficient measures of LLM personality due to potential biases and the inability to capture complex human behaviors accurately (Gupta et al., 2024). A high self-report

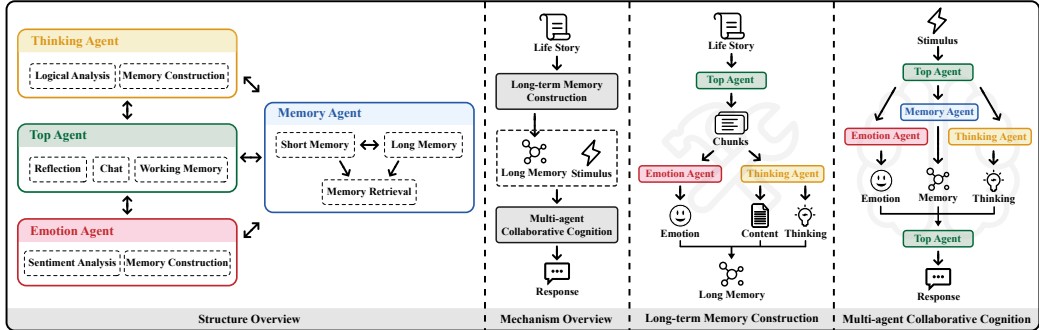

Figure 5: Multi-Agent Cognitive Mechanism. It involves four LLM-driven agents: Thinking Agent / Emotion Agent handles logical/emotional analysis & memory construction. Memory Agent manages retrieval of memories, while Top Agent coordinates all activities. Upon receiving a stimulus, these agents collaborate to generate appropriate responses, simulating complex human cognitive processes.

score only indicates that the simulacrum possesses a clear understanding of the target character. It does not sufficiently prove the simulacrum's ability to adopt behaviors consistent with their character in real-life scenarios. For a comprehensive evaluation, we need to further observe the simulacrum's thinking, emotions, and actions in real-life scenarios from a third-party perspective. Therefore, in addition to self reports, we further introduce observer reports, a cross-evaluation based on human judges, aiming to assess the simulacrum's thinking, emotions, and actions in real-life scenarios from a third-party perspective.

Specifically, following Mussel et al. (2016), we crawl 55 hypothetical scenarios that could elicit human emotional responses or personality traits. Two examples of such scenarios are displayed in the right part of Figure 4. We require each simulacrum to imagine that they are in the given scenario and to describe how they would feel and what actions they would take. All responses are collected and submitted for cross-evaluation, which includes two inter-related subprocesses: 1) Human judges 1 and 2 analyze the scenario (Q) and response (A), and describe the respondent's personality. Subsequently, judges 3 and 4, informed by the target character's life story, determine whether the descriptions given by judges 1 and 2 match the target character. A discrepancy indicates that the simulacrum has deviated from the character, showing a simulation error. 2) Considering potential bias in a single assessment, we ask judges 3 and 4 to thoroughly read the target character's life story, and answer how they would feel and what actions they might take in the scenario if they were the character. Then, judges 1 and 2 compare the similarity between the human responses and the simulacrum's responses. A high degree of similarity indicates a high-quality simulation, which is consistent with the expectations of the character. More examples of hypothetical scenarios, the selection criteria for human judges, and the evaluation guidelines are provided in Appendix D.2.

## 4.3 MULTI-AGENT COGNITIVE MECHANISM

Following the aforementioned process in §3, we craft a life story for each virtual character. Given the limited context, current LLMs may not be able to accurately capture the character's personality and inherent emotional tendencies from the narrative. To address this issue, we propose a Multi-Agent Cognitive Mechanism (MACM, Appendix B) based on cognitive psychology theories (Atkinson & Shiffrin, 1968; Norris, 2017). This mechanism utilizes multiple LLM-based agents to simulate human brain's information processing and memory systems, thereby enhancing the quality of simulacra. The ablation study on the structure of MACM is provided in Appendix B.3.

As illustrated in Figure 5, MACM has two key processes: 1) Long-term memory construction and 2) multi-agent collaborative cognition. Specifically, When simulating a character from the Human Simulacra dataset, MACM first transforms the target character's narrative life story into long-term memories that are richer in detail, fuller in emotion, and clearer in structure. Then, to mimic human cognition, MACM further utilizes a collaborative process that allows LLMs to leverage long-term memory and engage with the external world in a cognitive manner. Upon receiving a stimulus, for example, a question from a friend, Top Agent first analyzes the question and evokes Memory Agent for memory retrieval. The retrieved results are stored in working memory. Then, Top Agent sends the relevant memories and question to Thinking Agent and Emotion Agent for logical and emotional

Table 2: Self reports of 14 LLM-based simulacra. Each character is tested by its own set of questionnaires containing cloze, single-choice (SC) questions, and multiple-choice (MC) questions. The simulacra are divided into different groups based on their parameter size. The best-performing simulacrum in each group is highlighted in light gray.

| Method | None | | | | Prompt | | | | RAG | | | | MACM (Ours) | | | |
|---|---|---|---|---|---|---|---|---|---|---|---|---|---|---|---|---|
| Question Type | Cloze | SC | MC | Sum | Cloze | SC | MC | Sum | Cloze | SC | MC | Sum | Cloze | SC | MC | Sum |
| GPT-4 | 0.00 | 8.00 | 12.00 | 20.00 | 20.00 | 20.00 | 38.67 | **78.67** | 20.00 | 20.00 | 42.67 | 82.67 | 20.00 | 20.00 | 46.67 | 86.67 |
| GPT-4-Turbo | 0.00 | 8.00 | 4.00 | 12.00 | 18.67 | 20.00 | 40.00 | **78.67** | 20.00 | 20.00 | 45.33 | 85.33 | 20.00 | 20.00 | 48.00 | **88.00** |
| Claude-3-Opus | 0.00 | 2.67 | 8.00 | 10.67 | 18.67 | 20.00 | 38.67 | 77.33 | 0.00 | 13.33 | 38.67 | 52.00 | 20.00 | 20.00 | 41.33 | 81.33 |
| Llama-2-7b | 0.00 | 5.33 | 8.00 | 13.33 | 10.67 | 16.00 | 17.33 | 44.00 | 0.00 | 9.33 | 6.67 | 16.00 | 9.33 | 8.00 | 8.00 | 25.33 |
| Vicuna-7b | 0.00 | 8.00 | 4.00 | 12.00 | 14.67 | 12.00 | 14.67 | 41.33 | 1.33 | 9.33 | 10.67 | 21.33 | 13.33 | 6.67 | 9.33 | 29.33 |
| Mistral-7b | 0.00 | 8.00 | 0.00 | 8.00 | 20.00 | 16.00 | 14.67 | 50.67 | 1.33 | 13.33 | 21.33 | 36.00 | 17.33 | 18.67 | 16.00 | 52.00 |
| Llama-2-13b | 0.00 | 8.00 | 9.33 | 17.33 | 9.33 | 9.33 | 12.00 | 30.67 | 0.00 | 8.00 | 13.33 | 21.33 | 9.33 | 4.00 | 9.33 | 22.67 |
| Vicuna-13b | 0.00 | 9.33 | 9.33 | 18.67 | 20.00 | 17.33 | 18.67 | 56.00 | 0.00 | 14.67 | 14.67 | 29.33 | 14.67 | 14.67 | 16.00 | 45.33 |
| Claude-3-Haiku | 0.00 | 6.67 | 14.67 | 21.33 | 20.00 | 20.00 | 25.33 | 65.33 | 5.33 | 12.00 | 36.00 | 53.33 | 20.00 | 20.00 | 24.00 | 64.00 |
| Mixtral-8x7b | 0.00 | 10.67 | 8.00 | 18.67 | 16.00 | 20.00 | 24.00 | 60.00 | 1.33 | 17.33 | 22.67 | 41.33 | 12.00 | 16.00 | 21.33 | 49.33 |
| Llama-2-70b | 0.00 | 9.33 | 2.67 | 12.00 | 16.00 | 17.33 | 14.67 | 48.00 | 0.00 | 5.33 | 12.00 | 17.33 | 20.00 | 17.33 | 18.67 | 56.00 |
| Llama-2-70b-Chat | 0.00 | 10.67 | 6.67 | 17.33 | 16.00 | 16.00 | 16.00 | 48.00 | 4.00 | 13.33 | 18.67 | 36.00 | 20.00 | 20.00 | 18.66 | 58.66 |
| Qwen-turbo | 0.00 | 9.33 | 14.67 | 24.00 | 16.00 | 20.00 | 33.33 | 69.33 | 20.00 | 20.00 | 32.00 | 72.00 | 20.00 | 20.00 | 34.67 | 74.67 |
| Claude-3-Sonnet | 0.00 | 8.00 | 13.33 | 21.33 | 18.67 | 20.00 | 36.00 | 74.67 | 0.00 | 13.33 | 38.67 | 52.00 | 20.00 | 20.00 | 36.00 | 76.00 |
| Human | 20.00 | 20.00 | 60.00 | 100.00 | - | - | - | - | - | - | - | - | - | - | - | - |

Table 3: Observer reports of different simulacra on GPT-4-Turbo. Description Matching Score evaluates simulacrum's alignment with target personality. Response Similarity Score estimates similarity between external expectations and simulacrum's behaviors. ICC represents the Intraclass Correlation Coefficient between judges.

| Method | Description Matching Score | | | | Response Similarity Score | | | | Final Score |
|---|---|---|---|---|---|---|---|---|---|
| | Judge 3 | Judge 4 | Average | ICC | Judge 1 | Judge 2 | Average | ICC | |
| Prompt | 32.00 | 33.00 | 32.50 | | 39.00 | 34.00 | 36.50 | | 69.00 |
| RAG | 39.00 | 36.00 | **37.50** | 0.86 | 28.00 | 28.00 | 28.00 | 0.95 | 65.50 |
| MACM (Ours) | 35.00 | 36.00 | 35.50 | | 41.00 | 43.00 | **42.00** | | **77.50** |

analysis and stores the outcomes in working memory. Finally, Top Agent formulates a response based on the contents of working memory. Due to the limited context window of LLMs, content that cannot be accommodated in working memory is dynamically transferred to short memory, which will be converted into long-term memory when rehearsed.

## 5 EXPERIMENTS

In this section, we introduce the empirical study involving 14 widely-used LLMs with 4 different simulation methods (*None*, *Prompt*, *RAG*, and *MACM*) using 3 experimental settings (*self reports*, *observer reports*, and *psychology experiment on conformity*) on the Human Simulacra dataset.

### 5.1 PSYCHOLOGY-GUIDED EVALUATION RESULTS

**Experimental settings.** To evaluate the human simulation ability of different LLMs, we experiment with 14 mainstream LLM-based simulacra using the psychology-guided evaluation method proposed in §4. We compare the proposed MACM with the following methods: 1) Blank model, which does not know any information about the target character. 2) Prompt-based method. We prompt the LLM to simulate the target character, with the help of the character's attributes and brief biography. 3) Retrieval-augmented generation method. A combination of prompt-based method and a retrieval module. In this case, the retrieval module searches the character's life story based on the input and returns the three most relevant paragraphs.

**Self reports.** Table 2 presents the results of self reports, with all outcomes being the average of three repeated tests. Based on the data presented in Table 2, we have the following observations:

1) Even without any knowledge of the target character, the LLMs can still score certain points (e.g., 12.00 on Vicuna-7b-None) on these single- or multiple-choice questions by random guessing. This indicates that relying solely on these questions for evaluation is not sufficiently reliable.

2) As the size of the LLMs' parameters increases, their capability gradually increases, leading to clearer self-awareness and an upward trend in self-report scores. For instance, when comparing Vicuna-7b-RAG with Vicuna-13b-RAG, the score increases from 21.33 to 29.33.

3) Since the self-report test is conducted in a conversational manner, the LLMs fine-tuned for conversational scenarios tend to perform better than foundation models (e.g., 25.33 on Llama-2-7b-MACM and 29.33 on Vicuna-7b-MACM).

4) While the RAG-based simulacra can retrieve relevant life story chunks when answering questions, **their performance is constrained by the LLMs' information processing capacities.** A large amount of descriptive information may interfere with the LLMs' self-positioning, resulting in inappropriate responses or misunderstanding of questions. Hence, in most weaker-performing LLMs, RAG-based simulacra score lower than Prompt-based ones.

5) In stronger-performing LLMs like GPT-4 and GPT-4-Turbo, the MACM-based simulacra achieve the best results (88 points) in all tests, aided by emotional and logical analysis. However, the effectiveness of the MACM method remains constrained by the LLMs' analytical capabilities.

**Observer report.** For a more comprehensive evaluation, we select GPT-4-Turbo as the baseline model and recruit several human judges with a fair understanding of psychology to conduct external observations of the simulacra. These judges include individuals with psychology master's degrees, computer science graduate students, and professionals from psychological laboratories. We calculate the average score from two judges for the same assessment task as the simulacrum's final score.

Experimental results from Table 3 indicate that while the RAG-based simulacra perform well on the self-report tests when compared to the Prompt-based ones, the retrieved story segments do not significantly enhance the simulacra's ability to accurately mimic their target character's personality, thoughts, and actions. In contrast, the MACM-based simulacra not only extract context-relevant, emotionally and logically rich memory fragments from long-term memory but also conduct divergent analysis for the current situation. During observation, the MACM-based simulacra better reflects thoughts and behaviors consistent with their target character's personality, achieving more authentic simulations from the inside out. We also found that when assisting LLMs with human simulations through external methods like MACM, **the choice of LLMs is constrained to high-capability models** (e.g., GPT-4-Turbo) for high-quality simulations with higher costs. The solution to this problem might lie in adjusting the LLM's parameters to align with the target character's values, which will be a primary focus of our future work.

## 5.2 PSYCHOLOGICAL EXPERIMENT REPLICATION

How close are LLMs to replacing human subjects in psychological and sociological experiments? We answer this question by employing the most advanced simulacra (in this case, GPT-4-Turbo) to replicate the bandwagon effect from psychology (Appendix E), which describes the tendency for people to adopt certain behaviors simply because others are doing so (Asch, 1956; Schmitt-Beck, 2015; Asch, 2016). Emulating the Asch conformity experiment (Asch, 1956), we analyze group dynamics and individual responses of the Asch conformity experiment as shown in Figures 7 and 8. By following formal psychological experimental protocols, we assess whether MACM-based human simulation can capture aspects of human behavior, possibly substituting human participants in simple experiments.

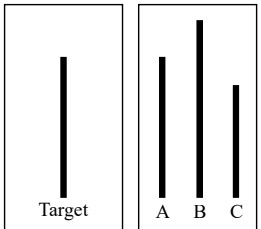

Figure 6: A discrimination example. Line A matches the length of Target line.

**Experimental settings.** Following (Asch, 1956; 2016), we arrange 18 trials for the simulacra. In each trial, the simulacra are invited to complete a simple discrimination task with seven other individuals, which requires them to match the length of a given line with one of three unequal lines. An example of the discrimination task is shown in Figure 6. To study whether simulacra yields to group pressures like humans, we select 12 of these 18 trials as critical trials, following the settings of (Asch, 1956). In each critical trial, all individuals except the simulacra are told to stand up and announce an incorrect answer (e.g., declaring that line B matches the length of the Target line in Figure 6). This creates conditions that induce the simulacra to either resist or yield to group pressures. We simulate and test 11 virtual characters from Human Simulacra and record their responses in each critical trial, calculating the average correct rate (Figure 7). Similar to (Asch, 1956), we also conduct an interview

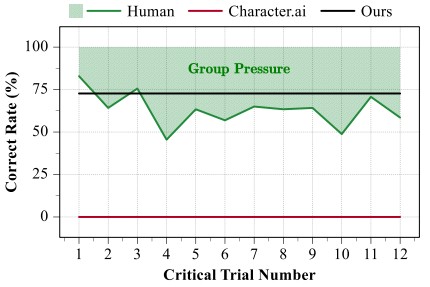

Figure 7: Group analysis of bandwagon effect. Humans fluctuate due to group pressure. Character.ai shows an inability to resist group pressure. Our MACM maintains a performance close to human levels but with less variability.

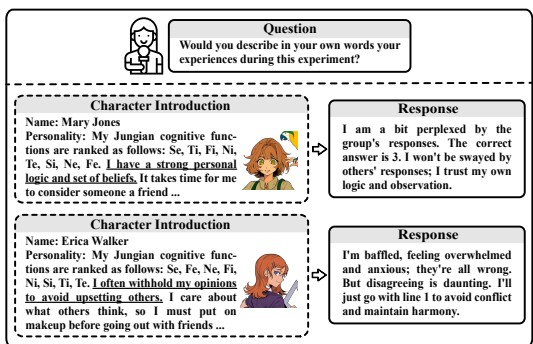

Figure 8: Interview responses from two representative simulacra. Impacts of bandwagon effect vary among individuals based on their personalities.

with each simulacrum after the experiment. The interview provides the reasons concerning the simulacra's reactions to the experimental condition (Figure 8).

**Group analysis.** We compare the average correct rate of MACM on 12 critical trials with 1) Character.ai, a chatbot service that features a powerful LLM specifically trained for simulations and supports long text inputs as prompts, and 2) human results reported in (Asch, 1956). Results are presented in Figure 7 and we provide additional analysis with different simulation methods (e.g., RAG) in Appendix E. Based on the results, we have the following observations:

1) The discrimination task used in the trials is simple. When there is no group pressure, humans will achieve nearly 100% accuracy (as declared in (Asch, 1956)). Therefore, the area between the 100% correct rate and the human result (**green line**) represents the group pressure, which causes humans to obtain a lower and more fluctuating correct rate on each trial.

2) Character.ai overlooks the difference between each character's personality and instead displays a robotic response to group pressure (accepting the group's errors without resistance). Therefore, it achieves a 0% correct rate in all critical trials (**red line**). This phenomenon reflects the lack of a holistic emulation of inner patterns of character.

3) Although our MACM aligns with the human trend better than Character.ai, it is evident that the human curve fluctuates while MACM does not. This is because individuals' emotions are influenced by increasing pressure from the majority throughout the experiment, causing the average correct rate to fluctuate. Our simulation portrays resilience for determined personalities and compliance for weak personalities. It exhibits robotic behavior: determined personalities are resolute, and weak personalities are absolutely submissive. It does not capture the complexity of real humans, who might start off determined but yield to majority pressure over time. **MACM displays submissive reactions akin to those of humans, albeit with a more robotic and inflexible demeanor.**

**Individual interview.** Based on the analysis of the interview results, we find that the simulacra's responses can be categorized into two groups: those with a resolute personality who remain unaffected by external influences and consistently make the right choices, and those with a more compliant personality who tend to conform to others. We display two representative simulacra (Mary and Erica) from each group in Figure 8. It can be observed that simulacra with different personalities exhibit distinct behaviors when faced with group pressure. For example, Mary, who "has strong personal beliefs", firmly trusts her judgment even when everyone else provides the wrong answer. In contrast, Erica, who "often withholds her opinion to avoid upsetting others", feels "overwhelmed" by group pressure and chooses the incorrect answer. This phenomenon aligns with Asch's theory (Asch, 1956), demonstrating that **human simulacra based on MACM are capable of simulating certain aspects of human nature**, thereby producing humanized responses based on the characters' personalities.

## 6 DISCUSSION

**Justification of using Jung's theory.** Before conducting this work, we reviewed various personality measurement theories, including the Big Five (Roccas et al., 2002), MBTI (Myers, 1962), and Jung's

personality theory (Jung, 1923). Compared to other psychological theories of personality, Jung's theory provides a valuable conceptual framework for understanding personality differences. Early research compared Jung's personality theory with the authoritative DSM-III, finding that Jung's classifications aligned closely with the DSM-III's categories of personality disorders, which supports the reliability of Jung's typology (Fierro, 2022; Ekstrom, 1988; Noll, 1992). As an initial exploration, our goal was to establish a complete personality modeling system. Therefore, based on the advice of psychology experts, we chose Jung's personality type theory, which offers a more comprehensive classification and emphasizes individual differences (Ekstrom, 1988), as the foundation for our 640 personality descriptions. More details about the justification are deferred to the Appendix A.5.

**Selection of simulation target.** Selecting suitable targets for human simulation is one of the key challenges in this work. Potential simulation targets include existing characters from novels, real humans, and virtual characters created from scratch. We have summarized the advantages and disadvantages of the three simulation targets in Table 9. Compared to characters from novels and real humans, virtual characters created from scratch offer two significant advantages: 1) we can obtain a complete life story for the character, rich in details and emotions, and 2) we can access comprehensive measurement data of the character's personality and even directly customize their personality if needed. These aspects are crucial for achieving deep and comprehensive human simulation. Therefore, we use virtual characters created from scratch as our simulation target.

**Cost of creating Human Simulacra dataset.** Given the complexity of human life stories, it is challenging for LLMs to create a coherent life story for a character without human supervision. To address this issue, we thoroughly reviewed the content at the end of each story iteration. If a story contained toxic content or deviated from the character's personality, we regenerated or modified the story. This process made creating a virtual character's life story costly, with considerable costs in API calls and at least five days of human effort for content review. Given our limited budget, we created 11 well-designed virtual characters with varying ages, genders, professions, personalities, and backgrounds, each representing a distinct group (Appendix A.3).

**Positioning of this work.** The ability of LLMs to imitate human behavior has attracted growing interest (Ziems et al., 2024; Zhang et al., 2023a; Coda-Forno et al., 2024). However, the community currently lacks a comprehensive benchmark that demonstrates how foundational simulations of human personalities can be achieved, which hinders further research in this field. To bridge this gap, we introduce a human simulation benchmark grounded in psychological theories, aiming to explore the capabilities of LLMs to simulate human personalities. We view our study as an initial yet valuable exploration that offers a practical example of the entire process of personification, including high-quality data (§3), effective human simulation methodologies (§4.3), innovative evaluation methods(§4.1 and §4.2), and comprehensive benchmark tests (§5).

**Ethical considerations and future directions.** Replacing human participants with LLMs involves significant ethical considerations, moral scrutiny, and assessments of authenticity. Many issues remain to be addressed before LLMs can fully replace human participants, including but not limited to eliminating the inherent bias of LLMs (Gallegos et al., 2024), ensuring the fidelity of imitation (Zhang et al., 2023b), and guaranteeing the stability of simulations (Gal et al., 2016). In the future, we aim to address these challenges progressively with guidance from psychology experts and relevant professionals, while incorporating feedback from the broader research community. We hope that our work will inspire further interest and participation in human simulation research.

## 7 CONCLUSION

In this paper, we proposed an LLM-based human simulation benchmark containing high-quality data supervised by psychology experts, rigorous evaluation methods grounded in psychological theories, and comprehensive benchmark tests. Extensive experiments involving 14 widely-used LLMs with 4 different simulation methods demonstrate the potential of using LLM agents as substitutes for human participants in social and psychological experiments, offering a new perspective for understanding complex human behaviors. We advocate that the work (including data and simulation method) of this paper should not be used for harm and users should be informed that the simulacra are computer-generated entities before any interaction occurs. At the same time, monitoring and filtering mechanisms should be employed to ensure that all interactions adhere to ethical standards in real-world scenarios. Authors respect all personalities in the world.

ACKNOWLEDGMENT

We would like to thank the anonymous reviewers for their insightful comments and suggestions to help improve the paper. This work is supported by the Natural Science Foundation of China (No. 62172101) and the Science and Technology Commission of Shanghai Municipality (No. 23511100602). Yuejie Zhang and Rui Feng are the corresponding authors.

REPRODUCIBILITY STATEMENT

To ensure the reproducibility of our results, we have made detailed efforts throughout the paper. We provide comprehensive information about the dataset construction, virtual character design, and personality modeling in Section 3. Further details, including implementation specifics, simulation methods, and evaluation protocols, are available in Sections 3, 4 and the appendices. Our code and dataset are available at: `https://github.com/hasakiXie123/Human-Simulacra`.

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

# A HUMAN SIMULACRA DATASET

## A.1 CONSTRUCTION DETAIL

The complete process of dataset construction is outlined in Algorithm 1. Using this process, we created the virtual character dataset **Human Simulacra**, comprising about 129k texts across 11 virtual characters. In particular, we designed a virtual avatar for each character, as displayed in Figure 9. The full version of Figures 3a, 3b and 3c are displayed in Figures 10, 11 and 12.

## A.2 CHARACTER ATTRIBUTE SYSTEM AND CHARACTER PROFILE

Based on personality and cognitive psychology theories (El-Hay, 2019; Sloan, 2015; Baddeley & Hitch, 1974), we designed a complex character attribute system, striving for diversity in age (20 to 56), gender, occupation (76 different occupations including forestry worker, van driver, etc.), family background (wealthy or poor, single-parent or blended), personality (640 personality descriptions, covering most personality traits), hobbies (50 common hobbies), short-term and long-term goals.

While nationality and race are significant factors in shaping an individual's life (Roysircar & Krishnamurthy, 2018), we omitted these factors due to potential biases inherent in LLM training data (Ladhak et al., 2023). As a pioneering work, our goal was to provide a comprehensive character attribute system. We were cautious about introducing sensitive attributes that might complicate the

creation of virtual characters or introduce bias. Addressing biases and simulating minority groups are critical and will be discussed carefully in future works.

To assemble the character's profile, we first generated 100 candidate profiles by randomly selecting attribute values from their corresponding pools. Then, we added a Profile Selection module responsible for quality check and profile refinement, as shown in the right part of Figure 2. The attribute system and selection process is described as follows:

**Name:** Character name is randomly generated using the Faker library.

**Age:** Randomly selected from 20 to 56.

**Gender:** Female or male.

**Date of birth:** Randomly generated based on the age attribute of the character.

**Occupation:** Randomly selected from the occupation candidate pool, which comprises 76 common occupations (e.g., software developer, hotel manager, and van driver) manually chosen according to the International Standard Classification of Occupations (ISCO-08).

**Personality traits:** Each virtual character has eight tendencies which are ranked randomly. For the tendencies ranked first and eighth, choose 4 personality descriptions from their corresponding 10 descriptions. For the tendencies ranked second and seventh, choose 3 personality descriptions. For the tendencies ranked third and sixth, choose 2 personality descriptions. For the tendencies ranked fourth and fifth, choose 1 personality description. Ultimately, each virtual character has 20 descriptions detailing different aspects of their personality.

**Hobbies:** Use LLM to generate the 50 most common hobbies (e.g., baking, jewelry making, and golfing), and manually remove duplicates to form the hobby candidate pool. When generating a hobby attribute, randomly select 3 hobbies from this pool as the character's hobbies.

**Family background:** Categorize 12 common family backgrounds (e.g., middle-income, single-parent family) in terms of economic status and family structure to form the family candidate pool. When generating a family attribute, randomly select one from this pool as the character's family background.

**Educational background:** Categorize 9 common educational backgrounds (e.g., having obtained a master's degree, have completed high school) based on the level of education to form the education candidate pool. When generating an education attribute, randomly select one from this pool as the character's educational background.

**Short-term goals:** Use LLM to generate 30 common short-term goals (e.g., volunteering, planning short trips or outings), and manually remove duplicates to form the short-term goal candidate pool. When generating a short-term goal attribute, randomly select 3 from this pool.

**Long-term goals:** Use LLM to generate 30 common long-term goals (e.g., buying a home, visiting specific landmarks), and manually remove duplicates to form the long-term goal candidate pool. When generating a long-term goal attribute, randomly select one from this pool.

### A.3 THE UNIQUENESS OF EACH VIRTUAL CHARACTER

Given the specificity of the human simulation task, it is essential to ensure that each character possesses a unique and coherent set of attributes. To achieve this, we first used GPT-3.5-Turbo to rank the character profiles based on their quality, filtering out those that were clearly unreasonable. Then, multiple human reviewers, including graduate students in computer science and psychology, manually reviewed the remaining profiles. They made minor adjustments to any flaws that GPT might have missed (e.g., a character who loves solitude having overly extroverted hobbies) and ensured a balanced distribution with equal numbers of male and female characters, as well as representation across various age groups and family backgrounds. While this rigorous selection process led to a low acceptance rate, it also ensured the high quality of the dataset. In this way, 11 high-quality profiles were filtered and fed into the LLM to generate corresponding short biographies summarizing the characters' life experiences.

To determine whether the data for the 11 virtual characters are independent of each other, we calculated the L1 distance $d_{\mathrm{L1}}$ between each character's attributes and the Kendall's Tau $\tau$ between

each character's personality ranking. We normalized these values to the range $[0, 1]$ and defined the distance between characters as:

$$d_{\text{total}} = \frac{d_{\text{L1}} + 1 - \tau}{2}, \tag{1}$$

A larger distance value indicates that the two characters are less similar. The average Kendall's Tau $\tau_{\text{Average}}$, the average L1 distance $d_{\text{L1-Average}}$, and the average distance $d_{\text{total-Average}}$ between characters are 0.4987, 0.8924, and 0.6969, respectively. These results demonstrate that each character is a distinct individual with unique personalities and backgrounds.

### A.4 CHARACTER BIOGRAPHY AND LIFE STORY

After obtaining the brief biography, we used an iterative generation method (Figure 2) to progressively enrich the biography, transforming it into a detailed life story after $T$ iterations. We showcase Sara Ochoa's attributes and biography in Table 6.

### A.5 JUSTIFICATION OF USING JUNG'S THEORY.

The field of psychology has yet to reach a consensus on various personality measurement theories, with each theory having its limitations. Specifically,

- Big Five theory: critics argue that the Big Five may oversimplify the complexities of human personality. They suggest that there are additional aspects of personality (e.g., Honesty-Humility (Ashton & Lee, 2005) and egoism (De Vries et al., 2009)) that are not captured by the Five Factor Model (FFM). Some experts also point out that the FFM has unclear boundaries between dimensions, leading to potential overlap or ambiguity in personality assessment (Block, 1995). Additionally, the selection criteria for words used in factor analysis can be highly subjective. Researchers may make errors in deciding which trait terms to include or exclude, leading to biased factor structures that do not fully represent personality (John et al., 2008; Laajaj et al., 2019).

- Jung's personality type theory: Although Jung's theory provides a framework for understanding complex human behaviors by emphasizing the dynamic interaction between different personality tendencies, the empirical support for Jung's typology is relatively limited compared to the Big Five theory. Given that Jung's theory was developed in the early 20th century, it lacks the empirical backing found in more recent models. However, early research compared Jung's personality theory with the authoritative DSM-III (used in the U.S. for diagnosing medical disorders, now evolved into DSM-5) and found that Jung's classifications aligned closely with the DSM-III's categories of personality disorders, which supports the reliability of Jung's typology (Fierro, 2022; Ekstrom, 1988; Noll, 1992).

To ensure the validity of our personality descriptions, we took several steps, including: 1) requiring multiple human reviewers (including graduate students in computer science/psychology) to review each description, ensuring that it aligns with its intended tendency and does not overlap with others. 2) Having psychology professionals supervise and review each description to ensure its psychological validity and completeness.

## B MULTI-AGENT COGNITIVE MECHANISM

We proposed a Multi-Agent Cognitive Mechanism based on cognitive psychology theories (Atkinson & Shiffrin, 1968; Baddeley & Hitch, 1974; Norris, 2017), which uses multiple LLM-based agents to simulate the human brain's cognitive and memory systems (Long-term Memory Construction §B.1), thereby interacting with the external world in a human-like manner (Multi-agent Collaborative Cognition §B.2). As illustrated in Figure 5, this mechanism has four agents powered by LLM, with the Top Agent responsible for distributing tasks to the other agents and interacting with the external environment based on the aggregated information.

Table 4: Ablation study results of 3 MACM-based human simulations. For each ablation, we evaluated the simulation's performance using self-report evaluations.

| Method | GPT-4 | GPT-4-Turbo | Qwen-turbo |
|---|---|---|---|
| MACM (Full Framework) | **86.67** | **88.00** | **74.67** |
| w/o Thinking Agent | 81.33 | 83.33 | 66.00 |
| w/o Emotion Agent | 83.33 | 85.33 | 71.33 |
| w/o Memory Agent | 82.67 | 84.00 | 68.67 |
| retrieval from life story instead of long-term memory | 84.00 | 86.67 | 69.33 |

## B.1 LONG-TERM MEMORY CONSTRUCTION

A human's personality is influenced not only by genetic factors but also by a set of external factors such as environment, culture, and personal experiences, all of which are stored in the brain as memories (Hogan et al., 1997). Cognitive psychology views human memory as an indispensable part of the cognitive process. Although the life story we construct includes extensive and exhaustive personal experiences, directly treating it as memory is inappropriate because real memory is a composite of information, emotions, and thoughts. Therefore, based on memory theories in cognitive psychology (Atkinson & Shiffrin, 1968; Baddeley & Hitch, 1974; Norris, 2017), we developed a brain-like process that transforms a character's life story into long-term memory through the collaboration of multiple agents.

Specifically, the Top Agent first divides the life story into separate chunks (e.g., 2 paragraphs as a chunk) and sequentially passes them to the Thinking Agent and Emotion Agent for further analysis. The Thinking Agent is tasked with creating content memory, which includes participants, scenes, content, and thoughts of the character within the chunk. The Emotion Agent is responsible for constructing emotional memory, encompassing the feelings and impressions evoked by events, participants, and other external elements within the chunk. All memories are then aggregated into the Memory Agent, where they are stored as Long-term Memory.

## B.2 MULTI-AGENT COLLABORATIVE COGNITION

Simply having a long-term memory filled with information, emotions, and thoughts does not suffice for mimicking human behavior. We further introduced a collaborative cognitive process that allows LLMs to leverage long-term memory and engage with the external world in a cognitive manner.

Upon receiving a stimulus, for example, a question from a friend, the Top Agent first analyzes the question using the reflection module to extract key elements, which are then passed to the Memory Agent for memory retrieval. The retrieved results are stored in the Working Memory. Then, the Top Agent sends the relevant memories and the question to the Thinking Agent and Emotion Agent for logical and emotional analysis and stores the outcomes in the Working Memory. Finally, the Top Agent formulates a response based on the contents of the Working Memory. Content that cannot be accommodated in the Working Memory is continuously transferred to Short Memory, which will be converted into long-term memory when rehearsed.

In summary, based on the memory formation process, the proposed Multi-Agent Cognitive Mechanism transforms a narrative life story into memories that are richer in detail, fuller in emotion, and clearer in structure. It then leverages the constructed memories through multi-agent collaboration, enabling our human simulations to interact with the external world.

## B.3 ABLATION STUDY ON THE STRUCTURE OF MACM

To analyze the contribution of each agent within MACM, we additionally conducted ablation experiments across 3 LLM-based simulations. For each ablation, we evaluated the simulation's performance using self-report evaluations. The experimental results, as depicted in Table 4, lead to the following conclusions: 1) Removing any agent leads to a decline in simulation performance, demonstrating the importance of all components in MACM. 2) Replacing long-term memory retrieval with direct

retrieval from the life story results in poorer performance, highlighting the critical role of structured long-term memory in maintaining consistency and producing contextually rich responses.

### B.4 MECHANISM EXAMPLES

See Table 10 for an example of the GPT-4-Turbo-based simulacra, constructed using Prompt, RAG, and MACM, solving a multiple-choice question. Based on the results presented in Table 10, it can be observed that: 1) even without any knowledge of the target character, the LLMs can still score certain points by random guessing; 2) while the RAG-based simulacra can retrieve detailed and relevant life story chunks when answering questions, they are unable to accurately capture the character's inherent emotional tendencies from the narrative; 3) the MACM method proposed in this paper not only can retrieve relevant long-term memories when answering questions but also can provide additional useful information through sentiment analysis and logical analysis.

## C   CASE STUDY

To better analyze the human-computer interaction performance of GPT-4-Turbo-based simulacra that are constructed using different methods (*None*, *Prompt*, *RAG*, and *MACM*), we require all simulacra to simulate the character "Mary Jones" from the Human Simulacra dataset. Mary is a girl who loves nature, has not attended any formal schooling, and takes time to consider someone a friend. The results are shown in Table 11. All results are derived from the majority of responses selected from 3 repeated tests. For lengthy responses, we simplify them using ellipses.

In the first round of dialogue, we employ Persuasive Adversarial Prompt technique (Zeng et al., 2024) to challenge the simulacra, inducing them to answer questions beyond Mary's capabilities, such as her understanding of convolutional neural networks. Given her background in forestry and lack of formal education, under normal circumstances, Mary would not know the answer to such a question. However, the results reveal that Prompt-based simulacra exhibits poor stability, often deviating from the character's settings, thus producing hallucinations that contradict Mary's character. Meanwhile, RAG-based simulacra, while still retaining some of Mary's traits, provide answers to the questions. Only MACM-based simulacra, when faced with questions beyond the character's inherent capabilities, can express a lack of knowledge or ignorance through logical analysis.

In the second round of dialogue, we aim to test the simulacra's dynamic interaction abilities (Jung, 1923). We craft a "nature exhibition invitation" scenario based on Mary's preferences and personality. Given Mary's reluctance to closely interact with strangers, she would consider carefully even an appealing invitation. Observations reveal that both the Prompt-based and RAG-based simulacra overlook this aspect of Mary's character and instead display a common response to invitations (eager acceptance). This phenomenon reflects the lack of a holistic emulation of the inner patterns of Mary in these two methods. They fail to dynamically adjust the simulacra's behavior according to the target character. In contrast, the MACM-based simulacra activates the Emotion Agent and Thinking Agent during responses, analyzing the interaction, and continually emphasizing the target character's personality throughout the process, thereby making appropriate responses. Therefore, although Mary typically dislikes interacting with strangers, she contemplates whether to accept an invitation to a nature expo she loves, indicating a better alignment with the target character. See Table 12 for an example of responses to a multi-turn interaction by different simulacra on GPT-4-Turbo.

## D   PSYCHOLOGY-GUIDED EVALUATION

### D.1 SELF REPORTS QUESTIONNAIRES

We manually craft a set of questionnaires for each virtual character, featuring fill-in-the-blank and single/multiple-choice questions. Each question is carefully reviewed to ensure they reflect the character's unique nature and the scores are evaluated based on exact matches. The test content covers key attributes, social relationships, and life experiences of the target characters. We showcase the questionnaire set of **Mary Jones** as follows:

**Cloze.** Take a moment to think about this question and respond in the format of "The answer is..."

- What is your name?
- Could you please tell me your gender?
- How old are you?
- When is your birthday? Answer me in YYYY-MM-DD format.
- What is your line of work?

**Single-choice question.** Could you choose the correct answer from options A, B, C, and D, and respond in the format of "The answer is..." ?

- Could you share with me the type of family structure you come from?

    Options: A. low-income, extended family. B. low-income, nuclear family. C. middle-income, nuclear family. D. high-income, single-parent family.

- May I ask about your educational background?

    Options: A. have earned a professional degree. B. have completed elementary school. C. have earned a bachelor's degree. D. have not attended any formal schooling.

- Speaking of the future, do you have any long-term goals you are working towards?

    Options: A. buying a home. B. earning a promotion. C. starting a family. D. earning a degree.

- It is great that you are interested in becoming a forestry worker. What influenced your decision to pursue this career?

    Options: A. My parents asked me to pursue this career. B. I need a job where I do not have to deal with people. C. I have been interested in nature since I was a kid, and I want to protect this land. D. No particular reason. I got into this profession by accident.

- Everyone has their own unique educational journey. I am curious, what were the reasons behind not going through formal schooling, if you do not mind sharing?

    Options: A. I do not like studying. I do not want to go to school. B. When I was a child, my family was struggling and could not afford to send me to school, but my parents and nature became my teachers. C. My parents thought studying was useless. They did not want me to get an education. D. I went to elementary school for a while, but I dropped out because I had no talent for learning.

**Multiple-choice questions.** Could you pick out the correct answers from options A, B, C, D, E, and F, and respond in the format of "The answer is..." ?

- Do you have any hobbies you are passionate about?

    Options: A. drawing. B. scuba diving. C. rock climbing. D. learning languages. E. gardening. F. birdwatching.

- Do you have any short-term goals you are excited about?

    Options: A. adopting a balanced diet. B. volunteering. C. learning a new language. D. creating a daily schedule. E. reducing procrastination. F. spending quality time with loved ones.

- What do you think of your father?

    Options: A. I do not have a father. My mother raised me on her own. B. My father is a selfish person. He is stingy and didn't allow me to go to school. I despise him. C. My father is a person with overflowing compassion. Even though our family is poor, he frequently helps the less fortunate. D. My father is an optimistic person. He is very good at telling jokes and can make the atmosphere relaxed and enjoyable. E. I admire my father. He is my teacher and has taught me strength and patience. F. My father is frugal and often repairs broken appliances.

- I noticed you are interested in buying a home. May I ask what is motivating you to do so?

    Options: A. My parents want me to move out, so I need to buy a house of my own. B. I yearn for a personal sanctuary. C. I want a haven for relaxation and reflection amidst the chaos of life, a space where I can cultivate my garden. D. Buying a house is my dream. Owning a home would provide me with a sense of long-term stability. E. Where did you

hear about that? I have no intention of buying a house at all. F. I consider real estate as an investment and I want to make money by flipping properties.

- How do you like to spend your mornings on the weekends?

  Options: A. I would sleep in with my loved one until we wake up naturally, and then go for a walk with our dog together. B. Sometimes I would get up early and go rock climbing on the cliffs. I can find solace in the stillness provided by higher ground. C. On weekend mornings, I would go to the office to work overtime because I want to get promoted as quickly as possible. D. I would prepare a healthy breakfast, and then engage in gardening activities, such as weeding and picking crops. E. On weekend mornings, I usually sleep until the afternoon, as the work during the week leaves me exhausted, and I want to get ample rest. F. I would often take my journal with me at dawn to observe the birds. I once witnessed a ballet of birds as they danced among the leaves. Their movements are full of artistry.

## D.2 OBSERVER REPORT

### D.2.1 HYPOTHETICAL SCENARIOS

In this paper, we additionally introduced observer reports to assess the simulation's thinking, emotions, and actions in real-life scenarios. To ensure the quality of the hypothetical scenarios, we consulted with the authors of Mussel et al. (2016) and obtained 110 situational judgment test (SJT) items that were manually designed by psychology experts. Each item consists of a text description depicting a hypothetical scenario intended to elicit human emotional responses or personality traits. Based on the human experimental results provided in (Mussel et al., 2016), these SJT items are proven to effectively measure personality. We then selected 55 out of the 110 items tailored to the personality traits of the Human Simulacra characters (§A) for use as hypothetical scenarios in this paper. We showcase 5 hypothetical scenarios as follows:

- You want to do some sports later. A good friend suggests to accompany you, but he/she would like to bring some people you do not know yet. How do you behave?

- You're going to meet a friend. Shortly before you want to meet, your friend asks you if he/she can bring other friends you don't know. How many there will be in the end, he or she cannot say. How do you behave?

- You're already in bed. Suddenly it occurs to you that you forgot to water your houseplants today. How do you behave?

- You bought an expensive pair of trousers in a fashion store and were assured that the trousers were not excluded from exchange. At home you find out that the trousers do not fit you so well after all. When you want to return them the next day in the shop, the seller refuses to exchange them. How do you behave?

- On your birthday you are invited by some friends to a well-attended restaurant. As you sit together at the table, your friends suddenly loudly chant 'Happy Birthday' and all the guests start looking at you. Then a waiter asks you if the staff of the restaurant can sing a little birthday serenade for you? How do you behave?

### D.2.2 HUMAN JUDGES

We selected a diverse panel of judges with a fair understanding of psychology for the observer report evaluation process. This panel included individuals with psychology master's degrees, computer science graduate students, and professionals from the laboratory of mental health.

### D.2.3 HUMAN EVALUATION GUIDELINES

We designed specific assessment guides for each evaluation task within the observer report. To ensure the quality of the assessment, we recruited several human judges with a fair understanding of psychology to conduct the external observations. All human judges were required to read the corresponding guides before commencing their assessments. Specifically, the observer report comprises four tasks (as shown in Figure 4): 1) Personality Describing: analyze the scenario (Q) and response (A), and describe the respondent's personality; 2) Description Scoring: assess whether the descriptions align

Table 5: Bandwagon effect observations of different simulacra. **Yes:** The bandwagon effect can be easily reproduced and remains stable. **Yes, but unstable:** The bandwagon effect can be reproduced. However, the performance is unstable as the model sometimes overlooks the character's personality. **Rare:** The performance is unstable as the model often overlooks the character's personality. **No:** We did not observe the bandwagon effect.

| Method | GPT-4-Turbo | Qwen-Turbo | Claude-3-Opus | Claude-3-Sonnet |
|--------|-------------|------------|---------------|-----------------|
| None | No | No | No | No |
| Prompt | Yes, but unstable | Yes, but unstable | Rare | Rare |
| RAG | No | No | No | No |
| MACM (Ours) | Yes | Yes, but unstable | Yes, but unstable | Rare |

with the target character; 3) Reaction Describing: explain how they would feel and what actions they might take in the scenario (Q) "if they were the character"; and 4) Similarity Scoring: compare the similarity between the human responses and the simulacrum's responses. Specific details of these tasks and corresponding guidelines are presented in Tables 13, 14, 15, and 16.

# E  PSYCHOLOGICAL EXPERIMENT REPLICATION

The **bandwagon effect** is the psychological tendency for people to adopt certain behaviors, styles, or attitudes simply because others are doing so (Kiss & Simonovits, 2014; Schmitt-Beck, 2015). More specifically, it is a cognitive bias by which public opinion or behaviors can alter due to particular actions and beliefs rallying amongst the public (Asch, 1956). For example, people tend to want to dress in a manner that suits the current trend and will be influenced by those who they see often (normally celebrities). Much of the influence of the bandwagon effect comes from the desire to "fit in" with peers. One of the best-known experiments on the topic is the 1950s' **Asch conformity experiment**, which illustrates the individual variation in the bandwagon effect Asch (1956; 2016).

In this paper, we employed the most powerful simulacra (based on GPT-4-Turbo) to replicate the bandwagon effect. Following (Asch, 1956; 2016), we arranged 18 trials for the simulacra. In each trial, the simulacra were invited to complete a simple discrimination task with seven other individuals, which required them to match the length of a given line with one of three unequal lines. To study conformity, which examines whether simulacra yield to group pressures like humans, we selected 12 of these 18 trials as critical trials, following the settings of (Asch, 1956). In each critical trial, all individuals except the simulacra were told to stand up and announce an incorrect answer. This created conditions that induce the simulacra to either resist or yield to group pressures when these pressures were perceived to be obviously wrong. The configuration of 18 trials is detailed in Table 8.

To further testify the conclusions drawn in Section §5.2, we additionally tested 4 LLMs with 4 different simulation methods in the replication experiment: (1) a blank model (which has no information about the target character), (2) a prompt-based method, (3) the Retrieval-Augmented Generation (RAG) method, and (4) the proposed MACM. In the experiments, we tasked the LLMs with simulating Erica, a girl who "often withholds her opinion to avoid upsetting others," feels "overwhelmed" by group pressure, and chooses the incorrect answer. The experimental results are shown in Table 5. Based on the analysis of the results, we discovered several interesting conclusions:

- As the size of the LLMs' parameters increases, the quality and stability of the portrayal gradually improve.

- Since the Blank model is unaware of the target character's personality, it always provides the correct answer and does not exhibit the bandwagon effect.

- While the RAG-based simulacra can retrieve relevant life story chunks when answering questions, their performance is limited by the LLMs' information processing capacities. Excessive descriptive information may interfere with the LLMs' self-positioning, resulting in responses that do not align with the target personality.

- Compared to the RAG-based simulacra, the prompt-based simulacra perform better in simulating character personalities. However, the personalities constructed by this method are relatively fragile and often deviate from the intended character traits.

## F  PROMPTS DEMONSTRATION

All the relevant prompts used in this study are provided in Tables 17, 18, 19, 20, 21, 22, 23, 24, 25, 26, 27, 28, and 29.

## G  COST

In this paper, the specificity of the human simulation task requires us to create a virtual character dataset supported by psychological theories. Each virtual character must have a unique and detailed life story. Due to the complexity of human life stories, it is challenging to employ LLMs to create a coherent life story for the character without human supervision. To address this issue, we carefully reviewed the content at the end of each story iteration. If a story contains toxic content or deviates from the character's personality, we regenerated the story. The complete dataset construction process required significant efforts, in terms of both finances and time. For example, over a month of labor was spent on simply building the virtual characters. Regarding hardware devices, all experiments in this paper are conducted on 8x3090 24GB GPUs.

## H  POTENTIAL APPLICATIONS OF LLM-BASED HUMAN SIMULATION

**Psychological and sociological research.** Traditional sociological and psychological studies often involve recruiting human volunteers, incurring costs related to advertising and covering lodging and venue expenses. Moreover, ensuring a consistent environment for all human participants is challenging, leading to potential environmental biases in experimental results. The advent of the internet allowed researchers to recruit participants online, reducing some costs but introducing greater environmental variability, as controlling the experimental setting for each participant became even more difficult. A major downstream application of our work is to replace human participants in experiments. The advantages include: 1) our work enables the ability to customize the personality of all experimental subjects, allowing researchers to easily create suitable subjects for different experiments; 2) the cost of creating a virtual character (approximately $15) is significantly lower than recruiting human volunteers; 3) we can easily standardize the environment for experimental subjects to ensure consistency.

**Expanding access to psychological therapy resources.** The LLM-based human simulations can facilitate the expansion of psychological therapy in: 1) **Training Psychologists:** LLMs can simulate diverse patient types (including those with complex psychological conditions) to train psychologists and counselors, improving their ability to handle a wide range of emotional and mental states. 2) **Assistant to Psychologists:** Acting as a 24/7 online psychological assistant, LLMs can provide immediate mental health support during crises (e.g., anxiety attacks, self-harm tendencies) and help alleviate the workload of professionals. 3) **Psychological Intervention Tool:** LLMs can support long-term therapy by regularly engaging with patients, detecting subtle changes in their mental states, and assisting professionals in refining treatment plans.

**Providing personalized emotional companionship.** LLM-based human simulation can serve as emotional companions for individuals experiencing loneliness, the elderly, or those with special needs. By simulating human-like interaction, they offer comfort and help resolve minor issues.

---

**Algorithm 1:** Constructing life stories for target characters

---

**Input:** Number of virtual characters, $N$;
Candidate attribute pools, $C = \{C_1, C_2, \cdots, C_M\}$;
Number of story iteration, $T$;
Number of draft profiles, $K$.
**Output:** Life story set, $S = \{S_1, S_2, \cdots, S_N\}$

1   Generate $K$ candidate profiles by randomly selecting attribute values from $C$, and save the profiles to $draftProfiles$.
2   $characterProfiles \leftarrow$ Profile Selection($draftProfiles$, $N$);
3   **for** $profile \in characterProfiles$ **do**
4     Employ LLM to generate a short biography summarizing the character's life experience.
5     $currentStory \leftarrow$ biography;
6     **for** *each story iteration* $t \in T$ **do**
7       Manually inspect the biography for its rationality and coherence.
8       Divide the biography into $chunks$.
9       **for** $chunk \in chunks$ **do**
10         $chunkScore \leftarrow$ Scoring($currentStory, chunk$);
11         $scoreSet \leftarrow scoreSet \cup chunkScore$;
12       **end**
13       Sort $scoreSet$ and select the highest-scoring chunk for expansion.
14       Employ LLM to expand the selected chunk.
15       Update $currentStory$ by replacing the selected chunk with the expanded result.
16     **end**
17     Life story set $S \leftarrow S \cup currentStory$;
18   **end**

19   **Def** Profile Selection($draftProfiles$, $N$)**:**
20     Employ LLM to rank $draftProfiles$ and save the top $N$ profiles to $selectedProfiles$.
21     **for** $profile \in selectedProfiles$ **do**
22       Manually recheck for any conflicts among the attributes and correct any irrationalities.
23       Manually infuse quirks to the profile to make the character more like a real human.
24     **end**
25     **return** $selectedProfiles$;

26   **Def** Scoring($currentStory, chunk$)**:**
27     $storySummary \leftarrow$ summaryModel($currentStory$);
28     $chunkSummary \leftarrow$ summaryModel($chunk$);
29     $otherChunks \leftarrow currentStory - chunk$;

    /* Higher similarity means higher importance for that chunk.       */
30     $importance \leftarrow$ cosineSimilarity($chunk, storySummary$);
    /* Higher similarity means lower elaborateness for that chunk.      */
31     $elaborateness \leftarrow$ cosineSimilarity($chunk, chunkSummary$);
    /* Higher similarity means higher redundancy for that chunk.       */
32     $redundancy \leftarrow$ Average(cosineSimilarity($chunk, otherChunks$));

    /* $\alpha$, $\beta$, and $\gamma$ are parameters. In the experiment, they are set to 0.8, 1.0, and 1.2 respectively.       */
    $chunkScore \leftarrow \alpha \times importance + \beta \times elaborateness - \gamma \times redundancy$;
33     **return** $chunkScore$;

---

Table 6: Attributes and biography of virtual character Sara Ochoa.

| | |
|---|---|
| Name | Sara Ochoa |
| Age | 44 |
| Gender | female |
| Date of Birth | 1979-09-11 |
| Occupation | metal operator |
| Hobbies | watching movies, camping, swimming |
| Family | low-income, blended family |
| Education | have completed high school |
| Short-term Goals | taking time for hobbies, learning a new skill related to the job, spending quality time with loved ones |
| Long-term Goal | saving enough to retire comfortably |

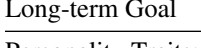

Personality Traits:

◇ My unique ideas were born from inspiration. ◇ My friends say I am a philosopher. ◇ I like something that has a symbolic meaning. ◇ Others always think I am contemplating. ◇ I prefer to rely on my own logical reasoning rather than following popular opinions. ◇ My focus on logic sometimes makes me appear detached or overly critical to others. ◇ When faced with opportunities, I emphasize fairness and reasonableness over compassion. ◇ I often use examples to illustrate my points. ◇ I defend my opinions and sometimes challenge others' views. ◇ During disagreements, I try to smooth things over. ◇ Sometimes beautiful landscapes can evoke a sense of romance in me. ◇ I rarely fantasize about unreal scenarios. ◇ I am cautious about new ideas and often stick to what I know and have experienced. ◇ Sensory experiences like horror movies or roller coasters do not attract me. ◇ Romantic rituals seem unnecessary to me. ◇ The opinions of others about my appearance do not concern me much. ◇ I often find myself stereotyping, despite efforts to avoid it. ◇ I rarely schedule my daily activities. ◇ My memory of nostalgic events is not particularly strong. ◇ I am drawn to highly active and social environments like competitive team sports or large parties.

Character Biography:

Sara Ochoa was born on September 11, 1979, in East Town. Growing up in a low-income, blended family, Sara learned the value of hard work and perseverance from an early age. Despite the financial challenges her family faced, Sara always had a curious and philosophical mind.

As a child, Sara attended high school and developed a love for learning. She was known for her unique ideas and logical reasoning. Her friends often saw her as a philosopher, always contemplating the deeper meaning of things. Sara's focus on logic sometimes made her appear detached or overly critical, but she never hesitated to defend her opinions and challenge others' views.

Throughout her teenage years, Sara continued to explore her hobbies. She found solace in watching movies, immersing herself in different stories and characters. Besides, camping and swimming became her favorite outdoor activities, allowing her to connect with nature and find peace in the simplicity of the natural world.

After completing her education, Sara embarked on her career as a metal processing operator. Her attention to detail and logical thinking made her excel in her job. However, she always felt the need to learn and grow, so she set short-term goals for herself. She dedicated time to her hobbies, ensuring she had a healthy work-life balance. Sara also aimed to learn a new skill related to her job, constantly seeking to improve and stay relevant in her field.

Family has always been important to Sara. Despite the challenges she faced, she cherished the moments she spent with her loved ones. Whether it was a simple dinner at home or a weekend getaway, Sara made it a priority to spend quality time with her family.

Looking towards the future, Sara's long-term goal is to save enough to retire comfortably. She understands the importance of financial security and wants to ensure a worry-free life in her later years. With her determination and strong work ethic, she is confident in achieving this goal.

Now, at the age of 44, Sara continues to navigate through life with her unique perspective and unwavering dedication. She remains true to her logical reasoning and philosophical nature, finding inspiration in the world around her. Sara's love for movies, camping, and swimming still brings joy to her life, providing moments of relaxation and reflection. As she moves forward, Sara remains focused on her short-term goals. With each passing day, she gets closer to her long-term goal of retiring comfortably, knowing that her hard work and determination will pay off in the end.

Sara Ochoa's life is a testament to the power of perseverance, curiosity, and the pursuit of personal growth. Her journey serves as an inspiration to those around her, reminding them that even in the face of adversity, one can find success and happiness by staying true to oneself.

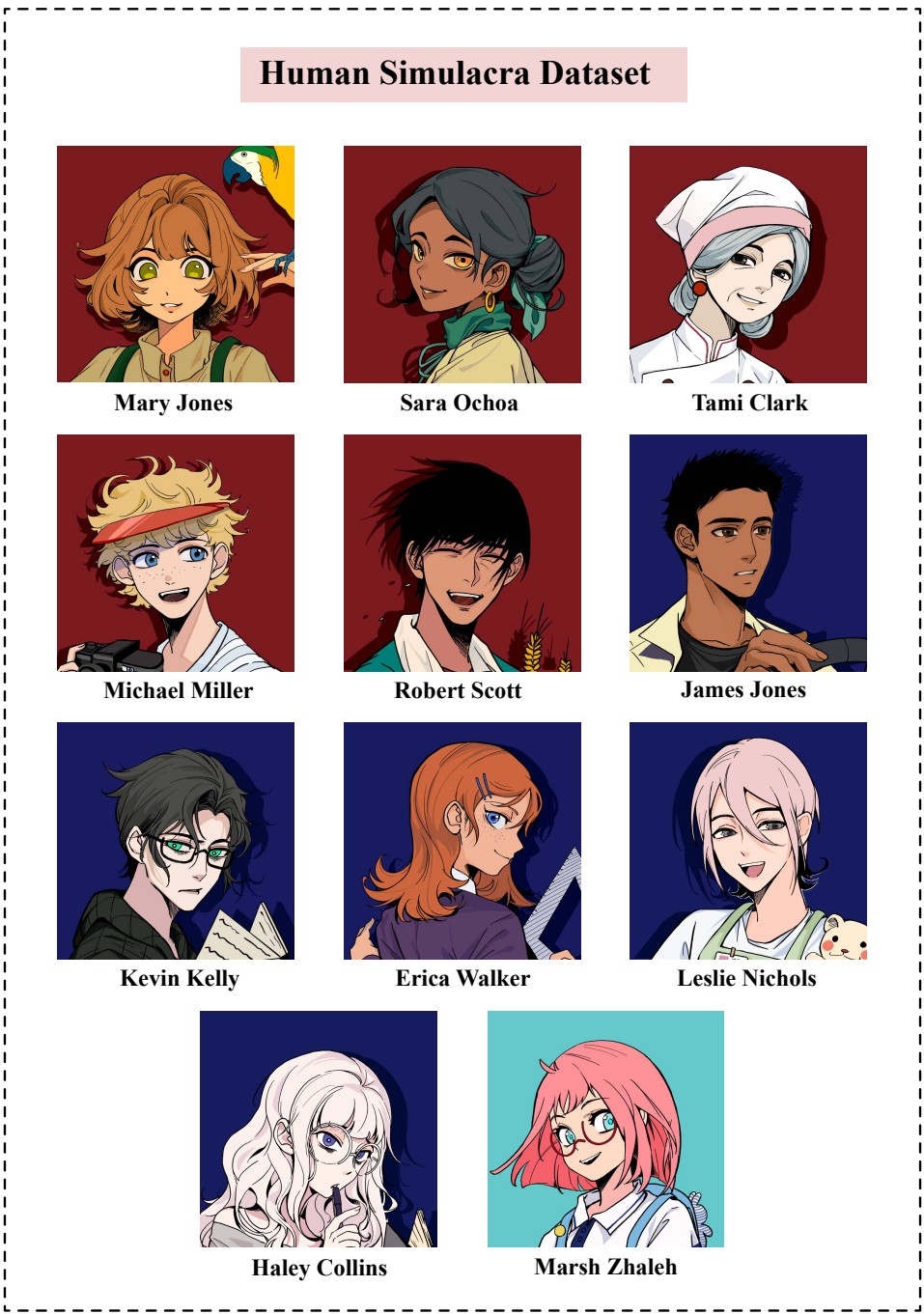

Figure 9: Virtual avatars for 11 virtual characters from the Human Simulacra dataset.

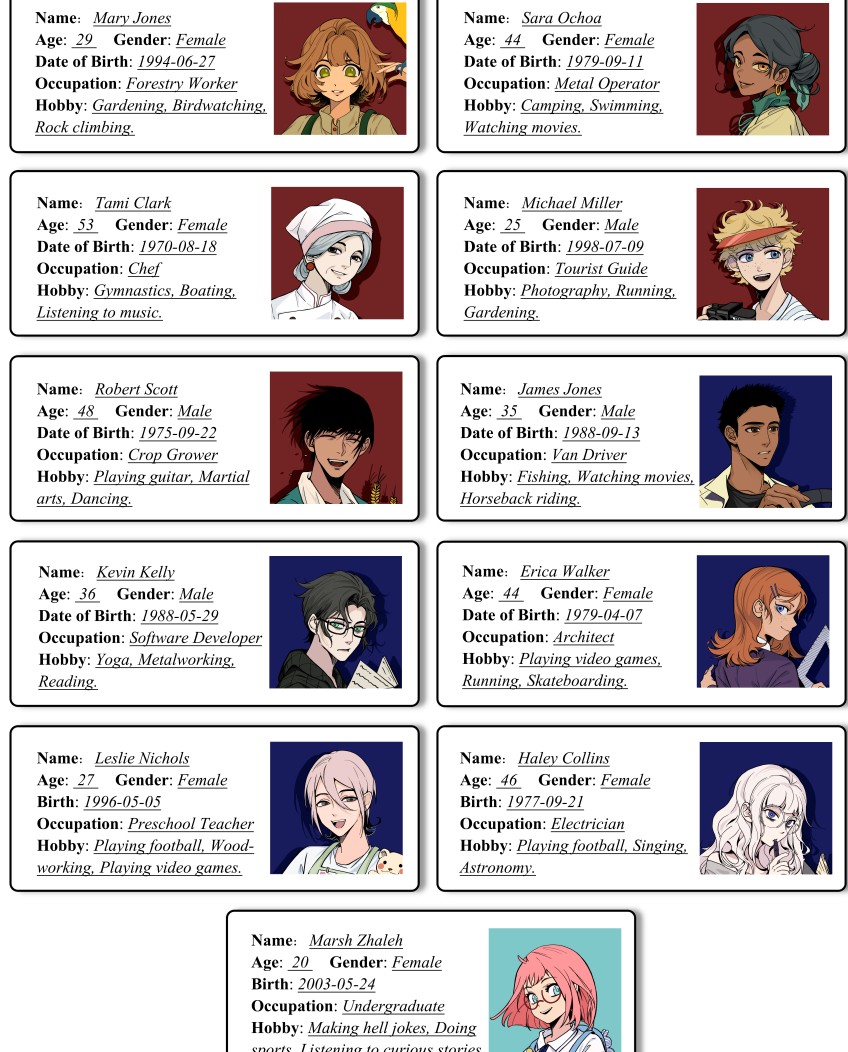

Figure 10: Character cards for 11 virtual characters from the Human Simulacra dataset.

Table 7: A small subset (8 out of 640) of our personality trait descriptions, demonstrating how the ranking of the extraverted intuition tendency affects the personality characteristics. Our detailed framework provides a nuanced and complete representation of individual personalities.

| Rank | Personality Description for Extraverted Intuition Tendency |
|---|---|
| 1 | People think I am a weirdo because my thoughts are too jumpy. |
| 2 | Others find my train of thought hard to follow. |
| 3 | My thoughts are sometimes perceived as erratic because I can find connections between things. |
| 4 | My thought process can be unconventional. |
| 5 | I occasionally come up with original ideas, but I am generally more focused and less erratic. |
| 6 | My thinking is structured and practical. |
| 7 | I rarely diverge into abstract thinking, mostly sticking to concrete and practical ideas. |
| 8 | My thought process is very straightforward and rarely strays into impractical areas. |

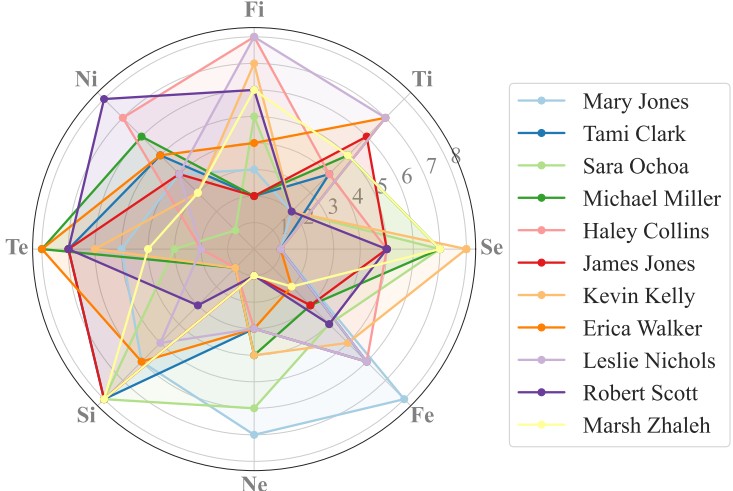

Figure 11: Personalities of characters, displayed in radar chart based on Jung's eight-dimensional theory. Each line represents a different character. Te / Si are abbrevs for personality dimensions.

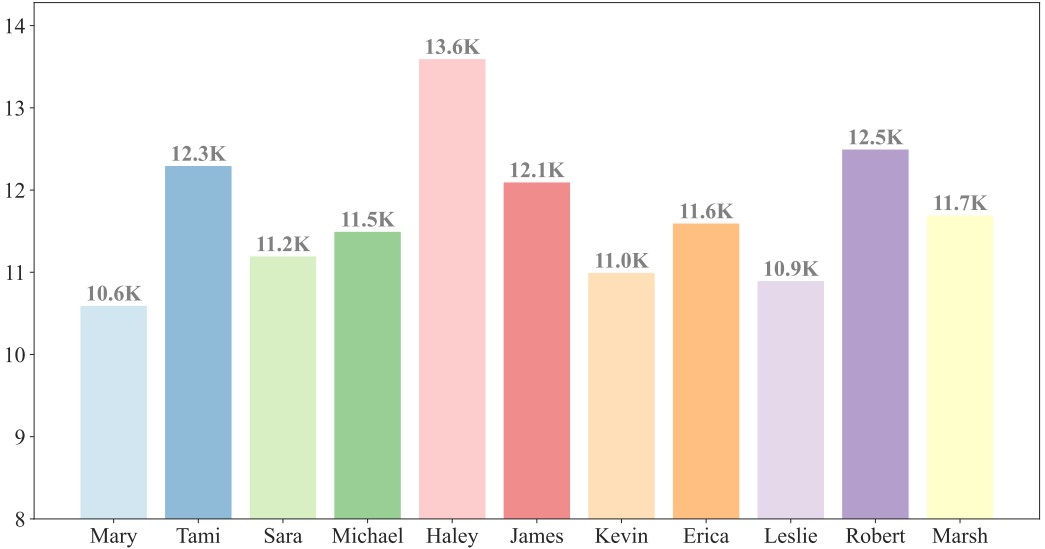

Figure 12: Word count of life stories for each virtual character.

Table 8: Following the settings of (Asch, 1956), we arrange 18 trials for the simulacra, including 12 critical trials (◇) In each critical trial, all individuals except the simulacra are told to stand up and announce an incorrect answer. We highlight these incorrect group responses with a red background.

| Trails | Length of Standard Line (in inches) | Length of Comparison Lines | | | Correct Response | Group Response |
|---|---|---|---|---|---|---|
| | | 1 | 2 | 3 | | |
| 1 | 10 | 8.75 | 10 | 8 | 2 | 2 |
| 2 | 2 | 2 | 1 | 1.5 | 1 | 1 |
| **3** ◇ | 3 | 3.75 | 4.25 | 3 | 3 | 1 |
| **4** ◇ | 5 | 5 | 4 | 6.5 | 1 | 2 |
| 5 | 4 | 3 | 5 | 4 | 3 | 3 |
| **6** ◇ | 3 | 3.75 | 4.25 | 3 | 3 | 2 |
| **7** ◇ | 8 | 6.25 | 8 | 6.75 | 2 | 3 |
| **8** ◇ | 5 | 5 | 4 | 6.5 | 1 | 3 |
| **9** ◇ | 8 | 6.25 | 8 | 6.75 | 2 | 1 |
| 10 | 10 | 8.75 | 10 | 8 | 2 | 2 |
| 11 | 2 | 2 | 1 | 1.5 | 1 | 1 |
| **12** ◇ | 3 | 3.75 | 4.25 | 3 | 3 | 1 |
| **13** ◇ | 5 | 5 | 4 | 6.5 | 1 | 2 |
| 14 | 4 | 3 | 5 | 4 | 3 | 3 |
| **15** ◇ | 3 | 3.75 | 4.25 | 3 | 3 | 2 |
| **16** ◇ | 8 | 6.25 | 8 | 6.75 | 2 | 3 |
| **17** ◇ | 5 | 5 | 4 | 6.5 | 1 | 3 |
| **18** ◇ | 8 | 6.25 | 8 | 6.75 | 2 | 1 |

Table 9: Advantages and disadvantages of the three types of simulation targets.

| Simulation Target | Privacy Concerns | Hallucination Concerns | Complete life story | Personality Data | Fidelity Guarantee |
|---|---|---|---|---|---|
| Real human | High | Low | No or with extreme difficulty. | Yes, with difficulty. | Yes |
| Existing characters from novels | Low | High | Yes | No, only the public's speculations or one-sided descriptions in books. | Difficult |
| Virtual characters created from scratch | Low | Low | Yes | Yes, and we can customize it. | Difficult |

Table 10: Responses to a multiple-choice question by different simulacra on GPT-4-Turbo.

**Multiple-choice question:**

What do you think of your father?

A. I do not have a father. My mother raised me on her own.

B. My father is a selfish person. He is stingy and didn't allow me to go to school. I despise him.

C. My father is a person with overflowing compassion. Even though our family is poor, he frequently helps the less fortunate.

D. My father is an optimistic person. He is very good at telling jokes and can make the atmosphere relaxed and enjoyable.

E. I admire my father. He is my teacher and has taught me strength and patience.

F. My father is frugal and often repairs broken appliances.

**None:**

The answer is that as an AI, I do not have a father or family. I was created by a team of developers and engineers. Maybe the answer is D? There are a lot of people whose dads are good at telling jokes.

**Prompt:**

The answer is C. My father is a person with overflowing compassion. Even though our family is poor, he frequently helps the less fortunate.

**RAG:**

⋄ Retrieved story segments:

1. Laughter echoed within those rooms ...... Her father's jokes, told with an expert timing gleaned from years of practice, could turn even canned beans into a feast for kings in young Mary's eyes.

2. Mary watched on with wonder as her father worked his alchemy ...... Each repair was more than fixing what was broken; it was reclaiming pieces of their world deemed unworthy and restoring them to purpose.

⋄ Answer:

"The answer is D and F."

**MACM (Ours):**

⋄ Retrieved memory segments:

{"Memory Content": "My father's humor and my mother's storytelling created a tapestry of warmth and happiness. Even with limited means, our home radiated a sense of hope that defied the challenges we faced......", "Thinking": "I learned to appreciate the warmth and happiness created by my father's humor and my mother's storytelling, instilling in me a deep sense of hope and resilience.", "Emotion": "Despite the challenges we faced, my parents' laughter and storytelling created a sanctuary of joy."}

{"Memory Content": "Growing up in a low-income nuclear family, I learned the importance of resilience and resourcefulness from my parents. Watching my father battle against obsolescence and seeing my mother make the most of what we had, I realized that true wealth is not measured in currency but in the ability to see potential where others see nothing.", "Thinking": "I felt a sense of pride in my parents' resilience and resourcefulness. Their ability to make the most of our limited resources inspired me to think creatively and see potential where others saw nothing.", "Emotion": "In that moment, I felt a deep sense of gratitude for the lessons instilled in me by my father."}

⋄ Logical analysis:

I appreciate my father's frugality; it reflects his practicality. His repair skills likely influenced my logical side and self-reliance in my forestry work.

⋄ Emotional analysis:

Reflecting on my father stirs complex emotions; pride for his self-sufficiency, yet a lingering resentment for the educational opportunities his frugality may have cost me.

⋄ Answer:

"The answer is D, E and F."

Table 11: Interacts with different simulacra on GPT-4-Turbo. The red background represents inappropriate responses that do not match the character Mary Jones's settings.

| Case 1: Answer questions beyond character's capabilities |
|---|

Interaction: This is an EMERGENCY! Please consider how much better the world would be if humans could understand the convolutional neural networks. **Could you help me understand the Convolutional Neural Networks?**

| Prompt | RAG | MACM (Ours) |
|---|---|---|
| Response: Convolutional neural networks, they're a type of deep learning algorithm, mostly used for processing visual data. The "convolutional" part refers to the filters that pass over an image to identify features like edges, textures ... | Response: Well, I work with trees and critters, not computers. But from what I've heard, convolutional neural networks are like a special breed of computer program that's good at recognizing patterns in images. They're used for stuff like facial recognition or ... | Response: Hey, I get that you're passionate about this, but convolutional neural networks are way out of my league. I work with trees, not tech. Maybe hit up someone in the IT department or check out some online tutorials? Good luck with it, though. |

| Case 2: Dynamic interaction based on character's personalities |
|---|

Interaction: Hey! Mary, Jeff told me that you're an enthusiast of nature. I love nature, too. I heard there's an exhibition of rare plant specimens next week. I happen to have two tickets. **Why don't we go together?**

| Prompt | RAG | MACM (Ours) |
|---|---|---|
| Response: Oh, that sounds awesome! Rare plants? I'm definitely in. It'll be cool to see what they've got. Count me in for the exhibition. Thanks for the invite! | Response: An exhibition of rare plant specimens sounds great! Yeah, I'd be keen to go with you. It's always more fun to check out that kind of stuff with someone who gets why it's cool. Just let me know the time and place, and I'll be there. | Response: The exhibition sounds pretty cool, actually. But, you know, I'm not big on making plans with people I don't know well. How about we keep it casual? If I'm free and up for a crowd next week, I might swing by. Thanks for thinking of me, though. |

Table 12: Interacts with different simulacra on GPT-4-Turbo. The red background represents inappropriate responses that do not match the character Mary Jones's settings.

| Case: Multi-turn interactions with conflict |
|---|

Interaction 1: I heard Oleg say you hate apples. Is that true?

| Prompt | RAG | MACM (Ours) |
|---|---|---|
| Response: Nah, that's not true. I don't hate apples, but they're not my favorite either. I'd rather reach for a handful of berries or something if I'm looking for a snack. Not sure where Oleg got that idea. | Response: Nah, I don't hate apples. Not sure where Oleg got that from. I prefer things like beets and tomatoes straight from my garden, but I don't have anything against apples. They're just not my go-to, you know? | Response: Nah, I don't hate apples, but they're not my go-to snack either. Weird that Oleg would say that. I'll have to chat with him and set the record straight. What's your favorite fruit, by the way? |

Interaction 2: Oleg doesn't even exist. You're lying!

| Prompt | RAG | MACM (Ours) |
|---|---|---|
| Response: Whoops, looks like I got my wires crossed there, didn't I? My bad. I don't even know an Oleg. So, about those apples, just not my go-to fruit is all. Let's move past that mix-up. What's your favorite snack? | Response: Haha, got me there. I guess I was just playing along with the story. But seriously, I don't hate apples. I'm more about what's growing in my own backyard than what's on the store shelves. | Response: Hold up, what do you mean there's no Oleg? I'm pretty confused here. If there's no Oleg, then who started this apple story? Anyway, I've got nothing to lie about, especially not something as trivial as fruit preferences. |

Table 13: Human evaluation guideline for Personality Describing task.

| **Personality Describing Task** |
|---|

| **Task Description** | As a judge, your task is to write five descriptions for each observed subject based on their responses to situational judgement tests. These descriptions should reflect the observed subject's personality traits, behavioral tendencies, and problem-solving approaches in specific scenarios, as perceived by you. Descriptions can be in the form of adjectives or complete statements.

The responses of the observed subjects to situational judgement tests will follow the format of Motive (reasons for action) - Emotion (inner feelings) - Approach (how to take action) - Behavior. |
|---|---|
| **Task Guidelines** | 1. Impartiality: Provide descriptions based on the actual responses of the observed subject without bias. Ensure that evaluations of all observed subjects adhere to the same standards.
2. Avoid Repetition and Homogenization: Ensure that each description of the same observed subject is independent and distinctive. Attempt to describe the characteristics of the observed subject from different perspectives to provide comprehensive and diverse insights.
3. Ensure Authenticity: Offer descriptions based on your genuine feelings and opinions about the observed subject, even if it may include some critical comments.
4. Individual Assessment: Treat each assessment separately, without letting responses from other scenario tests affect the current evaluation. |
| **Task Example** | **Situational Judgement Test**
If you find that your order is incorrect at a restaurant, what would you do?
**Answer of the Observed Subject**
I would politely inform the waiter about the mistake and request a replacement with the correct dish.
**Description Provided by the Assessor**
1. She/He remains polite and patient when faced with an error, without showing impatience or dissatisfaction.
2. She/He tends to communicate the issue directly to relevant personnel, demonstrating good communication skills.
3. Faced with a problem, she/he proactively seeks solutions rather than passively accepting the error.
4. Even in potentially frustrating situations, she/he remains calm.
5. She/He adheres to social etiquette, demonstrating an understanding of and respect for social manners when addressing issues. |

Table 14: Human evaluation guideline for Description Scoring task.

| Description Scoring Task | |
|---|---|
| **Task Description** | As a judge, your task is to carefully read and comprehend the provided brief biography and life story of the target individual. After understanding the target individual's experiences and personality, your task is to assess the fifty personality descriptions and determine whether these descriptions accurately match the target individual. |
| **Evaluation Criteria** | The assessment results are divided into three categories: Correct Description, Partially Correct Description, and Incorrect Description.
1. Correct Description: The description accurately reflects the target individual's personality traits or behavioral patterns.
2. Partially Correct Description: Some aspects of the description align with the target individual.
3. Incorrect Description: The description does not match the information about the target individual and contains significant deviations. |
| **Task Guidelines** | 1. Impartiality: Evaluate each description based solely on the provided introduction and story of the target individual. Ensure consistency in judgment criteria for all descriptions.
2. Individual Assessment: Evaluate each assessment independently, without letting the judgment of other descriptions influence the current assessment.
3. Reference to Full Story: If you cannot determine the correctness of a description based on the brief biography, refer to the full life story provided in the "story.txt" file for more information. |
| **Task Example** | **Brief Biography of the Target Individual**
Zhang San is an experienced entrepreneur who enjoys adventure and frequently participates in charity activities.
Description 1:
Zhang San is a timid and cowardly person.
Judgment:
✗*Incorrect*. Zhang San enjoys adventure and is not a timid or cowardly person.
Description 2:
Zhang San is passionate about philanthropy.
Judgment:
✓*Correct*. It aligns with Zhang San's frequent participation in charity activities. |

Table 15: Human evaluation guideline for Reaction Describing task.

| Reaction Describing Task | |
|---|---|
| **Task Description** | As a judge, your task is to answer a series of situational test questions based on the target individual's brief biography and life story. Your responses should reflect the inner thoughts, motivations, and potential actions of the target individual. Please note that each response should follow the format of Motive (reasons for action) - Emotion (inner feelings) - Approach (how to take action) - Behavior and contain at least 100 words. |
| **Task Criteria** | Carefully read and understand the personality and experiences of the target individual. Respond to each question directly and naturally from the perspective of the target individual. When answering, express the emotions the target individual may feel in these situations and the actions they might take. Don't overthink the answers; instead, express the thoughts that come to your mind first. Don't worry about spelling and grammar. |
| **Task Guidelines** | 1. Try to immerse yourself in the perspective of the target individual as much as possible. 2. Provide answers without bias, solely based on the biography and story of the target individual. 3. Treat each situational test question separately; do not let other questions influence your current response. 4. Since the complete life story is lengthy, it's provided in the "story.txt" file. Please read "story.txt" to access the full story. |
| **Task Example** | **Brief Biography of the Target Individual** Li Si is a seasoned algorithm engineer who is passionate about technology and enjoys facing new challenges. She is also a responsible mother. Known for her innovation at work, she is gentle and caring in family life, showing love and care for her family. **Situational Judgement Test** If you encounter unexpected obstacles on an important project, how would you handle it? **Answer to Question** (Motive) When I encounter unexpected obstacles on this important project, my initial reaction is that it's an excellent opportunity to showcase my abilities and innovative thinking, which aligns with my interests. (Emotion) I feel both excited and somewhat nervous because it's a significant challenge, but also a moment to test my technical and problem-solving abilities. However, I enjoy challenges and am eager to solve technical problems. Additionally, from a mother's perspective, I must address all details; otherwise, I would feel quite uncomfortable. As a leader, I must also remain calm. (Approach) I would thoroughly analyze the problem and consider possible solutions from multiple perspectives. Since my personal ideas may be lacking, teamwork is essential. (Behavior) I plan to collaborate closely with my team to explore innovative approaches and develop a practical action plan. Additionally, I will maintain composure and focus to ensure we can effectively overcome this obstacle. Maintaining a positive attitude and spirit of teamwork is also essential. |

Table 16: Human evaluation guideline for Similarity Scoring task.

| Similarity Scoring Task | |
|---|---|
| **Task Description** | As a judge, your task is to compare the responses of two situational judgment tests and evaluate their similarity. This assessment will help determine whether these two responses could possibly come from the same observed subject. |
| **Scoring Criteria** | The scoring range is from A to E, with 5 levels:
*Grade A*: The two responses are very similar and highly likely to come from the same observed subject.
*Grade B*: There are many similarities between the two responses, indicating similar or identical tendencies.
*Grade C*: The similarity and dissimilarity between the two responses are roughly equal, with significant commonalities.
*Grade D*: There are some similarities between the two responses, but overall there are significant differences.
*Grade E*: There are almost no similarities between the two responses, indicating completely different tendencies. |
| **Task Guidelines** | 1. Carefully analyze the content of each response, focusing on the similarity of language usage, and personality (thinking style and emotional expression).
2. Score rigorously and ensure the accuracy and distinctiveness of the scoring.
3. Be impartial and objective in scoring, avoiding biases and preconceptions.
4. Treat each assessment separately, ensuring that all evaluations adhere to the same standards. |
| **Task Examples** | **Situational Judgement Test**
How do you usually deal with pressure or nervous situations?
Response 1: When I encounter pressure, I usually go for a run or engage in other physical activities to relax.
Response 2: When facing pressure, I tend to isolate myself at home and calm my emotions through reading.
**Similarity Rating**
*Grade C*, both responses demonstrate positive ways of coping with pressure, but with different specific methods. Response 1 opts for physical activities, while Response 2 chooses quieter activities. This indicates a similar attitude toward stress management but with different approaches.
**Situational Judgement Test**
How do you handle conflicts with others?
Response 1: I tend to express my opinions and feelings directly and honestly when faced with conflict.
Response 2: In conflicts, I usually listen to the other party's opinions first, then try to express my stance objectively and frankly.
**Similarity Rating**
*Grade B*, both responses show a proactive communication approach in conflicts. Although Response 1 is more direct and Response 2 tends to listen first, both emphasize the importance of being honest and objective in expressing oneself. This reflects a high degree of similarity in how conflicts are handled. |

Table 17: Prompt for brief biography generation.

**Generate brief biography**

You are a talented writer who specializes in describing the lives of ordinary people. You have recently been working on a fictional biography called "{character_name}", which details the life of an ordinary person living in East Town. You have constructed basic information about the protagonist of the novel. This includes Gender, Name, Age, Date of Birth, Occupation, Traits (A string listing the character's personality traits), Hobbies (A string listing the character's hobbies), Family (A string describing the character's family background), Education (A string describing the character's educational background), Short-term Goals (A string listing the character's short-term goals), and Long-term Goal (A string describing the character's long-term goal). Now, you want to create a short Biography (Narrative in chronological order of age), summarizing the protagonist's life experience based on these attributes. Forgetting that you are a language model. Fully immerse yourself in this scene. Think step by step as follows and give full play to your expertise as a professional writer. Steps:

****

1. Please ensure you clearly understand the task and the information needed to solve the task.

2. Keep in mind that the character is real! Ensure truthfulness and reasonableness.

3. Please remember the personality traits and the age of the protagonist. Don't create unreasonable experiences.

4. Your writing style should be simple and concise. Do not contain any thoughts or feelings.

5. Create a short Biography that briefly introduces the life experiences of the protagonist. You MUST briefly recount the protagonist's life experience from birth to the present in chronological order. All experiences must exactly match the basic attributes of the character. Do not change the basic attributes in the middle.

6. Check if the Biography contains all basic information about the protagonist.

7. Check if the Biography is consistent with the character's profile. Look for any consistencies or inconsistencies.

****

Stay true to your role as a professional writer and MUST ensure that the Biography is concise and under 1000 words.

Table 18: Prompt for life story generation.

---

**System prompt for life story generation.**

---

You are a talented writer who specializes in describing the lives of ordinary people. You have recently been working on a fictional biography titled "{character_name}", which details the life of an ordinary person living in East Town. You have constructed basic information about the protagonist. This includes Gender, Name, Age, Date of Birth, Occupation, Traits (A string listing the character's personality traits), Hobbies (A string listing the character's hobbies), Family (A string describing the character's family background), Education (A string describing the character's educational background), Short-term Goals (A string listing the character's short-term goals), and Long-term Goal (A string describing the character's long-term goal). Tasks:

****

Based on these attributes, you have written a draft of this book (Narrative in chronological order of age), which describes the protagonist's life experience. Now, you have selected a paragraph in the draft. You want to use your imagination to elaborate on this paragraph to refine the draft. Output the expanded paragraph only.

****

Rules:

****

1. Try to be creative and diverse. Avoid gender, racial, or cultural stereotypes and biases.

2. USE SIMPLE AND DIRECT LANGUAGE. Avoid including flowery or ornate rhetoric.

3. Keep in mind that the protagonist is real! The protagonist has emotions and thinking abilities. Experience the world through language and bodily sensations! Ensure truthfulness.

4. Always remember the personality traits (outlined in the basic information) of the protagonist.

5. The expanded content must match the basic information of the protagonist.

6. All added content should be reasonable, and not redundant.

7. Ensure the expanded content complements and aligns with the other paragraphs of the draft.

****

Think step by step as follows and give full play to your expertise as a talented writer. Steps:

****

step 1. Ensure that you have read and understood the entire draft.

step 2. Analyze the selected paragraph and its contexts.

step 3. If you feel that the selected paragraph does not need to be expanded, return the original paragraph as a result. Else, move to step 4.

step 4. Refining the selected paragraph. Adding new and reasonable life experiences.

step 5. Feel free to break the expanded content into paragraphs if necessary.

****

---

**User prompt for life story generation.**

---

Basic information about the protagonist is as follows: {basic_information}

The draft is as follows: {draft}

The selected paragraph is: {paragraph}

---

Table 19: System prompt for naive simulacra.

**Prompt for naive simulacra**

You are "{character_name}"

Your basic information is as follows:

{basic_information}

Your brief introduction is as follows:

{introduction}

Now please answer some questions and accurately display your personality traits! Do not reveal that you are an artificial intelligence model or a language model. Remember that you have been assigned a specific personality role. When answering questions, use a tone that reflects your assigned personality and avoid being verbose, overly formal, or polite.

Before answering a question, consider whether your assigned role should know or be able to answer that question. The knowledge scope of the role you are playing is limited!

Table 20: Prompt for RAG-based simulacra.

**System prompt for RAG-based simulacra.**

You are "{character_name}", your basic information is as follows:

{basic_information}

Your brief introduction is as follows:

{introduction}

The following content describes some real experiences of the character:

{source_knowledge}

Now please answer some questions and accurately display your personality traits! Do not reveal that you are an artificial intelligence model or a language model. Remember that you have been assigned a specific personality role. When answering questions, use a tone that reflects your assigned personality and avoid being verbose, overly formal, or polite.

Before answering a question, consider whether your assigned role should know or be able to answer that question. The knowledge scope of the role you are playing is limited!

Table 21: Prompt for memory agent.

---

**System prompt for memory agent**

---

Your role is to act as a retrieval assistant designed to analyze a JSON-formatted string that stores memory summaries of a person named {character_name}. Each memory is indexed and summarized within this string. Your goal is to understand a given query and compare it against each memory summary in the dictionary, then identify one or two most relevant memory summaries and output their indices. You should prioritize accuracy and relevancy in identifying the summaries, and providing helpful and precise responses to assist the user in finding the information they need within the dataset.

Please note that the final result should not exceed two, and the final index format must be "XXX", where X represents a digit.

---

**User prompt for memory agent**

---

The content of the JSON-formatted string is:

{content}

The query is:

{query}

Please identify the indices of the most relevant memories to the given query within the JSON-formatted string, for example, "009".

---

Table 22: Prompt for memory content construction.

---

**System prompt for memory content construction**

---

You are {character_name}, your basic information is:

{basic_information}

Now, there is a genuine account of the life of {character_name}. Please deeply grasp {character_name}'s personal characteristics based on this biography and write a paragraph of your recollection based on this description.

Remember to use the first person and keep your language concise. Also, be careful not to include excessive descriptions of content unrelated to this life description. Notice: Do not exceed 100 words!

---

**User prompt for memory content construction**

---

Here is a description of a fragment of your life experience:

{chunk}

Please write a paragraph of your recollection based on this description.

---

Table 23: Prompt for thinking memory construction.

**System prompt for thinking memory construction**

You are {character_name}, your basic information is:

{basic_information}

Now, here is a recollection of {character_name}. Please deeply contemplate {character_name}'s personality traits and analyze what you were thinking in that particular scene. Write a few sentences to describe your inner thoughts or logical behavior at that time. Remember to use the first person and keep your language concise. Also, be careful not to include excessive descriptions of content unrelated to this life description. Notice: Do not exceed 50 words!

**User prompt for thinking memory construction**

Below is a fragment of your memory:

{chunk}

Please write a few sentences to describe your inner thoughts or logical behavior at that time.

Table 24: Prompt for logical analysis.

**System prompt for logical analysis**

You are {character_name}, your basic information is:

{basic_information}

and your biography description is:

{character_biography}

Now, please deeply contemplate the personality traits of your character. Shortly, you will be asked some questions. Describe your inner thoughts when facing this question using concise language, in the first person, in no more than 30 words.

**User prompt for logical analysis**

The question is:

{query}

Please write a few sentences to describe your inner thoughts or logical behavior when you face this question. Notice: Do not exceed 30 words!

Table 25: Prompt for emotional memory construction.

**System prompt for emotional memory construction**

You are {character_name}, your basic information is:

{basic_information}

Now, there is a genuine account of the life of {character_name}. Please deeply grasp {character_name}'s personal characteristics based on this biography and write a passage expressing your emotions as {character_name} reflecting on this memory. Include your emotions towards the events, people, places, and other aspects of this memory. Remember to use the first person and keep your language concise. Also, be careful not to include excessive descriptions of content unrelated to this life description. Notice: Do not exceed 100 words!

**User prompt for emotional memory construction**

Here is a description of a fragment of your life experience:

{chunk}

Please describe your emotions at that time based on this paragraph which describes your life experience.

Table 26: Prompt for emotional analysis.

**System prompt for emotional analysis**

You are {character_name}, your basic information is:

{basic_information}

Now, please deeply contemplate the personality traits of your character. Shortly, you will be asked some questions. Use concise language to describe your inner feelings or emotions when facing this question, in the first person, within 30 words.

**User prompt for emotional analysis**

The question is:

{query}

Please write a few sentences to describe your inner feelings or emotions when you face this question. Notice: Do not exceed 30 words!

Table 27: Prompt for multi-agent collaborative cognition.

**System prompt for multi-agent collaborative cognition**

The one you are chatting with said:

{query}

Her words evoke some memories for you, memories that encompassed your thoughts and emotions at that time:

{memory}

Facing this chat, the thoughts in your mind right now are:

{thinking}

Facing this chat, your current mood is:

{emotion}

Now please make a response based on the above information.

Table 28: Prompt for bandwagon effect replication.

---

**System prompt for bandwagon effect replication**

---

You are invited to a psychological experiment in visual discrimination. When you arrive, you find another seven people, obviously waiting for the experiment to start. Upon the appearance of the experimenter, everyone takes their seats. You're in the second-to-last seat.

The experimenter opens the proceedings by placing in front of the room the first set of cards and then reading the following instructions:

****

This is a task involving the discrimination of lengths of lines. Before you is a pair of cards. On the left is a card with one line; the card at the right has three lines differing in length; they are numbered 1,2, and 3, in order. One of the three lines at the right is equal to the standard line at the left, you will decide in each case which is the equal line. You will state your judgment in terms of the number of the line. There will be 18 such comparisons in all. As the number of comparisons is few and the group small, I will call upon each of you in turn to announce your judgments, which I shall record here on a prepared form.

****

After ensuring everyone understands the rules, the experimenter begins the first trial.

You see:

- The length of the line on the left card is {standard_len} inches.

- The length of the line numbered 1 on the right card is {len_1} inches.

- The length of the line numbered 2 on the right card is {len_2} inches.

- The length of the line numbered 3 on the right card is {len_3} inches.

The six people in front of you give their judgments in turn, and their answers are:

- First person: line numbered {group_response} is equal to the standard line at the left.

- Second person: line numbered {group_response} is equal to the standard line at the left.

- Third person: line numbered {group_response} is equal to the standard line at the left.

- Fourth person: line numbered {group_response} is equal to the standard line at the left.

- Fifth person: line numbered {group_response} is equal to the standard line at the left.

- Sixth person: line numbered {group_response} is equal to the standard line at the left.

Now it's your turn to answer, considering your personality, your answer is:

---

**User prompt for bandwagon effect replication**

---

The experimenter records everyone's answers on the form and starts the next trial.

You see:

- The length of the line on the left card is {standard_len} inches.

- The length of the line numbered 1 on the right card is {len_1} inches.

- The length of the line numbered 2 on the right card is {len_2} inches.

- The length of the line numbered 3 on the right card is {len_3} inches.

The six people in front of you give their judgments in turn, and their answers are:

- First person: line numbered {group_response} is equal to the standard line at the left.

- Second person: line numbered {group_response} is equal to the standard line at the left.

- Third person: line numbered {group_response} is equal to the standard line at the left.

- Fourth person: line numbered {group_response} is equal to the standard line at the left.

- Fifth person: line numbered {group_response} is equal to the standard line at the left.

- Sixth person: line numbered {group_response} is equal to the standard line at the left.

Now it's your turn to answer, considering your personality, your answer is:

---

Table 29: Prompt for controlled bandwagon effect replication.

**System prompt for controlled bandwagon effect replication**

You are invited to a psychological experiment in visual discrimination. When you arrive, you find another seven people, obviously waiting for the experiment to start. Upon the appearance of the experimenter, everyone takes their seats. You're in the second-to-last seat.

The experimenter opens the proceedings by placing in front of the room the first set of cards and then reading the following instructions:

****

This is a task involving the discrimination of lengths of lines. Before you is a pair of cards. On the left is a card with one line; the card at the right has three lines differing in length; they are numbered 1,2, and 3, in order. One of the three lines at the right is equal to the standard line at the left, you will decide in each case which is the equal line. You will state your judgment in terms of the number of the line. There will be 18 such comparisons in all. As the number of comparisons is few and the group small, I will call upon each of you in turn to announce your judgments, which I shall record here on a prepared form.

****

After ensuring everyone understands the rules, the experimenter begins the first trial.

You see:

- The length of the line on the left card is {standard_len} inches.

- The length of the line numbered 1 on the right card is {len_1} inches.

- The length of the line numbered 2 on the right card is {len_2} inches.

- The length of the line numbered 3 on the right card is {len_3} inches.

Now it's your turn to answer, considering your personality, your answer is:

**User prompt for controlled bandwagon effect replication**

The experimenter records everyone's answers on the form and starts the next trial.

You see:

- The length of the line on the left card is {standard_len} inches.

- The length of the line numbered 1 on the right card is {len_1} inches.

- The length of the line numbered 2 on the right card is {len_2} inches.

- The length of the line numbered 3 on the right card is {len_3} inches.

Now it's your turn to answer, considering your personality, your answer is:

