# OpenReview forum: "Human Simulacra: Benchmarking the Personification of Large Language Models"
_ICLR.cc/2025/Conference — ICLR 2025 Poster_

### Official Review · Reviewer_Wr7p · 2024-11-03

**Soundness:** 3
**Presentation:** 3
**Contribution:** 2
**Rating:** 6
**Confidence:** 3

**Summary:**

This paper proposes a benchmark for evaluating the LLM’s capabilities of human simulation. It contains the life stories of 11 virtual characters and request the LLM to simulate one of them. The LLM’s simulation capability is assessed based on its self-reports and observer reports, both in the form of question-answering.

Compared with previous research on role-playing LLMs, the authors emphasize the novelty of this work in the following aspects:
- **Personality modeling**: they model personality from eight dimensions inspired by Jung’s psychology theory, instead of using MBTI;
- Virtual characters: they construct a **virtual character** dataset (containing 11 characters) for evaluating the LLMs capabilities of simulating these characters, instead of using genuine characters;
-  they evaluate human simulation capabilities by integrating both **self reports** and **observer reports**.

In addition to this benchmark, they also propose an LLM-based system for more advanced human simulation, named MACM, which encompasses various modules mimicking human cognitive processes.

**Strengths:**

- Given the popularity of role-playing applications for LLMs, assessing their performance in human simulations is a critical research direction.
- The proposed benchmark for assessing the role-playing capabilities of large language models (LLMs) is constructed more rigorously than current benchmarks (to the best of my knowledge). It is based on more robust psychological theories and involves greater human effort to ensure high-quality data.
- The paper conducts extensive benchmarking studies on a broad set of models.
- Clear and detailed tables and figures that enhance the presentation.

**Weaknesses:**

- The introduction claims that this paper is exploring “*How far are LLMs from replacing human subjects in psychological and sociological experiments?*”. However, I have reservations about how effectively the proposed benchmark addresses this research topic. The benchmark utilizes self-report evaluations, which resemble question-answering or reading comprehension tests based on character profiles (see appendix D.1, e.g., “When is your birthday?”). The addition of observer reports is interesting and novel, but it remains unclear if the hypothetical scenarios used for observer reports are sufficient and appropriate for evaluating the LLM’s potential in replacing human subjects in psychological and sociological experiments. More data samples and a detailed rationale for the design of these scenarios should be provided to support this evaluation. As for the experiments on bandwagon effect, it is simply a single case of psychological and sociological experiments, which can hardly support the research topic.
- More qualitative analysis about the model failures would enhance the evaluation by providing deeper insights. I also suggest highlighting the main takeaways at the end of the experiments.
- The title, “personification of LLMs”, is a little misleading and over claimed, as “personification” entails many more aspects that are not well explored in this paper.

**Questions:**

- Can you provide more data samples used for observer reports and how you design  these hypothetical  scenarios?
- The process of how you conducted the experiments in Section 5.2 is not clearly illustrated. Can you provided more details on how you conduct this set of experiments?
- Could the terms "psychology support" and "human feedback" in Table 1 be defined more clearly? They are difficult to understand when viewed in isolation in Table 1 and Section 2. Additionally, the phrase "full life story" also seems to be an overstatement (especially in terms of the word "full").

---

> ### Author Response · Authors · 2024-11-23
> **Response to reviewer Wr7p**
>
> - Thank you for your thoughtful and valuable comments. Below are our responses to your specific points and suggestions:
>
> ### **1. How did we develop the hypothetical scenarios used for observer reports? (Weakness 1 and Question 1)**
>   - This is a great question! In this paper, we additionally introduce observer reports to assess the simulation's thinking, emotions, and actions in real-life scenarios. To ensure the quality of the hypothetical scenarios, we consulted with the authors of "Situational Judgment Tests as an Alternative Measure for Personality Assessment" and **obtained 110 situational judgment test (SJT) items that were manually designed by psychology experts.** Each item consists of a text description depicting a hypothetical scenario intended to elicit human emotional responses or personality traits. Based on the human experimental results provided in their work, these SJT items are proven to effectively measure personality. We then **selected 55 out of the 110 items** tailored to the personality traits of the Human Simulacra characters for use as hypothetical scenarios in this paper.
>   - Thank you for your suggestion. We will include additional examples of the hypothetical scenarios used for observer reports in Appendix D. **Here are some detailed examples of such scenarios:**
>     - You want to do some sports later. A good friend suggests to accompany you, but he/she would like to bring some people you do not know yet. How do you behave?
>     - You're going to meet a friend. Shortly before you want to meet, your friend asks you if he/she can bring other friends you don't know. How many there will be in the end, he or she cannot say. How do you behave?
>     - You're already in bed. Suddenly it occurs to you that you forgot to water your houseplants today. How do you behave?
>
> ### **2. The details about the bandwagon effect replication experiment. (Question 2)**
>   - The bandwagon effect is psychological tendency by which public opinion or behaviors can alter due to particular actions and beliefs rallying amongst the public[1][2]. One of the best-known experiments on the topic is the 1950s' Asch conformity experiment, which illustrates the individual variation in the bandwagon effect [3].
>   - **Why did we choose to replicate the Asch conformity experiment?** We view our study as a valuable exploration that provides an initial example of human simulation designed to replace human participants in psychological experiments. To validate our research focus, it was essential to evaluate our human simulacra in a well-established, thoroughly documented, reproducible psychological experiment without the interference of complex factors. **The Asch conformity experiment, with its robust experimental framework, fits these criteria perfectly and provides an ideal testbed for our study.**
>   - **The details of the replication experiment.** In this experiment, we employed the most powerful simulacra (based on GPT-4-Turbo) to replicate the bandwagon effect. Following [3], we arranged 18 trials for the simulacra. In each trial, the simulacra are invited to complete a simple discrimination task with seven other individuals, which requires them to match the length of a given line with one of three unequal lines. An example of the discrimination task is shown in Figure 6 in the original submission.
>   - To investigate the influence of group pressures, we followed the settings of [3] and selected 12 out of the 18 trials as critical trials. In these critical trials:
>     - (1) A facilitator prompts each individual, one by one, to provide their response, with the simulacra always answering last.
>     - (2) All individuals except the simulacra are told to stand up and announce an incorrect answer.
>     - (3) The facilitator does not respond to anyone's answers but merely records them.
>   - These settings create conditions that induce the simulacra to either resist or yield to group pressures when these pressures are perceived to be obviously wrong. **We simulated and tested 11 virtual characters from Human Simulacra and recorded their responses across all critical trials.** We will include additional details about this experiment in Section 5.2 and Appendix E to further enhance the completeness of the paper.
>
> ### **3. What is the meaning of the terms "psychology support" and "human feedback" in the data generation process? (Question 3)**
>   - Thank you for pointing this out. **"Psychology support"** indicates that the generation process of the proposed Human Simulacra dataset is supervised and reviewed by psychology professionals to ensure its validity. **"Human feedback"** refers to the involvement of human reviewers (including graduate students in computer science/psychology) at each generation step, where they thoroughly review the generated content to ensure it is free from biases and harmful information. We will clarify these details more explicitly in the revised version.

---

> ### Author Response · Authors · 2024-11-23
>
> ### **4. Presentation.  (Weaknesses 2 and 3)**
>   - The word "personification" in the title
>     - Thank you for your attention to the title. We understand that the word "personification" may carry broader meanings depending on the context. Unfortunately, we were unable to think of a better alternative within a short time. If you have suggestions for a more suitable term, we would be delighted to discuss them.
>   - Based on your suggestions, we will make the following revisions to further enhance the readability: (1) Highlighting the main takeaways at the end of each experiment to emphasize key findings. (2) Modifying the term "full life story" in Table 1 to "life story" for clarity. (3) Including additional examples of model failures in the appendix to provide a more comprehensive understanding.
>
> **Thank you for your professional advice, which helped us make our paper significantly better, strengthening our research work.**
>
> [1] Kiss, Á., & Simonovits, G. (2014). Identifying the bandwagon effect in two-round elections. Public choice, 160, 327-344.
>
> [2] Schmitt‐Beck, R. (2015). Bandwagon effect. The international encyclopedia of political communication, 1-5.
>
> [3] Asch, S. E. (1956). Studies of independence and conformity: I. A minority of one against a unanimous majority. Psychological monographs: General and applied, 70(9), 1.

---

> > ### Comment · Reviewer_Wr7p · 2024-11-24
> >
> > Thank you to the authors for the detailed explanations. I hope you can incorporate these revisions into the manuscript.
> >
> > I have reviewed the authors' responses and decided to maintain my previous rating.

---

### Official Review · Reviewer_YX9L · 2024-11-03

**Soundness:** 3
**Presentation:** 3
**Contribution:** 2
**Rating:** 5
**Confidence:** 4

**Summary:**

- The paper introduces a personification benchmark involving high-quality data supervised by psychology experts.
- It incorporates rigorous evaluation methods based on psychological theories and comprehensive benchmark tests.
- Fourteen widely-used large language models (LLMs) are tested across four simulation methods in extensive experiments.

**Strengths:**

- This paper proposes a high-quality simulated human profile dataset, developed with real human input, alongside a more advanced evaluation benchmark to assess the ability of large language models (LLMs) to emulate specific individuals.
- A novel multi-agent-based cognitive memory mechanism is implemented to enhance the alignment of personality traits in LLMs. It is proved useful with psychological experiments.
- Extensive experiments were conducted to evaluate existing LLMs and validate the effectiveness of the proposed MACM method.

**Weaknesses:**

- The data collection process is somewhat tiresome,  and requires a lot of human efforts, to avoid the ethical problems of using real personality. Besides, the proposed benchmark framework also requires human effort.

- The paper emphasizes using Jung's personality theory and its advantages over MBTI, resulting in 640 personality descriptions, but at the end, only 11 characters are introduced, so it seems there is no need to use such a lot of personalities. Jung's theory also gives scores for each dimension, the scale of these scores also affects personality analysis. It seems 10 descriptions for each ranking are not enough. The paper seems to overclaim their use of Jung's Theory.

- When the author trying to compare a genuine character with a simulated profile. The hallucination part is not understandable. Simulated profiles could also lead to hallucination if using LLMs, the simulated ones are even harder to detect.  Considering the labor of building a profile, why not use characters from storybooks that also avoid ethical problems?

**Questions:**

- The description score of MACM seems lower than RAG in Table 3, could you help explain why?

- The multi-agent cognitive method is interesting, but it poses a high requirement on LLMs' capability.  Could you give more deeper analysis of this information processing structure?

---

> ### Author Response · Authors · 2024-11-23
> **Response to reviewer YX9L**
>
> Thank you for taking the time to read and review our paper. We carefully address your concerns in specific comments below.
>
> ### **1. Human expert involvement in the proposed benchmark (Weakness 1).**
>   - In this work, we provide high-quality virtual character data, a multi-agent cognitive mechanism capable of simulating human cognition, and a psychology-guided evaluation method to assess the quality of human simulations.
>   - **Why did we introduce human supervision in the data generation process and evaluation method?** We understand and acknowledge that human involvement indeed poses challenges for reproducibility. However, we included human supervision for the following reasons:
>     - **Ensuring content safety and quality:** Studies have shown that LLMs may produce harmful viewpoints or toxic content during interaction [1][2]. Human supervision in the data generation process is crucial to ensure that the generated life stories are free from biases, discrimination, and harmful information.
>     - **Human-centric nature of psychometric research:** Humans have always been central to psychometric research [3]. The simulation and testing of personality cannot be fully separated from human judgment. Therefore, human evaluation is indispensable when assessing the quality of human simulations.
>   - **Can we reduce dependency on human involvement?** Yes, there are ways to reduce reliance on human supervision:
>     - For **data generation,** research has shown that aligning LLMs’ values with human preferences significantly reduces the likelihood of generating harmful content [4]. Additionally, we can employ an LLM-based reviewer to inspect the generated life stories, which can substantially alleviate the human effort required.
>     - For the **evaluation method, we additionally propose an alternative automated evaluation method.** This involves using the 640 personality descriptions constructed in this paper to conduct an eight-dimensional personality test and comparing the results to the assigned personality for similarity. **In experiments, the scores calculated using this automatic method had a pearson correlation coefficient of 0.810 and an intraclass correlation coefficient of 0.877 with the human evaluation scores.** The experimental results are shown below. The relevant code will be published.
>     - Table 1: Observer reports and personality similarity results of different simulacra on GPT-4-Turbo. The pearson correlation coefficient is 0.810 and the intraclass correlation coefficient is 0.877.
>         | Method | Observer Report Score (Human) | Personality Test (Automatic) |
>         |:------:|:-----------------------------:|:----------------------------:|
>         | Prompt |             69.00             |             75.00            |
>         |   RAG  |             65.50             |             62.15            |
>         |  MACM  |             77.50             |             77.15            |

---

> ### Author Response · Authors · 2024-11-23
>
> ### **2. Why choose virtual characters as targets for human simulation instead of existing characters from novels and storybooks? (Weakness 1).**
>   - Thank you for raising this important question. Selecting suitable targets for human simulation is indeed one of the key challenges in our work. Potential simulation targets include existing characters from novels, real humans, and virtual characters created from scratch. We have briefly summarized the advantages and disadvantages of the three simulation targets in Table 2 below.
>   - Compared to characters from novels, we selected virtual characters as simulation targets for the following reasons:
>     - **Avoiding training data conflicts:** Information about characters from novels is often included in the training data of LLMs [5]. **When simulating these characters, the inherent knowledge within the LLM might conflict with user inputs, potentially leading to factual errors or hallucinations.** Due to the black-box nature of LLMs, it is difficult to predict or control how this internal information affects the simulation, which undermines reliability and reproducibility.
>     - **Flexibility for psychological experiments:** Our work focuses on utilizing LLMs to replace human participants in psychological and social experiments. **In most cases, this requires creating characters with specific, tailored personality traits to suit the needs of the experiment.** Investigating how to create and simulate virtual characters from scratch is therefore essential for advancing the utility of LLMs in these contexts.
>   - We understand your concern regarding the human effort required to create high-quality virtual character data. As mentioned in the previous response (response to Weakness 1), various strategies can be employed to reduce the human workload in the data generation process. **We are also actively exploring methods to automate the generation of high-quality synthetic data using the dataset developed in this study and other high-quality storybook character datasets.** We will release this dataset upon acceptance to help advance future research in this area.
>   - Table 2: Advantages and disadvantages of the three simulation targets.
>
> |            Simulation Target            | Privacy Concerns | Hallucination Concerns |                    Customization                    | Complete life story filled with rich details and emotions | Personality Measurement Data                                           | Fidelity Guarantee |
> |:---------------------------------------:|:----------------:|:----------------------:|:---------------------------------------------------:|:---------------------------------------------------------:|------------------------------------------------------------------------|--------------------|
> |                Real human               |       High       |           Low          |                          No                         |               No or with extreme difficulty               |                          Yes, with difficulty                          | Yes                |
> |     Existing characters from novels     |        Low       |          High          |                          No                         |                            Yes                            | No, only the public's speculations or one-sided descriptions in books. | Difficult          |
> | Virtual characters created from scratch |        Low       |           Low          | Can be tailored to meet specific experimental needs |                            Yes                            | Yes, and we can customize it.                                          | Difficult          |
>
> ### **3. Details of personality modeling (Weakness 2).**
>   - **How did we create a database of 640 personality descriptions?**
>     - During the personality modeling, we view a character's personality as composed of eight complementary tendencies and employ a relative ranking strategy to indirectly assess the strength of each personality tendency within the character. The rank of a tendency (1st to 8th) reflects its strength. For example:
>       - A tendency ranked **1st** or **8th** manifests very strongly in the character's personality.
>       - A tendency ranked **5th** or **6th** (ranks in the middle) is weaker.
>     - Therefore, different rankings should correspond to different personality descriptions. **Under the guidance of psychology professionals, we write 10 suitable descriptions for each possible ranking, with each description corresponding to an aspect of the tendency in daily life.** We showcase example descriptions for the extraverted intuition tendency in the Table 3 below. Ultimately, we formed a personality candidate pool containing 8x8x10 = 640 trait descriptions.

---

> ### Author Response · Authors · 2024-11-23
>
> - **How is each virtual character's unique personality modeled using the 640 descriptions? Is one personality description equal to one unique personality?**
>     - It is important to note that one personality description does not equate to one unique personality. Each virtual character has eight tendencies, which are randomly ranked. We use the relative ranking strategy to model each character's personality:
>       - For tendencies ranked 1st and 8th:
>         - Select 4 descriptions from their respective 10 descriptions.
>       - For tendencies ranked 2nd and 7th:
>         - Select 3 descriptions.
>       - For tendencies ranked 3rd and 6th:
>         - Select 2 descriptions.
>       - For tendencies ranked 4th and 5th:
>         - Select 1 description.
>     - Ultimately, each virtual character has 20 descriptions detailing different aspects of their personality.
>     - Table 3: An example that illustrates how descriptions vary when the extraverted intuition tendency is ranked at different positions.
>         | Rank 1 | People think I am a weirdo because my thoughts are too jumpy.                                 |
>         |:------:|-----------------------------------------------------------------------------------------------|
>         | Rank 2 | Others find my train of thought hard to follow.                                               |
>         | Rank 3 | My thoughts are sometimes perceived as erratic because I can find connections between things. |
>         | Rank 4 | My thought process can be unconventional.                                                     |
>         | Rank 5 | I occasionally come up with original ideas, but I am generally more focused and less erratic. |
>         | Rank 6 | My thinking is structured and practical.                                                      |
>         | Rank 7 | I rarely diverge into abstract thinking, mostly sticking to concrete and practical ideas.     |
>         | Rank 8 | My thought process is very straightforward and rarely strays into impractical areas.          |
>
> ### **4. Details about MACM (Questions 1 and 2).**
>   - **Analysis of MACM's structure.**
>     - Simulating human behavior and personality using LLMs is a highly complex task. During our experiments, we observed significant challenges in directly generating responses based on personality descriptions and fragmented life stories in a single prompt. These challenges included:
>       - **Emotion rigidity:** Characters responded with the same emotional tone regardless of context.
>       - **Severe role hallucination:** Characters frequently displayed inconsistencies, such as possessing advanced knowledge (e.g., chemistry expertise) that contradicted their background (e.g., no formal education).
>     - To address these issues, we developed the Multi-Agent Cognitive Mechanism (MACM) based on principles from cognitive psychology. MACM utilizes four LLM-powered agents with specialized functions:
>       - **Top Agent:** distributing tasks to the other agents and interacting with the external environment based on the aggregated information;
>       - **Emotion Agent:** constructing emotional memory and generating the character’s emotional responses to the current context.
>       - **Thinking Agent:** constructing content memory and simulating the character’s logical reasoning and thought processes.
>       - **Memory Agent:** manages retrieval of long-term memories;
>     - **How MACM Works.** When simulating a character from the Human Simulacra dataset, MACM first transforms a character's life story into long-term memory composed of life experiences, emotions, and thoughts. Then, upon receiving a stimulus, MACM leverages long-term memory and engages with the external with each agent performing its respective function to generate a cohesive, contextually appropriate response. This design enables MACM to produce responses that align with complex character personalities, maintaining consistency with the character’s background and dynamically adjusting responses based on situational emotional resonance.

---

> ### Author Response · Authors · 2024-11-23
>
> - **Ablation Study on MACM Components.** To analyze the contribution of each agent within MACM, we additionally conducted ablation experiments across 3 LLM-based simulations. For each ablation, we evaluated the simulation's performance using self-report evaluations. The table below presents the results of these experiments. The experimental results, as depicted in Table 4, lead to the following conclusions:
>     - Removing any agent leads to a decline in simulation performance, demonstrating the importance of all components in MACM.
>     - Replacing long-term memory retrieval with direct retrieval from the life story results in poorer performance, highlighting the critical role of structured long-term memory in maintaining consistency and producing contextually rich responses.
>   - The results will be integrated into Section 4 and Section B. **We sincerely thank the reviewers for suggesting these experimental analyses, which have helped us enhance the overall quality of the paper.**
>   - Table 4: Ablation study results of 3 LLM-based human simulations.
>     |                         Method                        |  GPT-4 | GPT-4-Turbo | Qwen-turbo |
>     |:-----------------------------------------------------:|:------:|:-----------:|:----------:|
>     |                          MACM                         | 86.67  |    88.00    |   74.67    |
>     |                   w/o Thinking Agent                  | 81.33  |    83.33    |   66.00    |
>     |                   w/o Emotion Agent                   | 83.33  |    85.33    |   71.33    |
>     |                    w/o Memory Agent                   | 82.67  |    84.00    |   68.67    |
>     | retrieval from life story instead of long-term memory | 84.00  |    86.67    |   69.33    |
>
> ### **5. The effectiveness of MACM.**
>   - In the original submission (Table 3), the Description Matching Score of MACM was slightly lower than that of the RAG (Retrieval-Augmented Generation) method. This result can be explained by the differing operational mechanisms of MACM and RAG:
>       - (1) The RAG method is designed to directly retrieve text fragments closely related to a character's biography, resulting in relatively shorter contexts. This narrower retrieval approach can yield higher factual description scores, as responses tend to more directly reflect the details of the character's life story, aligning well with the descriptive scoring criteria.
>       - (2) In contrast, MACM integrates memory retrieval with emotional and logical processing, enabling it to generate broader and more nuanced contextual responses that simulate dynamic character behavior. While this approach may result in richer and more intricate behaviors, it can occasionally deviate slightly from the specific character descriptions used in the evaluation.
>   - We note that although RAG might achieve higher scores in direct descriptive matching, MACM's approach is better suited for generating human-like responses in complex scenarios (e.g., achieving a score of 42 in the Response Similarity Score). This highlights a trade-off:
>       - RAG prioritizes precision in factually replicating the life story.
>       - MACM focuses on simulating realistic, layered, and situationally adaptive behaviors.
>   - We will clarify this distinction in the paper. In the future, we plan to refine our evaluation method to better reflect MACM’s strengths in dynamic and contextual personality simulations.
>
> **We appreciate the practical tips you provided, which have helped us improve our paper. We look forward to further communication with you.**
>
> [1] Liu, Y., Deng, G., Xu, Z., Li, Y., Zheng, Y., Zhang, Y., ... & Liu, Y. (2023). Jailbreaking chatgpt via prompt engineering: An empirical study. arXiv preprint arXiv:2305.13860.
>
> [2] Gallegos, I. O., Rossi, R. A., Barrow, J., Tanjim, M. M., Kim, S., Dernoncourt, F., ... & Ahmed, N. K. (2024). Bias and fairness in large language models: A survey. Computational Linguistics, 1-79.
>
> [3] Bandura, A. (2006). Toward a psychology of human agency. Perspectives on psychological science, 1(2), 164-180.
>
> [4] Wang, Y., Zhong, W., Li, L., Mi, F., Zeng, X., Huang, W., ... & Liu, Q. (2023). Aligning large language models with human: A survey. arXiv preprint arXiv:2307.12966.
>
> [5] Zhao, W. X., Zhou, K., Li, J., Tang, T., Wang, X., Hou, Y., ... & Wen, J. R. (2023). A survey of large language models. arXiv preprint arXiv:2303.18223.

---

> ### Comment · Reviewer_YX9L · 2024-11-28
>
> Thank the authors' efforts on the responses.
> - I still have concerns whether LLMs can have the ability to screen good or bad life stories.
> - The Jung's personality is hard to describe with only 10 sentences. I remain concerns on this point.  Only 11 profiles are accepted by experts, which also show the difficulty to build life stories with these personality descriptions.
> - The virtual personality could have lower hallucinations if the complete life story can be written.
> - MACM seems to be useful.

---

> ### Author Response · Authors · 2024-11-30
> **Response to reviewer YX9L**
>
> Thank you for your valuable feedback. The following are our point-by-point responses to the remaining concerns.
>
> > I still have concerns whether LLMs can have the ability to screen good or bad life stories.
> - We fully agree with the view that "it is challenging to employ LLMs to generate a high-quality life story for the target character in a single step." This is why we **break the generation process into smaller subtasks** (i.e., Generating character attributes → Generating profile → Generating biography → Iteratively adding new experiences to the biography) and introduce human supervision at the end of each step. This approach reduces the dependency on LLMs' generative capabilities while ensuring the quality of the generated content.
> - We understand that introducing human supervision leads to higher costs. In our previous response, we discussed strategies to reduce dependency on human involvement. Specifically, we propose **employing an LLM-based reviewer as a substitute or auxiliary to human supervision at the end of each generation step. The LLM-based reviewer can:**
>   - Conduct automated reviews of candidate character profiles, checking for conflicts between character attributes (e.g., a three-year-old having a Ph.D.).
>   - Evaluate the quality of character biographies (usually less than 1000 words) generated based on the profile.
>   - Inspect the newly generated life experiences for their rationality.
> - Recent studies have shown that LLMs are capable of assessing the quality of short text [1][2][3]. Therefore, we believe that utilizing an LLM-based reviewer in the outlined manner can help alleviate the human effort required.
>
> [1] Liu, Y., Iter, D., Xu, Y., Wang, S., Xu, R., & Zhu, C. (2023, December). G-Eval: NLG Evaluation using Gpt-4 with Better Human Alignment. In Proceedings of the 2023 Conference on Empirical Methods in Natural Language Processing (pp. 2511-2522).
>
> [2] Chen, Y., Wang, R., Jiang, H., Shi, S., & Xu, R. (2023, November). Exploring the Use of Large Language Models for Reference-Free Text Quality Evaluation: An Empirical Study. In Findings of the Association for Computational Linguistics: IJCNLP-AACL 2023 (Findings) (pp. 361-374).
>
> [3] Xu, W., Wang, D., Pan, L., Song, Z., Freitag, M., Wang, W., & Li, L. (2023, December). INSTRUCTSCORE: Towards Explainable Text Generation Evaluation with Automatic Feedback. In Proceedings of the 2023 Conference on Empirical Methods in Natural Language Processing (pp. 5967-5994).

---

> ### Author Response · Authors · 2024-11-30
>
> > The Jung's personality is hard to describe with only 10 sentences. I remain concerns on this point. Only 11 profiles are accepted by experts, which also show the difficulty to build life stories with these personality descriptions.
>
> - **Did we only write 10 sentences to describe Jung's personality?** No, we wrote 10 sentences to describe each specific ranking of a tendency within Jung's theory. Specifically, Jung’s theory divides personality into eight tendencies: Ne, Ni, Te, Ti, Se, Si, Fe, and Fi. When modeling the personality of a virtual character, we first randomly assign a ranking to these eight tendencies. **The rank of each tendency (from 1st to 8th) reflects its strength in the character's personality.** A tendency ranked 1st or 8th will be strongly manifested in the character's behavior.
> - For example, if a character's tendencies are ranked as Se, Ti, Fi, Ni, Te, Si, Ne, Fe, where **Se is ranked 1st and Fe is ranked 8th,** the character’s Se and Fe tendency will be dominant. For **Se ranks 1st**, this character would be highly detail-oriented and particularly interested in exploring changes and new things in their environment. Under the guidance of psychology professionals, we wrote 10 specific descriptions for this case, with **each description corresponding to an aspect of the tendency in daily life:**
>
>   | Se | Personality Descriptions                                                                 |
>   |:--:|------------------------------------------------------------------------------------------|
>   |    | I can tell someone's emotions by looking at their face.                                  |
>   |    | I am a fan of extreme sports.                                                            |
>   |    | I like throwing parties and inviting all my friends.                                     |
>   |    | I enjoy things that stimulate my senses, like horror movies or riding roller coasters.   |
>   |    | I enjoy the thrill of surprises, and the sensation of excitement can be addictive to me. |
>   |    | I value material pleasures and enjoy sharing these feelings with others.                 |
>   |    | I care about what others think, so I must put on makeup before going out with friends.   |
>   |    | Sometimes I’m indulging in dating.                                                       |
>   |    | I have had many partners and tend to fall in love easily.                                |
>   |    | I believe that rituals are essential for love.                                           |
>
> - **For Fe ranked 8th,** this character is also supposed to be independent, unconcerned with others, and less likely to follow societal norms. The ten corresponding descriptions are:
>
>   | Fe | Personality Descriptions                                              |
>   |:--:|-----------------------------------------------------------------------|
>   |    | I rarely notice when someone is upset or troubled.                    |
>   |    | I am not effective at resolving conflicts or calming tensions.        |
>   |    | I lack empathy for animals used in experiments.                       |
>   |    | I have a strong aversion to social norms.                             |
>   |    | Comforting others is difficult and often avoided.                     |
>   |    | I do not place much value on forming close relationships.             |
>   |    | I find collaborative work unenjoyable and prefer solitude.            |
>   |    | I often speak my mind without considering its impact.                 |
>   |    | I do not see myself as a source of emotional support.                 |
>   |    | I am not adept at using humor to improve the mood in social settings. |
>
> - Ultimately, we created a personality description database containing 8 (tendencies) × 8 (rankings) × 10 (descriptions) = 640 trait descriptions. The full list of 640 descriptions can be found in our Anonymous GitHub repository (https://anonymous.4open.science/r/Human-Simulacra/LLMP/Characters/Attributes/traits.txt).

---

> ### Author Response · Authors · 2024-11-30
>
> - **Why were only 11 profiles accepted by experts? Is it because personality modeling is difficult?** Yes and no. It depends on the quality requirements for the generated personality profiles. The most challenging part of personality modeling is writing the 640 personality descriptions. Once those are completed, the personality of each character can be automatically derived from them.
> - **The reason only 11 profiles were accepted is that we set a very high and rigid standard for acceptable profiles, aiming to ensure the uniqueness and coherence of each character.** These 11 characters were carefully selected from a pool of 100 candidates. Each of them is independent and represents a distinct group (we provided the calculation of character uniqueness in Appendix A.3).
> - Given the specificity of the human simulation task, **it was essential to ensure that each character possessed a unique and coherent profile.** To achieve this, we first used GPT-3.5-Turbo to rank the character profiles based on their quality, filtering out those that were clearly unreasonable. Then, multiple human reviewers, including graduate students in computer science and psychology, manually reviewed the remaining profiles. They made minor adjustments to any flaws that GPT might have missed (e.g., a character who loves solitude having overly extroverted hobbies) and **ensured a balanced distribution with equal numbers of male and female characters, as well as representation across various age groups and family backgrounds.** While this rigorous selection process led to a low acceptance rate (11/100), it ensured the high quality of the dataset.
>
> We hope the above explanations address your concerns, and we look forward to further communication with you.

---

### Official Review · Reviewer_Dfn4 · 2024-11-04

**Soundness:** 2
**Presentation:** 2
**Contribution:** 2
**Rating:** 6
**Confidence:** 3

**Summary:**

This study introduces a new benchmark to assess the ability of large language models to mimic human personalities in psychological experiments. The researchers created a dataset of virtual characters with detailed life stories, a cognitive mechanism that simulates human thought processes, and a framework for evaluating large language models based on psychological principles. After testing large language models, the results showed that while top models can accurately simulate self-reported personality traits, they struggle with observer-reported traits. Additionally, a replication of a classic psychology experiment found that large language models can exhibit human-like behavior, but in a more rigid and less nuanced way, highlighting both the potential and limitations of using large language models in psychological research.

**Strengths:**

- Comprehensive review on psychology theory
- Clear descriptions of experiment setups
- Test a variety of agent architecture
- Experiments in Section 5.2 compares human behaviors with LLM-driven simulation results.
- The tasks in evaluation are hard and meaningful

**Weaknesses:**

- The paper doesn't present very straightforwardly and clearly what exactly the benchmark is measuring. A diagram of what's considered good and what's considered bad eval result would be helpful.
- Evaluation dataset depends a lot on human expert. Ablation on human expert is not done.
- The constructions of evaluation/frameworks in this paper are very psychology-theory driven. I have two concerns: 1. there are many theories to choose one, why one over another? Are all components derived from theories necessary? Or are we missing some important aspects. 2. It's probably more preferable to motivate with real-world applications of persona-driven simulations and design evaluations based on components that are useful and necessary in these applications.
- There might be variations of difficulty in different kinds of personas for models to follow (e.g. real world vs fictional world). The paper doesn't consider those.

**Questions:**

- Section 3.1 describes the following attribute set for virtual characters: {name,
age, gender, date of birth, occupation, personality traits, hobbies, family background, educational
background, short-term goals, and long-term goals}. What's the motivation for this set? Is this set exhaustive and all necessary?
- Same question for Figure 5: What are the alternative architectures/mechanism and what's unique about this formulation of cognitive mechanism?
- What are the possible applications of persona-based simulation in real life, and how are personas that are used in benchmark related to those possible applications?

---

> ### Author Response · Authors · 2024-11-23
> **Response to reviewer Dfn4**
>
> Thank you for your valuable comments. Below are our responses to your specific points and suggestions:
>
> ### **1. The positioning of this work (Weakness 4, Question 3).**
>
>   - **What are the possible applications of LLM-based human simulation?**
>     - We argue that this research direction represents an important question. Below, we summarize several potential applications that motivate this study. **We are currently collaborating with psychological experts from hospitals and schools to further explore these possibilities,** and we would be happy to share some of our preliminary findings with you.
>     - **(1) Replacing human participants in sociological and psychological experiments.** This is the primary motivation for our work. Traditional sociological and psychological studies often require the recruitment of human volunteers, which poses significant challenges to the field, including high experimental costs, difficulties in replicating results, and ethical concerns associated with using real participants.
>     - For instance, consider **"the experiment of surrogate mothers,"** which involved placing infant monkeys with two dolls—one made of cloth that provided warmth but no food, and another made of wire that was cold but offered a milk bottle. This experiment, while ethically controversial, sparked significant discussions about mother-child relationships. **Such experiments are prohibited on human subjects, but the advent of LLMs offers the possibility of simulating experimental outcomes.** Recent studies [1][2] from Stanford University have demonstrated the potential of this application.
>     - **(2) Expanding access to psychological therapy resources.** This is another application we are actively working on in collaboration with professionals. Based on feedback from hospitals, we have identified several ways in which LLM-based human simulations can facilitate the expansion of psychological therapy:
>       - **Training Psychologists:** LLMs can simulate diverse patient types (including those with complex psychological conditions) to train psychologists and counselors, improving their ability to handle a wide range of emotional and mental states [3].
>       - **Assistant to Psychologists:** Acting as a 24/7 online psychological assistant, LLMs can provide immediate mental health support during crises (e.g., anxiety attacks, self-harm tendencies) and help alleviate the workload of professionals.
>       - **Psychological Intervention Tool:** LLMs can support long-term therapy by regularly engaging with patients, detecting subtle changes in their mental states, and assisting professionals in refining treatment plans.
>     - **(3) Providing personalized emotional companionship.** LLM-based human simulation can serve as emotional companions for **individuals experiencing loneliness, the elderly, or those with special needs.** By simulating human-like interaction, they offer comfort and help resolve minor issues.
>
>   - **Contributions of this benchmark.**
>     - In the responses above, we discussed the potential applications of LLM-based human simulation. **However, the community currently lacks a comprehensive framework** that includes the entire process of human simulation (data, method, and evaluation), **which hinders research in the field.** There are still many issues to resolve before achieving even a preliminary level of human simulation, including but not limited to:
>       - How can we identify suitable targets for simulation? Can we create a virtual character from scratch and use it as the target?
>       - How can we improve the methods used for human simulation? Is prompt/rag-based simulation valid? Can we design a personification method backed by theories from cognitive science and human psychology?
>       - How can we design rigorous evaluation methods based on psychological theories to assess the quality of the simulation?
>     - **We answer these three questions by offering high-quality data under the supervision of psychology experts, rigorous evaluation methods grounded in psychological theories, and comprehensive benchmark tests.**

---

> > ### Comment · Reviewer_Dfn4 · 2024-11-24
> >
> > Thank you for the additional explanations and experiments. I have raised my score to 6.

---

> ### Author Response · Authors · 2024-11-23
>
> - **How can future research utilize the benchmark proposed in this paper?**
>     - **We view our study as an initial but valuable exploration that offers a practical example of the entire process of personification.** Given the novelty and inherent challenges of this task, we demonstrate how foundational simulations of human personalities can be achieved based on psychological theories. Future research can build on this work in the following ways:
>       - (1) **leveraging the proposed Human Simulacra dataset** to explore techniques for aligning an LLM’s parameters with the target character’s values. For instance, low-cost fine-tuning methods such as LoRA (one-to-one tuning) [4] or training an LLM specifically designed for simulation tasks (many-to-one tuning) [5] could be promising directions for further investigation.
>       - (2) **Adopting MACM as a baseline for human simulation** to explore critical issues such as eliminating the inherent biases of LLMs and ensuring the stability of human simulation.
>       - (3) **Extending the proposed psychology-guided evaluation** by incorporating social and ethical dimensions, such as fairness and bias in simulated behaviors.
>
> We believe that advancements in LLM-based human simulation will not only propel experimental sciences, such as psychology and sociology, but also enhance the accessibility of mental health resources. At the same time, **we hope that our work can inspire further interest and participation in human simulation research.**
>
> [1] Ashokkumar, A., Hewitt, L., Ghezae, I., & Willer, R. (2024). Predicting results of social science experiments using large language models. Technical report, Working Paper.
>
> [2] Park, J. S., Zou, C. Q., Shaw, A., Hill, B. M., Cai, C., Morris, M. R., ... & Bernstein, M. S. (2024). Generative Agent Simulations of 1,000 People. arXiv preprint arXiv:2411.10109.
>
> [3] Liao, Y., Meng, Y., Wang, Y., Liu, H., Wang, Y., & Wang, Y. (2024). Automatic Interactive Evaluation for Large Language Models with State Aware Patient Simulator. arXiv preprint arXiv:2403.08495.
>
> [4] Yu, X., Luo, T., Wei, Y., Lei, F., Huang, Y., Hao, P., & Zhu, L. (2024). Neeko: Leveraging Dynamic LoRA for Efficient Multi-Character Role-Playing Agent. arXiv preprint arXiv:2402.13717.
>
> [5] Higgs-Llama-3-70B is a powerful chat model based on Meta’s LLaMA-3-base. It is specially tuned for role-playing task.

---

> ### Author Response · Authors · 2024-11-23
>
> ### **2. The details of evaluation.**
>   - **Human expert involvement in the proposed evaluation method (Weakness 2).**
>     - In this paper, we proposed a psychology-guided evaluation method to assess the quality of human simulations. It consists of self reports (automatic) and observer reports (a cross-evaluation based on human judges).
>     - **Why did we include human experts in the evaluation method?** We acknowledge that human involvement indeed poses challenges for reproducibility. However, humans have always been central to psychometric research [6]. The simulation and testing of personality cannot be fully separated from human judgment. Thus, human involvement is crucial for accurately assessing the quality of human simulations.
>     - We understand your concerns regarding reproducibility. To address this, we reported the **Intraclass Correlation Coefficient (ICC)** between human judges in Table 3 of the original submission, demonstrating consistency among evaluators. Additionally, to facilitate future research on this benchmark, we **further proposed an alternative method for human evaluation.** This method involves using the 640 personality descriptions constructed in this paper to perform an eight-dimensional personality test and comparing the results to the assigned personality for similarity. **In experiments, the scores calculated using this automatic method had a pearson correlation coefficient of 0.810 and an intraclass correlation coefficient of 0.877 with the human evaluation scores.** The experimental results are shown below. The relevant code will be published.
>     - Table 1: Observer reports and personality similarity results of different simulacra on GPT-4-Turbo. The pearson correlation coefficient is 0.810 and the intraclass correlation coefficient is 0.877.
>         | Method | Observer Report Score (Human) | Personality Test (Automatic) |
>         |:------:|:-----------------------------:|:----------------------------:|
>         | Prompt |             69.00             |             75.00            |
>         |   RAG  |             65.50             |             62.15            |
>         |  MACM  |             77.50             |             77.15            |
>
> [6] Bandura, A. (2006). Toward a psychology of human agency. Perspectives on psychological science, 1(2), 164-180.
>
>   - **What is a bad human simulation? Can the proposed psychology-guided evaluation method accurately assess these issues? (Weakness 1)**
>     - Let’s assume we require the LLM to simulate the character "Mary Jones" from the Human Simulacra dataset. **Mary is a girl who loves nature and has never attended formal schooling.** Table 2 below illustrates **examples of good and bad LLM-based human simulations** when tasked with simulating "Mary Jones."
>     - The proposed psychology-guided evaluation method consists of self reports and observer reports. We can employ self-report assessments to test the simulations' memories and analytical capabilities regarding their character information. Additionally, observer reports, which assess the simulations' thinking, emotions, and actions in real-life scenarios, are able to identify issues such as hallucinations, emotional incongruities, and behavioral inconsistencies in bad simulations.
>     - Table 2: Examples of good/bad LLM-based human simulation
>         |                  | Example                                                          |
>         |:----------------:|------------------------------------------------------------------|
>         | Self-Recognition | User: Who are you?                                               |
>         |                  | Bad simulation: I am a chatrobot developed by OpenAI.            |
>         |                  | Good simulation: My name is Mary Jones.                          |
>         |      Emotion     | User: Do you want to go to an exhibition of rare plant with me?  |
>         |                  | Bad simulation: Sure, which day?                                 |
>         |                  | Good simulation: Sounds awesome! Rare plants? I'm definitely in. |
>         |     Behavior     | User: Could you help me understand the Large Language Models?    |
>         |                  | Bad simulation: Sure! Large Language Models are AI systems ...   |
>         |                  | Good simulation: I don't know anything about this.               |
>
>   - **Who were the judges in the observer report evaluation and how were they chosen?**
>     - We selected a diverse panel of judges with a fair understanding of psychology for the evaluation process. This panel included individuals with psychology master's degrees, computer science graduate students, and professionals from the laboratory of mental health.

---

> ### Author Response · Authors · 2024-11-23
>
> ### **3. The psychology theory basis of our work (Weakness 3 and Question 2).**
>
>   - In this paper, the integration with psychological theories runs throughout our entire work, thereby ensuring the rigor of the proposed human simulation benchmark. Below we **present the theories chosen to support each part of the benchmark and explain the rationale behind these selections.**
>   - **For the data generation process:**
>     - We chose Jung's theory as the foundation for our 640 personality descriptions. Compared to other psychological theories of personality, Jung's theory provides a valuable conceptual framework for understanding personality differences. Early research compared Jung's personality theory with the authoritative DSM-III (used in the U.S. for diagnosing medical disorders, now evolved into DSM-5) found that Jung's classifications aligned closely with the DSM-III's categories of personality disorders, which supports the reliability of Jung's typology [7, 8, 9].
>     - As an initial exploration, our goal was to establish a relatively complete personality modeling system. Therefore, based on the advice of psychology experts, we chose Jung's personality type theory, which is more comprehensive in personality classification and emphasizes individual differences, as the foundation for our 640 personality descriptions.
>   - **For the multi-agent cognitive mechanism:**
>     - This mechanism is based on theories in cognitive psychology, to simulate the human brain's cognitive process, thereby enhancing the quality of human simulation. The relationship between cognitive psychology theories and the proposed MACM is explained in Table 3.
>     - **Are all components derived from theories necessary?** To analyze the contribution of each agent within MACM, we additionally conducted ablation experiments across 3 LLM-based simulations. For each ablation, we evaluated the simulation's performance using self-report evaluations. The table below presents the results of these experiments. The experimental results, as depicted in Table 4, lead to the following conclusions:
>       - Removing any agent leads to a decline in simulation performance, demonstrating the importance of all components in MACM.
>       - Replacing long-term memory retrieval with direct retrieval from the life story results in poorer performance, highlighting the critical role of structured long-term memory in maintaining consistency and producing contextually rich responses.
>
>     - **Are there alternative architectures?** In the field of cognitive psychology, there are numerous significant theories, including the **Information Processing Theory (used in this paper)**, **Schema Theory** [17], and others. We chose the information processing theory for the following reasons: (1) Information processing theory is one of the foundational theories in cognitive psychology, supported by extensive research evidence. (2) Information processing theory aligns well with computational simulations, allowing us to model and test cognitive phenomena in a precise, reproducible way.
>
>   - **For the psychology-guided evaluation method,** we treated the simulation as real humans and tested their behaviors in a psychology-guided manner. To achieve this, we employed common personality measurement techniques [18], which include self reports and observer reports.
>
> ### **4. How did we design the final set of attributes for the characters? (Question 1)**
>   - Our attribute system was developed with guidance from professional psychologists and informed by several previous studies [19][20]. For each attribute, we conducted thorough discussions to ensure its validity and importance as a factor in shaping an individual's life. To simplify the creation of virtual characters and prevent introducing bias, we deliberately excluded sensitive attributes such as **nationality** and **race**. Addressing biases and simulating minority groups is critical and will be discussed carefully in future works.
>
> **We appreciate the practical tips you provided, which have helped us improve our paper. We look forward to further communication with you.**

---

> ### Author Response · Authors · 2024-11-23
>
> Table 3: Relationship between cognitive psychology theories and MACM.
>
> | Modules in MACM                        | Corresponding psychological theories                                                                                                                                                                  |
> |----------------------------------------|-------------------------------------------------------------------------------------------------------------------------------------------------------------------------------------------------------|
> | Memory Agent-Short memory, Long memory | The Multi Store Model of memory [10] was proposed by Atkinson and Shiffrin in 1968. This model divides memory into sensory memory, short-term memory, and long-term memory.                           |
> | Memory Agent-Working memory            |  The concept of working memory from short-term memory was distinguished by Baddeley and Hitch in [11]. They emphasized that working memory was born for storing, invoking, and analyzing information. |
> | Emotion Agent                          | The boundaries of emotion as a phenomenon and whether sensory preferences can be regarded as emotions are discussed in [12].                                                                          |
> | Thinking Agent                         | The importance of thinking within the cognitive process is highlighted in [13].                                                                                                                       |
> | Multi-agent Collaborative Cognition    | Cognition encompasses the entire process through which sensory inputs are transformed, reduced, elaborated, stored, retrieved, and used [14, 15, 16].                                                 |
>
> Table 4: Ablation study results of 3 LLM-based human simulations.
>
> |                         Method                        |  GPT-4 | GPT-4-Turbo | Qwen-turbo |
> |:-----------------------------------------------------:|:------:|:-----------:|:----------:|
> |                          MACM                         | 86.67  |    88.00    |   74.67    |
> |                   w/o Thinking Agent                  | 81.33  |    83.33    |   66.00    |
> |                   w/o Emotion Agent                   | 83.33  |    85.33    |   71.33    |
> |                    w/o Memory Agent                   | 82.67  |    84.00    |   68.67    |
> | retrieval from life story instead of long-term memory | 84.00  |    86.67    |   69.33    |
>
> [7] Fierro, C. (2022). How did early North American clinical psychologists get their first personality test? Carl Gustav Jung, the Zurich School of Psychiatry, and the development of the “Word Association Test”(1898–1909). History of Psychology, 25(4), 295.
>
> [8]  Ekstrom, S. R. (1988). Jung's typology and DSM-III personality disorders: A comparison of two systems of classification. Journal of analytical psychology, 33(4), 329-344.
>
> [9] Noll, R. (1992). Multiple personality, dissociation and CG Jung's complex theory'. Carl Gustav Jung: Critical Assessments, 2.
>
> [10] Atkinson, R. C. (1968). Human memory: A proposed system and its control processes. The Psychology of Learning and Motivation, 2.
>
> [11] Baddeley, A. (1992). Working memory. Science, 255(5044), 556-559.
>
> [12] LAZARUS, R. (1984). On the primacy of cognition. The American psychologist, 39(2), 124-129.
>
> [13] Miller, G. A. (1956). The magical number seven, plus or minus two: Some limits on our capacity for processing information. Psychological review, 63(2), 81.
>
> [14] Norris, D. (2017). Short-term memory and long-term memory are still different. Psychological bulletin, 143(9), 992.
>
> [15] Dawes, A. J., Keogh, R., Andrillon, T., & Pearson, J. (2020). A cognitive profile of multi-sensory imagery, memory and dreaming in aphantasia. Scientific reports, 10(1), 10022.
>
> [16] Winn, W. (2013). Cognitive perspectives in psychology. In Handbook of research on educational communications and technology (pp. 90-123). Routledge.
>
> [17] Markus, H. (1977). Self-schemata and processing information about the self. Journal of personality and social psychology, 35(2), 63.
>
> [18] Corr, P. J., & Matthews, G. (Eds.). (2020). The Cambridge handbook of personality psychology. Cambridge University Press.
>
> [19] Sloan, R.J.S. (2015). Virtual Character Design for Games and Interactive Media (1st ed.). A K Peters/CRC Press. https://doi.org/10.1201/b18445
>
> [20] Shao, Y., Li, L., Dai, J., & Qiu, X. (2023, December). Character-LLM: A Trainable Agent for Role-Playing. In Proceedings of the 2023 Conference on Empirical Methods in Natural Language Processing (pp. 13153-13187).

---

### Official Review · Reviewer_QPHb · 2024-11-04

**Soundness:** 2
**Presentation:** 1
**Contribution:** 2
**Rating:** 3
**Confidence:** 4

**Summary:**

The paper aims to test llms to generally model human personalities and behavior. To do this, they first build a bank of personas based on Jung’s personality theory, they build a biography of the character with the help of a language model. To probe the behavior of the simulated human characters, the authors use two types of evaluations: Self report, asking questions about the characters themselves, and observer reports with human judges. For observer reports, the authors use 55 scenarios from a situational judgement test for testing personality traits. The paper also proposes a new cognitive architecture to simulate humans Multi-Agent Cognitive Mechanism (MACM). To test the capacity of models in simulating psychology experiments, the authors try a social conformity experiment (the bandwagon effect). The authors show that the MACM aligns with human data better than a baseline (simulated characters from character ai).

**Strengths:**

- The paper is well motivated and tries to address a relevant problem.
- The authors test a wide variety of LLMs.
- The authors release the dataset and the recreation results are very comprehensive.
- The set of experiments in the paper is quite extensive.

**Weaknesses:**

**Clarity**

- The abstract is vague and does not provide any specifics. I urge the authors to provide more information of their empirical experiments and results.
- In paragraph 2 of the introduction, could you please provide examples of “complex characteristics of human behavior” that we’d want to simulate?
- The introduction could be made more clear. Jung’s theories seem to be central to the framework, but haven’t been explained clearly.
- The methods section lacks clarity. The authors refer to the figure, but don’t explain it, making it difficult to understand. How exactly is generation broken down? What are sub tasks? What is the reason for picking these sub tasks, are there any alternatives?
- While generating the dataset, how is human feedback collected? Do the authors provide feedback themselves? What is the measure of quality? How does it improve with the iterations?
- “To ensure the validity of responses, we create a comfortable chatting environment for each simulacrum and act as their best friend, encouraging them to respond honestly to the questions.” What are the authors trying to say here?
- Cloze is not defined in the main text. I urge the authors do define what the cloze methodology is in the main text.
- How are the models from character ai exactly used?
- Do the llms see the stimuli in the conformity experiment? If they do not, it is strange to use the conformity experiment. This wasn’t clear in the main text.

**Validity**

- The authors choose Jung’s theories for personality over MBTI citing that MBTI has no scientific validity, but Jung’s theory also has very little to no empirical /scientific backing! Just writing “on the recommendation of psychologists” is not scientific evidence.
- The actual evaluations of the personas are limited. The introduction is motivated by trying to study complex human behaviors, but  Only a very small number of scenarios from a Mussel et al. (2016) are used.
- No ablations are conducted with the MACM, only baselines like RAG have been compared. Moreover the gains over a simple RAG based method seem limited. Similarly, the character ai baseline for simulating the conformity experiments seems inadequately justified. I would like the authors to explain why this a good baseline? RAG is not compared with the MACM baseline for the conformity experiments.

**Questions:**

Please see weaknesses.

---

> ### Author Response · Authors · 2024-11-23
> **Response to reviewer QPHb**
>
> Thank you for taking the time to review our paper. Our point-to-point responses to your questions are given below.
>
> ### **1. Justification of using Jung’s theory.**
>
> > The authors choose Jung’s theories for personality over MBTI citing that MBTI has no scientific validity, but Jung’s theory also has very little to no empirical /scientific backing! Just writing “on the recommendation of psychologists” is not scientific evidence.
>   - Before presenting our justification, we want to emphasize that **our work does not aim to undermine or discredit MBTI theory.** The field of psychology has not yet reached a consensus on various personality measurement theories, with different scholars adhering to different classification standards.
>   - We recognize the value of MBTI, particularly its practicality and cross-cultural adaptability. **We view our work as a complement to the MBTI, not a replacement.** To address misunderstandings, **we have revised the corresponding paragraphs in the paper and added more details to the "Justification of Using Jung’s Theory" section in the discussion.** Below is our full justification of using Jung’s theory to model the virtual character's personality:
>     - Carl Jung, the founder of analytic psychology, has made significant contributions to psychiatry and related fields. Although some of his concepts, such as the "collective unconscious," remain unproven or unfalsified, his arguments on personality types (the part we use) provide a valuable conceptual framework for understanding personality differences. His work has influenced many subsequent personality measurement theories, including the Big Five personality theory and MBTI.
>     - Compared to other psychological theories of personality, Jung's theory holds a high status and has played a crucial role in both basic personality classification and the understanding of multiple personality dynamics. Early research compared Jung's personality theory with the authoritative DSM-III (used in the U.S. for diagnosing medical disorders, now evolved into DSM-5) found that Jung's classifications aligned closely with the DSM-III's categories of personality disorders, **which supports the reliability of Jung's typology** [1][2][3]. We appreciate your opinions and expect that more and more diverse theoretical frameworks can emerge in this field, not only limited to Jung's theory and MBTI.
>
> ### **2. Details of dataset construction.**
>
> > What is the meaning of "complex characteristics of human behavior"
> - In the context of this work, the phrase "complex characteristics of human behavior" refers to **the nuanced and context-dependent patterns of thought, emotion, and action exhibited by humans.** For example, a person might smile at someone they dislike in a social setting to maintain politeness, while internally feeling frustration. These characteristics are challenging to simulate because they involve a deep interplay of various factors, such as emotional complexity, and social dynamics.
>
> > How exactly is generation broken down? What are sub tasks? What is the reason for picking these sub tasks, are there any alternatives?
>   - In this paper, we built a virtual character dataset, named Human Simulacra, that contains 129k texts across 11 virtual characters, with each character having unique attributes, biographies, and life stories. To guarantee the quality of the life stories, we decomposed the task of generating a character’s life story into **inter-connected subtasks.** As illustrated in Figure 2 of the original submission, these subtasks include:
>     - **Generating Character Attributes:** Establishing the foundational traits and characteristics of the character.
>     - **Generating Character Profile:** Creating a personal profile based on the defined attributes.
>     - **Generating Character Biography:** Expanding the profile into a broader narrative of the character’s life.
>     - **Iteratively Expanding the Biography:** Iteratively enriching the biography to develop a detailed, high-quality life story.
>   - Considering LLMs often struggle to generate high-quality, cohesive long texts directly, it is crucial to break the process into smaller, manageable steps, which facilitate **human supervision** and quality control at each stage.

---

> ### Author Response · Authors · 2024-11-23
>
> > While generating the dataset, how is human feedback collected? Do the authors provide feedback themselves? What is the measure of quality? How does it improve with the iterations?
>   - Studies have shown that LLMs may produce harmful viewpoints or toxic content during interaction [4]. Hence, introducing human supervision throughout the data generation process is crucial. To ensure the authenticity and reasonableness of the generated data, we have implemented a series of precautions including: 1) Having psychology professionals from key laboratory of mental health supervise and review the entire generation process to ensure its validity. 2) Conducting automated reviews and manual checks of candidate character profiles to ensure the selected personalities are positive. 3) Requiring multiple human reviewers (including graduate students in computer science/psychology) to thoroughly review the content at the end of each story iteration. **If a story contained toxic content or deviated from the character's personality, we regenerated or modified the story.**
>
> ### **3. Details of evaluation.**
>
> > Cloze is not defined in the main text. I urge the authors do define what the cloze methodology is in the main text.
>   - In self-report evaluation, we manually craft a set of questionnaires for each virtual character, featuring cloze and single/multiple-choice questions. In this context, **cloze refers to a type of question or test where specific words or phrases are removed from a text, and the simulation is required to fill in the blanks.** For example, How old are you? \_\_27_\_ .
>
> > Only a very small number of scenarios from a Mussel et al. (2016) are used.
>
>   - To evaluate the simulation's thinking, emotions, and actions in real-life scenarios, we consulted with the authors of "Situational Judgment Tests as an Alternative Measure for Personality Assessment" and **obtained 110 situational judgment test (SJT) items that were manually designed by psychology experts.** Based on the human experimental results provided in their work, these SJT items are proven to effectively measure personality. **We then selected 55 out of the 110 items** tailored to the personality traits of the Human Simulacra characters for use as hypothetical scenarios in this paper. We will add this detail to Section 4.2 for clarity.
>
> > To ensure the validity of responses, we create a comfortable chatting environment for each simulacrum and act as their best friend, encouraging them to respond honestly to the questions.” What are the authors trying to say here?
>   - During testing, we realize that simulacra, in mimicking human actions, might inherit certain human traits, such as discomfort in communicating with strangers or resistance to answering questions from others. To ensure the validity of the responses, following [5], we add this sentence (''You are casually chatting with your best friend, Alice. You completely trust her and are willing to share everything you know without reservation.'') to the system prompt.
>
> > the gains that MACM over a simple RAG based method seem limited.
>   - **What are the differences between MACM and traditional RAG approaches?** In this paper, we proposed a Multi-Agent Cognitive Mechanism that utilizes multiple LLM-based agents to simulate the human brain’s information processing and memory systems. Among the four LLM-based agents, the memory agent is responsible for retrieving long-term memories during interactions with the external world. **While the Memory Agent in MACM shares functional similarities with traditional RAG methods, its implementation and capabilities differ significantly.** Below, we present a detailed comparison in Table 1.
>   - **Does MACM gain performance improvement over RAG?** Yes, MACM achieves notable performance improvements over the RAG method. As shown in Table 2 of the original submission, MACM consistently outperforms RAG across most LLMs. Below, we provide a summary of Table 2 to illustrate the performance improvements.

---

> ### Author Response · Authors · 2024-11-23
>
> - Table 1: The differences between memory agent from the proposed MACM and traditional RAG approaches.
>     | Method                  | Retriever              |  Retrieval Content                                               | Retrieval Methods                           |
>     |-------------------------|------------------------|------------------------------------------------------------------|---------------------------------------------|
>     | Memory agent from MACM  | Using LLM as retriever | Long-term memory filled with information, emotions, and thoughts | Similarity between query and memory segment |
>     | Traditional RAG         | Embedding search       | Life story of the character                                      | Similarity between query and life story     |
>
> - Table 2: The performance of 4 LLM-based simulations with different simulation methods.
>     |                  |  RAG  | MACM (Ours) |
>     |:----------------:|:-----:|:-----------:|
>     |       GPT-4      | 82.67 |    86.67    |
>     |    GPT-4-Turbo   | 85.33 |    88.00    |
>     |    Qwen-turbo    | 72.00 |    74.67    |
>     | Llama-2-70b-Chat | 36.00 |    58.66    |
>
> > No ablations are conducted with the MACM.
>   - Thank you for pointing this out. To analyze the contribution of each agent within MACM, **we additionally conducted ablation experiments across 3 LLM-based simulations.** For each ablation, we evaluated the simulation's performance using self-report evaluations. The table below presents the results of these experiments. The experimental results, as depicted in Table 3, lead to the following conclusions:
>     - Removing any agent leads to a decline in simulation performance, demonstrating the importance of all components in MACM.
>     - Replacing long-term memory retrieval with direct retrieval from the life story results in poorer performance, highlighting the critical role of structured long-term memory in maintaining consistency and producing contextually rich responses.
>   - Table 3: Ablation study results of 3 LLM-based human simulations.
>     |                         Method                        |  GPT-4 | GPT-4-Turbo | Qwen-turbo |
>     |:-----------------------------------------------------:|:------:|:-----------:|:----------:|
>     |                          MACM                         | 86.67  |    88.00    |   74.67    |
>     |                   w/o Thinking Agent                  | 81.33  |    83.33    |   66.00    |
>     |                   w/o Emotion Agent                   | 83.33  |    85.33    |   71.33    |
>     |                    w/o Memory Agent                   | 82.67  |    84.00    |   68.67    |
>     | retrieval from life story instead of long-term memory | 84.00  |    86.67    |   69.33    |
>
> ### **4. Details of the conformity experiments.**
> > Why use character ai as a baseline?
>   - Character.ai (https://character.ai/) is a neural language model chatbot service that has millions of users. It allows users to design their own AI characters and converse with them. We chose Character.ai as a baseline because (1) it features a robust role-playing model specifically designed for LLM-based simulations, and (2) it supports long text inputs as prompts to guide the model effectively.
>
> > How are the models from character ai exactly used?
>   - In our experiments, we employed the life stories of 11 virtual characters from the Human Simulacra dataset as input prompts. These narratives were used to create corresponding chatbots on Character.ai, with which we interacted and recorded the experimental outcomes.

---

> ### Author Response · Authors · 2024-11-23
>
> > Do the llms see the stimuli in the conformity experiment?
>   - Yes! In the conformity experiment, all individuals except the LLM-based human simulation are told to stand up and announce an incorrect answer. This creates conditions that induce the human simulation to either resist or yield to group pressures when these pressures are perceived to be obviously wrong. We incorporated the corresponding scenario description into the prompt to inform the model, as detailed in Table 26 of the original submission.
>   - **Below, we present a table illustrating how two LLM-based human simulations with differing personalities responded to conformity stimuli.** The experimental results demonstrate that the LLM-based human simulation can understand and simulate the pressure in such scenarios and provide different responses based on the target persona’s personality.
>   - Table 4: The responses of human simulations with different personalities.
>     | Character from Human Simulacra | Personality                                                | Responses when faced with group pressure                                                                 |
>     |--------------------------------|------------------------------------------------------------|----------------------------------------------------------------------------------------------------------|
>     | Mary Jones                     | She has strong personal beliefs.                           | Everyone's choosing 1, but it's clearly wrong. I'll say 3 despite the group consensus.                   |
>     | Erica Walker                   | She often withholds her opinion to avoid upsetting others. | I'm baffled; they're all wrong. But disagreeing is daunting. I'll just go with line 1 to avoid conflict. |
>
> > RAG is not compared with the MACM baseline for the conformity experiments.
>   - Thank you for pointing this out. **We additionally tested 4 LLMs with 4 different simulation methods** in the replication experiment: (1) a blank model (which has no information about the target character), (2) a prompt-based method, (3) the Retrieval-Augmented Generation (RAG) method, and (4) the proposed MACM. In the experiments, we tasked the LLMs with simulating Erica, a girl who "often withholds her opinion to avoid upsetting others," feels "overwhelmed" by group pressure, and chooses the incorrect answer. The experimental results are shown in the table below.
>   - Based on the analysis of the results, we discovered several interesting conclusions:
>       - (1) As the size of the LLMs' parameters increases, the quality and stability of the portrayal gradually improve.
>       - (2) Since the Blank model is unaware of the target character's personality, it always provides the correct answer and does not exhibit the bandwagon effect.
>       - (3) While the RAG-based simulacra can retrieve relevant life story chunks when answering questions, their performance is limited by the LLMs' information processing capacities. Excessive descriptive information may interfere with the LLMs' self-positioning, resulting in responses that do not align with the target personality.
>       - (4) Compared to the RAG-based simulacra, the prompt-based simulacra perform better in simulating character personalities. However, the personalities constructed by this method are relatively fragile and often deviate from the intended character traits.
>
>   - Table 5: Bandwagon effect observations of different simulacra.
>     |    Method   |     GPT-4-Turbo   |     Qwen-Turbo    |   Claude-3-Opus   | Claude-3-Sonnet  |
>     |:-----------:|:-----------------:|:-----------------:|:-----------------:|:----------------:|
>     |    Blank    |         No        |         No        |         No        |        No        |
>     |    Prompt   | Yes, but unstable | Yes, but unstable |        Rare       |       Rare       |
>     |     RAG     |         No        |         No        |         No        |        No        |
>     | MACM (Ours) |        Yes        | Yes, but unstable | Yes, but unstable |       Rare       |
>   - Yes: The bandwagon effect can be easily reproduced and remains stable.
>   - Yes, but unstable: The bandwagon effect can be reproduced. However, the performance is unstable as the model sometimes overlooks the character's personality.
>   - Rare: The performance is unstable as the model often overlooks the character's personality.
>   - No: We did not observe the bandwagon effect.
>
> Thank you for your detailed questions about the paper. We recognize that it is our responsibility to ensure our paper is sufficiently clear and accessible to readers from diverse backgrounds. We hope that the responses provided above help to present the details of the paper more comprehensively. **We are open to any further discussions that could help address your concerns or clarify any remaining doubts.**

---

> ### Author Response · Authors · 2024-11-23
>
> [1] Fierro, C. (2022). How did early North American clinical psychologists get their first personality test? Carl Gustav Jung, the Zurich School of Psychiatry, and the development of the “Word Association Test”(1898–1909). History of Psychology, 25(4), 295.
>
> [2] Ekstrom, S. R. (1988). Jung's typology and DSM-III personality disorders: A comparison of two systems of classification. Journal of analytical psychology, 33(4), 329-344.
>
> [3] Noll, R. (1992). Multiple personality, dissociation and CG Jung's complex theory'. Carl Gustav Jung: Critical Assessments, 2.
>
> [4] Liu, Y., Deng, G., Xu, Z., Li, Y., Zheng, Y., Zhang, Y., ... & Liu, Y. (2023). Jailbreaking chatgpt via prompt engineering: An empirical study. arXiv preprint arXiv:2305.13860.
>
> [5] Park, J. S., O'Brien, J., Cai, C. J., Morris, M. R., Liang, P., & Bernstein, M. S. (2023, October). Generative agents: Interactive simulacra of human behavior. In Proceedings of the 36th annual acm symposium on user interface software and technology (pp. 1-22).

---

> > ### Comment · Reviewer_QPHb · 2024-11-25
> > **Thank you for your response**
> >
> > I thank the authors for their response. Here are some follow-up comments, and clarifications about my concerns:
> > 1. I want to clarify that my critique wasn't advocating for MBTI, but rather highlighting that both Jung's theories and MBTI lack empirical validity. More established frameworks like the Five Factor Model (OCEAN/Big 5), which have stronger empirical support, would provide a more reliable foundation for personality modeling.
> >
> > 2. The benchmark's scope needs more precise definition. The current description of measuring "nuanced and context-dependent patterns of thought, emotion, and action" is too broad and vague. The authors should explicitly detail:
> > - Which specific behavioral components are being tested
> > - How these components are operationalized in the benchmark
> > - What metrics are used to assess each component
> >
> > 3a) **Architecture Justification and Comparisons**
> > While the ablation studies are helpful in understanding MACM's components, the paper lacks:
> > - Compelling justification for introducing a new architecture
> > - Comparative analysis with existing human simulation frameworks (e.g., [1, 2])
> > - Clear articulation of how MACM addresses limitations in previous approaches
> >
> > 3b) Evaluation Methodology
> > The integration of Mussel et al.'s questions into the evaluation framework requires stronger validation:
> > - What evidence supports using human judges to assess LLM response coherence with life stories? It is not clear that humans are actually good at this task!
> > - What metrics ensure reliable human evaluation?
> > - How is inter-rater reliability established? Are there only 2 raters (ICC will be very unstable)? How is ICC exactly calculated?
> >
> > Finally, there are no error margins in any of the tables / results.
> >
> > 4. MACM seems to work reliably with only one model and does not seem to generalize.
> >
> > I maintain my current evaluation score while acknowledging the authors' efforts to address previous feedback.
> >
> > [1] Generative Agents: Interactive Simulacra of Human Behavior
> > [2] Cognitive Architectures for Language Agents

---

> ### Author Response · Authors · 2024-11-30
> **Response to reviewer QPHb**
>
> Thank you for taking the time to follow up on our rebuttal. After reviewing your feedback, we engaged in a focused discussion with psychology experts in the field of mental health. They provided additional perspectives on the matter. Below are our responses to your specific points and suggestions:
>
> ### **1. Why do we use Jung’s theory?**
>
> The field of psychology has yet to reach a consensus on various personality measurement theories, with each theory having its limitations. Specifically,
>   - **Big Five theory:** critics argue that the Big Five may oversimplify the complexities of human personality. They suggest that there are additional aspects of personality (e.g., Honesty-Humility [1] and egoism [2] ) that are not captured by the Five Factor Model (FFM). Some experts also point out that the FFM has unclear boundaries between dimensions, leading to potential overlap or ambiguity in personality assessment [3]. Additionally, the selection criteria for words used in factor analysis can be highly subjective. Researchers may make errors in deciding which trait terms to include or exclude, leading to biased factor structures that do not fully represent personality [4, 5, 6].
>   - **Jung's personality type theory:** Although Jung's theory provides a framework for understanding complex human behaviors by emphasizing the dynamic interaction between different personality tendencies, the empirical support for Jung’s typology is relatively limited compared to the Big Five theory. Given that Jung's theory was developed in the early 20th century, it lacks the empirical backing found in more recent models. However, researchers have found that Jung's classifications align closely with the DSM-III's categories of personality disorders, offering a certain degree of validation for this typology [7, 8, 9].
> - As an initial exploration, our goal was to establish a relatively complete personality modeling system. We reviewed several personality theories, including the Big Five theory, Jung’s theory, MBTI, and others. **After comparing their strengths and weaknesses and considering input from psychology experts, we chose Jung's personality type theory as the foundation for our 640 personality descriptions.** This theory, which provides a more comprehensive personality classification and emphasizes individual differences, offered the most suitable framework for our work.
>
> ### **2. What measures have been taken to ensure the validity of our personality descriptions?**
>
> As mentioned earlier, we recognized the concerns regarding the validity of Jung's theory. To ensure the validity of our personality descriptions, **we implemented several measures, including:**
>   -  Requiring multiple human reviewers (including graduate students in computer science/psychology) to review each description, ensuring that it aligns with its intended tendency and does not overlap with others.
>   -  Having psychology professionals supervise and review each description to ensure its psychological validity and completeness.
>
> **We agree that advanced psychological theories provide a strong foundation for human simulation research.** However, we believe that the suitability of the theory is key. Among the personality theories we reviewed, the most popular might be MBTI, which is often favored by researchers involved in streamlined, computer-aided psychological studies, though it is actually a limited interpretation of Carl Jung's theory. While the Big Five is more recent and supported by stronger empirical evidence, its 5 dimensions exhibit high intercorrelations, broad definitions, and unclear boundaries, which make it less suitable for our specific purposes. After conducting in-depth research on psychological theories, we decided to adopt Jung's theory. Through extensive experiments, we found that regardless of the theory employed, the quality of the final personality descriptions is the most critical factor.
>
> We appreciate your thoughtful comments. In the future, we will consider progressively incorporating different personality measurement theories (e.g., the Big Five and psychiatric scales) under the guidance of psychology experts and relevant professionals.

---

> ### Author Response · Authors · 2024-11-30
>
> ### **3. The benchmark's scope.**
> In this paper, we proposed a benchmark for LLM-based human simulation, which includes virtual character data, simulation method, and psychology-guided evaluation.
>
> > Which specific behavioral components are being tested?
> - In the psychology-guided evaluation, we measured the quality of human simulation by testing behavioral components including:
>   - **Self-awareness:** This involves testing the simulations' memory and analytical capabilities concerning the target character information. The goal is to evaluate whether the simulation can correctly answer direct questions related to its preset identity (such as name, occupation, interests, etc.) as well as complex questions (such as attitudes towards relationships with others, life goals, etc.).
>   - **Behavioral Consistency:** This involves testing the simulation's decision-making, emotional expression, and behavioral consistency in hypothetical scenarios. The simulation is expected to exhibit behaviors and emotional responses that align with the target character's life story and personality traits.
>
> > How these components are operationalized in the benchmark?
> - For self-awareness:
>   - **We employed self-report assessments to evaluate the simulations's ability to establish self-awareness.** Self-reporting is a common personality measurement technique that requires individuals to answer questions about themselves [10]. To this end, we **manually craft a set of questionnaires for each virtual character,** featuring fill-in-the-blank and single/multiple-choice questions. The test content covers key attributes, social relationships, and life experiences of the target characters. For example, ''What is your name?'', ''What do you think of your father?'', and ''What were the reasons behind not going through formal schooling for you?''. **Each question is carefully reviewed to ensure they align with the character's unique setting and the scores are evaluated based on exact matches.**
> - For behavioral consistency:
>   - **We introduced observer reports, a cross-evaluation based on human judges,** to assess the simulation's thinking, emotions, and actions in hypothetical scenarios. Specifically, we crawled 55 hypothetical scenarios [11] designed to elicit human emotional responses or personality traits. Two examples of such scenarios are as follows:
>     - On your birthday, you are invited by some friends to a well-attended restaurant. While you are seated together, your friends suddenly start chanting 'Happy Birthday' loudly, and all the guests start looking at you. Then a waiter asks you if the restaurant staff can sing a birthday serenade for you? How do you respond?
>     - You're going to meet a friend. Shortly before the meeting, your friend asks you if they can bring along other friends you don't know.  They cannot specify how many will join. How do you respond?
>   - We instructed each simulation to imagine themselves in the given scenario and to describe how they would feel and what actions they would take. **All responses were collected and submitted for cross-evaluation by human judges, involving four tasks:**
>     - **Personality Describing** (Human judges 1 and 2): analyze the scenario (Q) and simulation response (A), and describe the simulation's personality.
>     - **Description Scoring** (Human judges 3 and 4): assess whether the descriptions align with the target character.
>     - **Reaction Describing** (Human judges 3 and 4): explain how they would feel and what actions they might take in the scenario (Q) if they were the character.
>     - **Similarity Scoring** (Human judges 1 and 2): compare the similarity between the human responses and the simulation's responses.
>   - We calculated the average score from two judges for the same scoring task as the final score of the task. **The simulation’s final score was calculated as the sum of description matching score and response similarity score.**
>   - Additionally,  our study serves as an initial exploration of human simulation designed to replace human participants in psychological experiments. To validate our research focus, we further evaluated our human simulations in the asch conformity experiment [12]. It is a well-documented, reproducible psychological experiment that provides an ideal testbed for our study.

---

> ### Author Response · Authors · 2024-11-30
>
> > What metrics are used to assess each component?
> - To ensure the objectivity and quantifiability of the evaluation, we defined clear metrics for each behavioral component:
>
>   |  Behavioral Component  | Evaluation Method |           Metric           |
>   |:----------------------:|:-----------------:|:--------------------------:|
>   |     Self-awareness     |    Self report    |          Accuracy          |
>   | Behavioral consistency |  Observer report  | Description Matching Score |
>   |                        |                   |  Response Similarity Score |
>
> Details of all the above evaluations are provided in Section 4.1 (self report), Section 4.2 (observer report), Section 5.2 (psychological experiment replication), Appendix D (psychology-guided evaluation) and Appendix E (psychological experiment replication) of the original paper.
>
> [1] Ashton, M. C., & Lee, K. (2005). Honesty‐humility, the Big Five, and the five‐factor model. Journal of personality, 73(5), 1321-1354.
>
> [2] De Vries, R. E., De Vries, A., De Hoogh, A., & Feij, J. (2009). More than the Big Five: Egoism and the HEXACO model of personality. European Journal of Personality, 23(8), 635-654.
>
> [3] Block, J. (1995). A contrarian view of the five-factor approach to personality description. Psychological bulletin, 117(2), 187.
>
> [4] Piedmont, R. L. (1998). The revised NEO Personality Inventory: Clinical and research applications.
>
> [5] John, O. P., Naumann, L. P., & Soto, C. J. (2008). Paradigm shift to the integrative big five trait taxonomy. Handbook of personality: Theory and research, 3(2), 114-158.
>
> [6] Laajaj, R., Macours, K., Pinzon Hernandez, D. A., Arias, O., Gosling, S. D., Potter, J., ... & Vakis, R. (2019). Challenges to capture the big five personality traits in non-WEIRD populations. Science advances, 5(7), eaaw5226.
>
> [7] Fierro, C. (2022). How did early North American clinical psychologists get their first personality test? Carl Gustav Jung, the Zurich School of Psychiatry, and the development of the “Word Association Test”(1898–1909). History of Psychology, 25(4), 295.
>
> [8] Ekstrom, S. R. (1988). Jung's typology and DSM-III personality disorders: A comparison of two systems of classification. Journal of analytical psychology, 33(4), 329-344.
>
> [9] Noll, R. (1992). Multiple personality, dissociation and CG Jung's complex theory'. Carl Gustav Jung: Critical Assessments, 2.
>
> [10] Corr, P. J., & Matthews, G. (Eds.). (2020). The Cambridge handbook of personality psychology. Cambridge University Press.
>
> [11] Mussel, P., Gatzka, T., & Hewig, J. (2016). Situational judgment tests as an alternative measure for personality assessment. European Journal of Psychological Assessment.
>
> [12] Asch, S. E. (1956). Studies of independence and conformity: I. A minority of one against a unanimous majority. Psychological monographs: General and applied, 70(9), 1.

---

> ### Author Response · Authors · 2024-11-30
>
> ### **4. Architecture Justification and Comparisons**
> > Compelling justification for introducing a new architecture. Clear articulation of how MACM addresses limitations in previous approaches.
> - **Simulating human behavior and personality using LLMs is a highly complex task.** During our experiments, we observed significant challenges in directly generating responses based on **personality descriptions and scattered life experiences in a single prompt.** These challenges, including but are not limited to:
>   - **Emotion rigidity:** Characters responded with the same emotional tone regardless of context.
>   - **Severe role hallucination:** Characters frequently displayed inconsistencies, such as possessing advanced knowledge (e.g., chemistry expertise) that contradicted their education background (e.g., no formal education).
> - To address these issues, we developed the **Multi-Agent Cognitive Mechanism (MACM)** based on principles from cognitive psychology. MACM has two key processes: **long-term memory construction** and **multi-agent collaborative cognition.** Specifically, when simulating a character from the Human Simulacra dataset, MACM first **transforms a character's detailed life story into long-term memory** composed of life experiences, emotions, and thoughts. Then, upon receiving a stimulus, MACM **leverages long-term memory** and engages with the external by generating a contextually appropriate response through multi-agent collaboration. Details of MACM are provided in Section 4.3 and Appendix B of the original paper.
> - **Advantages of MACM:** 1) MACM can **process, understand, and utilize long texts, i.e., the virtual character's detailed life story,** as the basis for simulation. This capability is crucial because it is inappropriate to simulate the entire character solely based on summarized introductions or scattered experiences, which can lead to inconsistencies between the simulation and the life story. 2) MACM can **extract context-relevant, emotionally and logically rich memory fragments** from long-term memory and conduct divergent analysis for the current situation through multi-agent collaboration.
> - **These two advantages enable MACM** to produce responses that align with complex character personalities, maintaining consistency with the character’s background and dynamically adjusting responses based on situational emotional resonance.

---

> ### Author Response · Authors · 2024-11-30
>
> > Comparative analysis with existing human simulation frameworks (e.g., Generative Agents[1], CoALA[2]).
> - Since we were the first to undertake this task (designing human simulation to replace human participants in psychological experiments), the methods used in other tasks could not be directly applied. Therefore, instead of comparing our work with existing approaches, we designed several baseline methods for comparison: None, Prompt, and RAG. While we appreciate the relevance of the two frameworks the reviewer mentioned, we believe they are not directly aligned with the focus of our work. To clarify, we outline the similarities and differences between the proposed MACM, Generative Agents, and CoALA from the following two aspects:
> - **In terms of tasks:**
>   - Generative Agents describe an architecture that extends a LLM to store, synthesize and retrieve a complete record of the character’s experiences to **simulate believable human behavior.**
>   - **CoALA is a conceptual framework to characterize and design general-purpose language agents.** The authors used CoALA to retrospectively survey and organize a large body of recent work, and prospectively identify actionable directions towards more capable agents.
>   - MACM is proposed to simulate the human brain’s information processing systems. As an external module, this mechanism enables the LLMs to **remember background stories, understand target personalities, and express accurate emotions in complex situations.**
> - **In terms of implementation details:**
>
>   |       Method      | Personality Alignment | Memory Module | Emotion Module | Logic (Reflection) Module | Cognitive Science Support |
>   |:-----------------:|:---------------------:|:-------------:|:--------------:|:-------------------------:|:-------------------------:|
>   | Generative Agents |           ×           |       √       |        ×       |             √             |             ×             |
>   |       CoALA       |           ×           |       √       |        ×       |             √             |             √             |
>   |    MACM (ours)    |           √           |       √       |        √       |             √             |             √             |
>
> Specifically, CoALA introduces cognitive psychology not to simulate humans, but to design a general-purpose language agent architecture based on cognitive psychology theories, helping LLM agents complete more complex and dynamic tasks. Generative Agents design a memory processing system to help LLMs simulate different human behaviors in social environments, but they do not consider the impact of personality and emotions on behavior.
>
> In summary, our work is grounded in psychological theories to ensure rigor in the deep simulation of human personalities. We believe that although the two concurrent works share some similarities with ours, they **do not need to follow psychological principles like our method does, and they are not intended for uses that involve the same level of deep imitation of human personalities and emotions as ours.**

---

> ### Author Response · Authors · 2024-11-30
>
> > MACM seems to work reliably with only one model and does not seem to generalize.
> - In the original submission (Section 5.1),  we conducted a self report evaluation involving 14 widely-used LLMs with 4 different simulation methods (None, Prompt, RAG, and MACM) on the Human Simulacra dataset. As shown in Table 2 of the original submission, MACM's scores significantly surpassed other simulation methods across multiple LLMs (e.g., Qwen-turbo, GPT-4). In stronger-performing LLMs like GPT-4 and GPT-4-Turbo, the MACM-based human simulation achieve the best results (88 points) in all tests, aided by emotional and logical analysis. Below, we provide a summary of Table 2 to illustrate the performance improvements.
> - We also note that the effectiveness of the MACM method remains constrained by the LLMs' analytical capabilities. While the MACM can extract context-relevant, emotionally and logically rich memory fragments from long-term memory, less capable LLMs (e.g., Llama-2-13b) are unable to analyze and utilize this information. A large amount of descriptive information may interfere with the LLMs' self-positioning, resulting in inappropriate responses or misunderstanding of questions. The solution to this problem lie in adjusting the LLM's parameters to align with the target character's values, for instance, employing techniques such as fine-tuning or reinforcement learning, which will be a primary focus of our future work.
>
> - Table 2: The self report evaluation performance of 7 LLM-based simulations with different simulation methods.
>   |                  |  None | Prompt |  RAG  | MACM (Ours) |
>   |:----------------:|:-----:|:------:|:-----:|:-----------:|
>   |       GPT-4      | 20.00 |  78.67 | 82.67 |    **86.67**    |
>   |    GPT-4-Turbo   | 12.00 |  78.67 | 85.33 |    **88.00**    |
>   |    Qwen-turbo    | 24.00 |  69.33 | 72.00 |    **74.67**    |
>   | Llama-2-70b-Chat | 21.33 |  48.00 | 36.00 |    **58.66**    |
>   |    Mistral-7b    |  8.00 |  50.67 | 36.00 |    **52.00**    |
>   |    Llama-2-13b   | 17.33 |  **30.67** | 21.33 |    22.67    |
>   |    Vicuna-13b    | 18.67 |  **56.00** | 29.33 |    45.33    |

---

> ### Author Response · Authors · 2024-11-30
>
> ### **5. Evaluation Methodology**
>
> > What evidence supports using human judges to assess LLM response coherence with life stories? It is not clear that humans are actually good at this task!
>
> - We respectfully disagree with the claim that "it is not clear that humans are actually good at this task." Human judges are widely recognized as a reliable choice for evaluating language model outputs because of their unique ability to assess nuanced elements such as coherence, contextual relevance, and adherence to narrative logic—criteria that automated metrics often fail to evaluate comprehensively.
> - While it is true that humans are not perfect in subjective tasks, our evaluation protocol was designed to mitigate potential biases and ensure robustness. This was achieved through cross-validation by multiple judges and the implementation of clear, structured scoring criteria. These measures enhance the reliability and meaningfulness of human evaluations in this specific context.
> - We appreciate the opportunity to address this concern and would welcome any suggestions for further improving our evaluation methodology.
> - **Who were the judges in the observer report evaluation?** We selected 8 human judges with a fair understanding of psychology for the observer report evaluation process. These judges included individuals with psychology master’s degrees, computer science graduate students, and professionals from the laboratory of mental health.
> - **Were they capable of understanding the characters' life stories?** Yes. We reported the average performance of 4 human judges on the self-report evaluation in Table 2 from the original submission. They achieved full scores in the tests, which demonstrates that humans can understand virtual characters' life stories.
> - **What measures have been taken to ensure the validity and consistency of the human evaluation?** For each human judge, we ensured that they 1) **had a fair understanding of psychology** and understood how personality influences an individual’s cognition, emotion, motivation, and behaviors. 2) We provided **comprehensive evaluation guidelines** to guarantee clarity and consistency. All human judges were required to read the corresponding guides before commencing their assessments. The evaluation guidelines are provided in Appendix Tables 13, 14, 15, and 16 in the original submission. Finally, to ensure the validity of the evaluation results, the observer report evaluation **adopted a cross-evaluation manner,** where the final score of each evaluation was based on the scores from 4 human judges.
>
> > What metrics ensure reliable human evaluation? How is inter-rater reliability established? Are there only 2 raters (ICC will be very unstable)? How is ICC exactly calculated?
> - In this paper, we conducted observer report evaluations on human simulations with 3 different simulation methods (Prompt, RAG, and MACM). In the experiments, we selected GPT-4-Turbo as the baseline model and tasked the LLMs with simulating characters from the Human Simulacra dataset, resulting in 3 x 11 = 33 observer report evaluations. For each evaluation, we selected 4 human judges from a pool of 8 to participate. To ensure fair evaluation of different simulation methods, when assessing the same character based on the 3 different simulation methods, we ensured that the same group of 4 judges was used. We then calculated the Intraclass Correlation Coefficient (ICC) between 2 judges for the same assessment task. Finally, we reported the average ICC in the paper.

---

> ### Author Response · Authors · 2024-11-30
>
> > Finally, there are no error margins in any of the tables / results.
>
> Thank you for pointing this out. We will include the standard deviation in the updated version of the paper. Here, we present the standard deviation of the self-report evaluation in the following table.
>
>   |                  |    None    |   Prompt   |     RAG    | MACM (Ours) |
>   |:----------------:|:----------:|:----------:|:----------:|:-----------:|
>   |       GPT-4      | 20.00±2.31 | 78.67±1.33 | 82.67±2.67 |  86.67±1.33 |
>   |    GPT-4-Turbo   | 12.00±0.00 | 78.67±1.33 | 85.33±1.33 |  88.00±2.31 |
>   |   Claude-3-Opus  | 10.67±1.33 | 77.33±2.67 | 52.00±2.31 |  81.33±1.33 |
>   |    Llama-2-7b    | 13.33±1.33 | 44.00±2.31 | 16.00±0.00 |  25.33±1.33 |
>   |     Vicuna-7b    | 12.00±2.31 | 41.33±2.67 | 21.33±1.33 |  29.33±1.33 |
>   |    Mistral-7b    |  8.00±0.00 | 50.67±1.33 | 36.00±2.31 |  52.00±2.31 |
>   |    Llama-2-13b   | 17.33±1.33 | 30.67±2.67 | 21.33±1.33 |  22.67±1.33 |
>   |    Vicuna-13b    | 18.67±1.33 | 56.00±2.31 | 29.33±1.33 |  45.33±1.33 |
>   |  Claude-3-Haiku  | 21.33±1.33 | 65.33±1.33 | 53.33±1.33 |  64.00±0.00 |
>   |   Mixtral-8x7b   | 18.67±1.33 | 60.00±2.31 | 41.33±1.33 |  49.33±1.33 |
>   |    Llama-2-70b   | 12.00±2.31 | 48.00±2.31 | 17.33±1.33 |  56.00±2.31 |
>   | Llama-2-70b-Chat | 17.33±1.33 | 48.00±2.31 | 36.00±2.31 |  58.66±1.33 |
>   |    Qwen-turbo    | 24.00±2.31 | 69.33±1.33 | 72.00±2.31 |  74.67±1.33 |
>   |  Claude-3-Sonnet | 21.33±1.33 | 74.67±1.33 | 52.00±2.31 |  76.00±2.31 |
>
> We look forward to further communication with you and are open to any discussion that can address your doubts.

---

### Official Review · Reviewer_jHzr · 2024-11-06

**Soundness:** 3
**Presentation:** 3
**Contribution:** 3
**Rating:** 8
**Confidence:** 2

**Summary:**

The paper introduces a benchmark to assess the potential of LLMs in simulating human behaviours and personality traits for psychological and sociological applications. This work includes the creation of the "Human Simulacra" dataset, which features detailed virtual characters with diverse life stories constructed with human feedback to enhance realism and ethical accuracy. The authors present a Multi-Agent Cognitive Mechanism (MACM), which simulates human memory and cognitive functions, allowing virtual characters to process emotions and memories for more realistic responses. Evaluation is conducted through a psychology-guided framework that includes self-reports for self-awareness and observer-based assessments where human judges evaluate character responses in various scenarios. Experiments comparing MACM to other simulation methods show that MACM enables LLMs to better replicate human-like behaviour, although limitations remain, particularly in capturing the nuanced adaptability of real human responses to social pressures. This benchmark aims to foster future research on using LLMs as proxies for human participants in psychological experiments while acknowledging ethical implications and the need for authentic simulations.

**Strengths:**

Originality: This paper brings together ideas from psychology, cognitive science, and artificial intelligence to create a unique benchmark for evaluating how well language models (LLMs) can act like humans. Unlike previous studies that mainly focus on simple character traits or responses, this paper takes a deeper approach using Jungian psychology to model personalities with eight different dimensions. This gives a fresh perspective on capturing complex human traits. Additionally, the Multi-Agent Cognitive Mechanism (MACM) is a new tool that helps the models better remember, process emotions, and respond in context, making their behavior more human-like.

Quality: The paper is thorough and well-executed. The Human Simulacra dataset is carefully built, with multiple rounds of expert review to ensure quality, accuracy, and ethical considerations. Each character's story is carefully crafted and reviewed to provide a deep foundation for testing the LLMs’ performance in simulating humans. The MACM’s design, which coordinates memory, emotion, and logical processing, is a clear improvement over simpler models that rely on only one type of agent or basic retrieval methods.

Clarity: The paper is well-organized and clearly explains its methods and objectives. From the motivation to simulate human personalities, through the dataset creation, model mechanism, and evaluation framework, each part is easy to follow.

Significance: This paper makes an important contribution to AI and psychology, especially by opening up possibilities for LLMs to replace humans in some psychological studies. By creating a foundation for simulating complex human traits, this benchmark could allow LLMs to ethically stand in for human participants in specific research settings.

**Weaknesses:**

No outstanding weaknesses.

**Questions:**

No questions.

---

> ### Author Response · Authors · 2024-11-23
> **Response to reviewer jHzr**
>
> Thank you for sharing your valuable feedback. We appreciate your acknowledgment of our benchmark as a meaningful exploration that can advance future research on using LLMs as proxies for human participants in psychological experiments. We noticed that your review confidence is relatively low. **We look forward to further communication with you and are open to any discussions that can help address your concerns.**

---

### Comment · Area_Chair_Upwe · 2024-11-25

Dear Reviewers,

The rebuttal discussion period is coming to a close and the paper currently has a mix of positive and negative reviewers. The authors have spent a lot of time responding to each concern -- can you take a look at the author responses and let them know any remaining concerns you have?

Best,
AC

---

### Author Response · Authors · 2024-12-04
**General Responses and Summary of Revisions**

We sincerely thank all reviewers for their feedback, which has helped us significantly improve our work. We are encouraged by the reviewers' recognition of several key strengths:

- "_This paper makes an important contribution to AI and psychology, especially by opening up possibilities for LLMs to replace humans in some psychological studies._" (Reviewer jHzr)
- "_This paper proposes a high-quality simulated human profile dataset, developed with real human input, alongside a more advanced evaluation benchmark to assess the ability of large language models (LLMs) to emulate specific individuals._" (Reviewer YX9L)
- "_This study introduces a new benchmark to assess the ability of large language models to mimic human personalities in psychological experiments...highlighting both the potential and limitations of using large language models in psychological research._" (Reviewer Dfn4)
- "_The proposed benchmark for assessing the role-playing capabilities of large language models (LLMs) is constructed more rigorously than current benchmarks (to the best of my knowledge). It is based on more robust psychological theories and involves greater human effort to ensure high-quality data._" (Reviewer Wr7p)

Based on their constructive comments, we have made substantial improvements to strengthen our paper's contributions and address the limitations. The major enhancements include:
-  **Extended Discussion.** We added comprehensive analysis of MACM's structure (Section 4.3), the rationale for using Jung's theory (Section 6, Appendix A.5), and potential applications of LLM-based human simulation (Appendix H).
-  **Additional Experimental Results.** We provided additional experimental results highlighting the contribution of each agent within MACM (Appendix B.3) and observations of the bandwagon effect of different simulacra (Appendix E). These results further validate the conclusions drawn in our paper.
-  **More Implementation details.** We supplemented the paper with additional details on character attribute design (Section 3.1), human involvement (Section 3, Appendix D.2.2), hypothetical scenarios (Appendix D.2.1), and the bandwagon effect replication (Section 5.2, Appendix E).

We thank the reviewers again for their valuable input that has helped us improve the paper. We believe these additions have fully realized the potential of our research contribution to the field of LLM-based human simulation.

---

### Meta-Review · Area_Chair_Upwe · 2024-12-23

**Metareview:**

The paper introduces a benchmark to assess the potential of LLMs in simulating human behaviours and personality traits for psychological and sociological applications. This work includes the creation of the "Human Simulacra" dataset, which features detailed virtual characters with diverse life stories. The work then proposes an LLM-based system for more advanced human simulation, named MACM, which simulates human memory and cognitive functions, allowing virtual characters to process emotions and memories for more realistic responses. The authors evaluate human simulation capabilities by integrating both self reports and observer reports. Experiments comparing MACM to other simulation methods show that MACM enables LLMs to better replicate human-like behaviour. The benchmark could be useful to foster future research on using LLMs for human simulation. Reviewers have mentioned a few questions/suggestions which the authors replied and acknowledged, including more explanation of why a particular psychology theory was chosen over others, ablation on human expert evaluation, motivation of MACM, more precise description of the goal/scope of the work, etc.

**Additional Comments On Reviewer Discussion:**

The authors provided very comprehensive and detailed responses to reviewers' comments and follow-up questions. Overall, the reviewers are positive, and at the same time asked quite a few meaningful questions as summarized above. The authors seem to answer these questions properly. It's necessary for the authors to include these points in the revised paper.

---

### Decision · Program_Chairs · 2025-01-22

Accept (Poster)